



**Functional spatial contextualisation of the effects of**
**multiple stressors in marine bivalves**
Antonio Giacoletti* and Gianluca Sarà
Dept. of Earth and Marine Sciences (DISTEM) University of Palermo, Palermo, Italy;
*Correspondence to*: Antonio Giacoletti (antonio.giacoletti@unipa.it)
**Abstract.** Many recent studies have revealed that the majority of environmental stressors experienced by marine
organisms (ocean acidification, global warming, hypoxia etc.) occur at the same time and place, and that their
interaction may complexly affect a number of ecological processes. Here, we experimentally investigated the
effects of pH and hypoxia on the functional and behavioural traits of the mussel *Mytilus galloprovincialis*, we
then simulated the potential effects on growth and reproduction dynamics trough a Dynamic Energy Budget
(DEB) model under a multiple stressor scenario. Our simulations showed that hypercapnia had a remarkable
effect by reducing the maximal habitat size and reproductive output differentially as a function of the trophic
conditions, where modelling was spatially contextualized. This study showed the major threat represented by the
hypercapnia and hypoxia phenomena for the growth, reproduction and fitness of mussels under the current
climate change context, and that a mechanistic approach based on DEB modelling can illustrate complex and
site-specific effects of environmental change, producing that kind of information useful for management
purposes, at larger temporal and spatial scales.
Key-words: Acidification; Climate change; DEB Model; Hypoxia; *Mytilus galloprovincialis*; Multiple-Stressor;
Mussel.



# 1 Introduction

Since the dawn of research investigating the possible effects of ocean acidification (OA) on aquatic organisms
(e.g. Bamber, 1990), most studies have shown that elevated $pCO_2$ levels, as predicted for the next century, may
affect  to some extent the functional traits (Schoener, 1986; Koehl, 1989) of marine organisms (Feely et al.,
2004; Navarro et al., 2013). Referring to functional traits, we consider all those specific traits that define each
species in terms of their ecological roles (Diaz & Cabido, 2001), and thereby the species' identity. In marine
ectotherms such as bivalves, crabs, sea urchins and fish, these traits include tolerance and sensitivity to
environmental conditions (e.g. physiological tolerance limits - Kearney & Porter, 2009) defining the ability of
each species to support their own metabolic machinery (Sokolova et al., 2012; Sarà et al., 2014), the ability to
obtain energy from food, the so-called functional response (Holling, 1959) or those behavioural (e.g. swimming
behaviour, habitat use, mating system) and morphological (e.g. shape, thickness) traits (Schoener, 1986) which
led to optimise the energetic income (Krebs & Davies, 1992) and lastly to reach the ultimate fitness (Roff, 1992).
Research performed over the last decade and summarized in the recent IPCC report (IPCC, 2014) clearly shows
that ocean acidification will affect marine organisms and ecosystems (Connell et al., 2017) in the coming
decades, and such projections have stimulated new research that aims to understand the impact on calcifying
marine organisms. Reductions in growth and calcification rates are just those kinds of the physiological impacts
of ocean acidification (Thomsen et al., 2013; Byrne, 2012; Beniash et al., 2010). While much research showed
that low pH may impair most functional traits, functions connected with energy uptake such as feeding and
assimilation seem to be reduced at a lager extent in many species with expected implications for the amount of
energy available for growth and reproduction (Kurihara et al., 2008; Appelhans, 2012; Navarro et al., 2013;
Zhang et al., 2015). Such information has been obtained through both acute and chronic exposure to OA but no
studies are yet available to assess the potential effects of OA on the magnitude of other Life History (LH) traits,
such as maximum habitat body size, fecundity, time to reach maturation and the number of spawning events
under future conditions of environmental change (*sensu* Kearney and Porter, 2009; Sarà et al., 2011; 2013b). To
obtain such LH traits, experiments should be long enough to assure a functional effect of lower pH for many
weeks or months but probably no existing lab mesocosm could currently assure the stability of seawater
acidification system for such a long time. Thus, apart from long term experiments carried out in few field sites
worldwide (e.g. Ischia [Hall-Spencer et al., 2008] and Vulcano [Duquette et al., 2015] islands) in the Southern
Mediterranean Sea and in other Seas (Maug Island [Pala, 2009] or CO2 vents in the southwest Pacific [Connell
et al., 2017]) where lowered pH seawater is naturally available through $CO_2$ natural emissions from vents, the





recent introduction of mechanistic functional trait-based (FT) models based on the Dynamic Energy Budget
theory (DEB; Kooijman, 2010) can offer a reliable opportunity for disentangling the effect of seawater
acidification on LH traits. The novelty of the FT-DEB approach relies on its intrinsic mechanistic nature deriving
from the fact that it is based on flux of energy and mass through an organism which are traceable processes that
are subject to conservation laws (according to the new posited concept of ecomechanics; Denny & Helmuth,
2009; Denny & Benedetti-Cecchi, 2012; Carrington et al., 2015). This provides an exceptionally powerful tool to
predict organismal functional traits, capturing variation across species to solve a very wide range of problems in
ecology and evolutionary biology (Lika et al., 2011; Kearney, 2012; Pouvreau et al., 2006; Pequerie et al., 2010;
Sarà et al., 2011; 2012; 2013a; 2013b; 2014). FT-DEB could provide information about the effect of seawater
acidification on the fecundity (as expressed by the number of gametes per life span, the so-called Darwinian
fitness; Bozinovic et al., 2011) and the degree of reproductive failure of species providing theoretical predictions
about LH traits having implications on population dynamics and community structure throughout the species
range (Sarà et al., 2013a). Here, we specifically exploited the FT-DEB model spatially and explicitly
contextualised along the Italian coasts under subtidal conditions (Kearney et al., 2010; Sarà et al., 2011; 2012;
2013a; 2013b), using four-year thermal series and satellite Chlorophyll-a (CHL-a) concentrations, to test the
multiple effect due to the combination of pH and hypoxia on the physiological and behavioural traits of our
target species, the bivalve *Mytilus galloprovincialis* (Lamarck 1819). Recent insights obtained by the
experimental research have shown that OA mainly affects feeding (FR), assimilation (AE) and maintenance cost
rates. Here, we translated the combined effects of hypoxia and hypercapnia on AE and oxygen consumption
rates as measured under different treatments into effects on assimilation and somatic maintenance costs as
expressed by the DEB [$\dot{p}_M$] parameter. This latter is a crucial functional trait used in recent bioenergetics based
on the DEB theory that mechanistically can be used to investigate the role played by multiple stressors on LH
traits of organisms by using first principles (Sarà et al., 2014). We further documented the effects of those
stressors on *M. galloprovincialis* shells through the use of a scanning electron microscope (SEM), and compared
the maximum breaking load of treated *vs.* control specimens. A behavioural analysis completed the frame
concerning the individual's response to both single and combined stressors. Carried out in a context of OA, this
exercise comprises a first step in linking the fields of ecomechanics and climate change ecology, which should
yield a more mechanistic understanding of how biodiversity will respond to environmental change (*sensu*
Buckley et al., 2012).



## 2 Materials and methods

This study articulated three steps: 1) laboratory investigation on the effects of pH and hypoxia on functional and
behavioural traits of *Mytilus galloprovincialis*; 2) collection of water temperature data, and Chlorophyll-a (CHL-
a) data from two Mediterranean sites (Trieste and Palermo), as a further forcing variable in the DEB model and
lastly 3) model running to simulate growth and fitness of *M. galloprovincialis* under stressful conditions by
using estimated DEB parameters arising from the activities in the first step.

**2.1 Sampling and experimental set-up.** Specimens of *M. galloprovincialis* (45 - 55 mm) were provided by the
Ittica Alimentare Soc. Coop. Arl. (Palermo) and transferred within 30 minutes to the laboratory. Mussels were
then carefully cleaned and placed in a 300L tank filled with natural seawater at room temperature (18-20°C),
field salinity (37-38 PSU), and fed *ad libitum* with cultured *Isochrysis galbana* (Sarà et al., 2011). According to
common experimental procedures for studying the bioenergetics of bivalves (Sarà et al., 2008; Ezgeta-Balic et
al., 2011), mussels were acclimated for two weeks to reduce stress generated by manipulation and transport (Sarà
et al., 2013a). Once acclimated, 200 specimens were randomly divided in groups of 25 organisms, transferred to
8 independent rectangular glass tanks of 120L capacity (100 cm long, 30 cm deep, 40 cm wide) and kept in a
conditioned room at 21°C. Tanks 1 to 4 were filled with aerated and recirculating sea water, while Tanks 5 to 8
were not aerated and covered with a plastic film disposed on the water surface, in order to avoid gas-exchanges
between air and water. Tanks 1-2 were used as a control (CTRL), while hypercapnia was imposed in Tanks 3-4
(Tr1), hypoxia (2 ppm) in Tanks 5-6 (Tr2), and both factor (pH 7.5 and hypoxia) in Tanks 7-8 (Tr3). Mussels
were acclimated to two different nominal pH treatments: (i) pH 8.0 in Tanks 1-2 (CTRL) and 5-6 (Tr2),
corresponding to present average pH at the sampling site; and (ii) pH 7.5 in Tanks 2-3 (Tr1) and 7-8 (Tr3),
deviating from present range of natural variability and relevant for 2100 ocean acidification scenarios. This last
point is considered the critical dissolution threshold of calcium carbonate in shelled animals as reported in
literature (Melzner et al., 2011; Gazeau et al., 2013). The carbonate system speciation ($p\mathrm{CO2}$, $HCO_3^-$, $CO_3^{2-}$,
$\Omega\mathrm{Ca}$ and $\Omega\mathrm{Ar}$) was calculated from $pH_{NBS}$, temperature, salinity and alkalinity ($T_A$ = 2.5 mM; Rivaro et al.,
2010) using CO2SYS (Lewis and Wallace, 1998) with dissociation constants from Dickson & Millero (1987).
The pH was manually controlled 8 times a day by an electronic pH-meter (Cyberscan 510, Eutech Instruments)
and gaseous $CO_2$ was injected directly into the aquarium when required. Tanks were siphoned at the end of each
working day, removing all the faecal material in order to avoid the accumulation of waste products.



**2.2 Behavioural observations.** The valve gape of mussels was recorded by means of the two simplest
behavioural categories reported in Jørgensen et al. (1988): closed valves and opened valves. Each observation
was carried out by an operator with the aim to record changes in the behavioural repertoire of bivalves in
response to the exposure to a single stressor (pH or hypoxia) and to both pH and hypoxia, compared to
individuals kept in normal environmental conditions. All experiments were conducted at environmental (37-38
PSU) salinity and with well-aerated sea water through a gentle flow (Ameyaw-Akumfi & Naylor, 1987), except
for specimens of Tank 5-6 and 7-8, that were not aerated in order to maintain the hypoxia level set through the
gaseous nitrogen. Behavioural observations were repeated six times a day, on day 7, 14, 21, 28, and involved 5
random specimens for each treatment.

**2.3 Oxygen consumption.** The rate of oxygen consumption was determined twice (week 1 and week 4) in a
respirometric glass chamber (0.3L) in a temperature-controlled water bath, in order to compare the effects of
multiple stressors by converting rates into the DEB parameter [$\dot{p}_M$] (expressed as J cm$^{-3}$ h$^{-1}$) linked to the
energetic cost of maintenance in order to integrate it in the standard DEB model. All determinations were
performed using filtered seawater with the same pH and oxygen content as that of the respective treatment,
stirred with a magnetic stirrer bar beneath a perforated glass plate supporting each individual (Sarà et al., 2008;
Ezgeta-Balic et al., 2011). The decline in oxygen concentration was measured by a PiroScience FirestingO2
respirometer, capable of four sensor connections. We used a total of n = 64 mussels per week, 16 for each
treatment (8 for each tank) acclimated as above, fed *ad libitum* until the day before the experiment. The decline
was continuously recorded for at least 1 h, excluding an initial period (~ 10 min) when usually there is a more
rapid decline in oxygen caused by a disturbance of the sensor's temperature equilibration. Respiration rate (RR,
µmol O$_2$ h$^{-1}$) was calculated according to (Ezgeta-Balic et al., 2011; Sarà et al., 2008; 2013b): $RR =$
$(C_{t0} - C_{t1})\, x\ Vol_r\, x60(t_1 - t_0)^{-1}$, where $C_{t0}$ is oxygen concentration at the beginning of the measurement, $C_{t1}$ is
the oxygen concentration at the end of the measurement, and $Vol_r$ is the volume of water in the respirometric
chamber.

**2.4 Assimilation efficiency.** Assimilation is the final step of food processing and it represents the efficiency with
which organic material is absorbed from the ingested food (Kooijman, 2010). The assimilation of food is
assumed to be independent of the feeding rate *per se*, but proportional to the ingestion rate. Here, 16 specimens
of *M. galloprovincialis* per treatment were collected twice (week 1 and week 4) and placed into separate beakers



containing 1L of filtered seawater and a magnetic stirrer bar. In order to allow the mussels to open their valves
and start their filtration activity, they were given 15 minutes before the introduction of food with an initial
concentration of ~ 15,000 *Isochrysis galbana* cells ml$^{-1}$. After a period of 2 h mussels were moved to cleaned 1L
glass beakers with filtered seawater for a period of 12 h, after that the water contained in each beaker was filtered
on pre-ashed and weighted GF/C fibreglass filters. Once filtered, filters were washed with 0.5 M ammonium
formate (purest grade) to remove adventitious salts (Widdows & Staff, 2006), dried in the oven (95°C for 24 h)
and then incinerated in a muffle furnace (450°C for 4 h). After each step, the samples were weighted using a
balance (Sartorius BL 120S ± 1µg). For the calculation of AE, together with the faeces collected from the
mussels, filters containing algal food were dried and incinerated as above. After respirometric measurement and
the collection of faeces each animal was killed by gentle freezing and dissected, and the shells were separated
from the body tissue in order to calculate their individual dry weights and standardize respiration rates to body
weights.

**2.5 Water temperature data.** The main forcing driver of shellfish LH inside DEB models is represented by
mean seawater temperature (Pouvreau et al., 2006; Kearney et al., 2010; Kooijman, 2010; Sarà et al., 2011;
2013). DEB simulations were run under subtidal conditions (body temperature was expressed by the mean
seawater temperature; Montalto et al., 2014) with 4 years-hourly data (Jan 2006 - Dec 2009) of seawater
temperature measured about 1 m below the surface at the closest meteo-oceanographic station held in Trieste
(LAT 45° 38′ 57.81″; LONG 13° 45′ 28.58″) and Palermo (LAT 38° 07′ 17.08″; LONG 13° 22′ 16.79″). The
period of 4 years is consistent with the normal life span of most Mediterranean shellfishes (Sarà et al., 2012;
2013). Both sites were chosen as they represent two opposite temperature and food conditions for mussel growth
in Italy, with Trieste as representative of lower temperature (average 16.98 ± 6.19 °C) and higher food levels
(average 1.36 ± 0.37 CHL-a), and Palermo of higher temperatures (average 20.19 ± 4.64 °C) and lower food
(average 0.19 ± 0.09 CHL-a). Data are available online from the Italian Institute of Environmental Research
(ISPRA) web page (http://www.mareografico.it/).

**2.6 CHL-a dataset.** Chlorophyll-a (CHL-a) derived from satellite imageries (µg L$^{-1}$) was adopted as a reliable
food quantifier for suspension feeders (Kearney et al., 2010; Sarà et al., 2011; 2012) and was downloaded from
the EMIS website (http://emis.jrc.ec.europa.eu/).





**2.7 Model description.** The Dynamic Energy Budget Theory provides a general framework that allows to
describe how physiological mechanisms are driven by temperature and food availability, and influences growth
and the reproductive performances in marine organisms (Monaco et al., 2014). Following the κ-rule (DEB
theory; Kooijman, 2010) a fixed energy fraction (κ) is allocated to growth and somatic maintenance, while the
remaining fraction (1-κ) is allocated to maturity maintenance plus maturation or reproduction. If the general
environmental condition deviates from common natural patterns (i.e. changes in temperature, food availability
etc.) reproduction and growth are consequently affected. According to DEB theory, a reduction in growth can be
caused either by reduced food assimilation ($\dot{p}_A$), enhanced maintenance costs ($\dot{p}_M$), or enhanced growth costs
($\dot{p}_G$). Using this approach, and through the DEB parameters derived from Sarà et al. (2012), except for the
variation in the maintenance costs ($\dot{p}_M$) and in the assimilation efficiency of food (AE) which were
experimentally estimated throughout this study, we performed simulations aimed at investigating the potential
variations in growth and fecundity of our model species. To run the DEB simulations, local thermal series of
selected sites were used together with satellite CHL-a concentrations, obtaining a first model with environmental
conditions. A second model was run with the $\dot{p}_M$ calculated from the oxygen measurements on specimens of *M.*
*galloprovincialis* from Tanks 3-4 (pH 7.5) simulating a chronic hypercapnia condition for the full cycle (4 years)
and the relative estimated AE. Subsequently, further models were run by simulating one random hypoxia event
for each of the four years of the cycle, then simulating two yearly events, and so on up to six monthly hypoxia
events. The month of each event was randomly chosen for every year with the use of a table of random digits.
The $\dot{p}_M$ calculated from the oxygen consumption rate measurements on specimens from Tanks 7-8 (pH 7.5 and
hypoxia) was used in substitution to $\dot{p}_M$ from pH 7.5 tanks 3-4, coupled with the relative estimated AE, when
simulating both stressors. Outputs of the DEB models (Sarà et al., 2014) were: the maximum theoretical total
length of shellfish (TL), the maximum total weight (TW), the total number of eggs (TRO) produced during a
life-span of 4 years, the total number of reproductive events (RE) and the time needed to reach gonadic maturity
(TM) for each treatment.

**2.8 Effects on shell: mechanical strength and SEM pictures.** The functional impact of exposure to pH and to
validate the pH effect on morphological structure of valves, was tested on mussels exposed to the two nominal
pHs for 4 weeks. Twice (week 1 and week 4), 16 mussels for each treatment were collected and dissected, and
both valves were cleaned and dried with absorbent paper. The left valve was then sliced transversely using a
circular saw (Dremel® 300 series) to section the whole length of the shell. Age was estimated using the analysis





of shell rings proposed by Peharda et al. (2011) by counting the number of rings with the use of a stereo
microscope (Leica EZ4). The right valves were instead evaluated for their mechanical properties at the
Department of Mechanical Engineering. Experimental crushing tests, in order to estimate the shell's maximum
breaking load (in N) as a further validation step, were realised with a home-made press previously calibrated by
an Instron 3367 machine controlled by the Bluehill 2.0 software. The effects of low pH exposure were
documented by the use of a scanning electron microscope (SEM; Zeiss LEO 440) that led to a thorough
investigation on the integrity of the mussels' external protein layer (*periostracum*) and on the underlying mineral
layer, rich in calcite and aragonite.

**2.9 Statistical analysis.** In order to test for significant differences in respiration rate and the assimilation
efficiency, ANOVAs were performed using Treatment (CTRL, Tr1, Tr2, Tr3) and Time (Week 1 and Week 4) as
fixed factors, with respectively four and two levels. In order to test for significant differences in behavioural
categories ANOVAs were performed using Treatment (CTRL, Tr1, Tr2, Tr3) as fixed factors, while Breaking
load was tested with Treatment (CTRL, Tr1, Tr2, Tr3) and Time (Week 1 and Week 4) as fixed factors. When
significant differences were detected, the Student-Newman-Keuls (SNK) post-hoc pair wise comparison of
means was used (Underwood, 1997). Cochran's test was used prior to ANOVA to test the assumption of
homogeneity of variance (Underwood, 1997). When no homogeneous variances were rendered with any type of
transformation, the significance level was set at 0.01 instead of 0.05, as ANOVA can withstand variance
heterogeneity, particularly in large balanced experiments, thereby reducing the possibility of a Type I error
(Underwood, 1997).

**3 Results**

**3.1 Water chemistry.** Experimental target pH values were constantly maintained at significantly different levels
in all tanks (normal pH and 7.5 pH treatments; Table 2; ANOVA, $p < 0.01$; SNK test: CTRL = Tr2, Tr1 = Tr3).
Oxygen $pO_2$ was also maintained at significantly different levels between normoxia ($7.3 \pm 0.02$ mg/l) and
hypoxia ($2.4 \pm 0.02$ mg/l) treatments (Table 2; ANOVA, $p < 0.01$; SNK test: CTRL = Tr1, Tr2 = Tr3). This
translated into significant different pCO2 levels in all treatments (Table 2; ANOVA, $p < 0.01$), and in different
$CO_3^-$, $\Omega Ca$ and $\Omega Ar$ levels in all tanks (Table 2; ANOVA, $p < 0.01$) except between Tr1 and Tr3 (SNK test: Tr1
= Tr3).

**3.2 Valve gaping.** During behavioural observations on *M. galloprovincialis*, specimens showed a significant
difference in the behavioural categories, showing respectively 64.5 ± 5.6 % (CTRL), 57.3 ± 0.2 (Tr1), 24.5 ± 0.3
(Tr2) and 12.7 ± 0.2 % (Tr3) of opened valves (Fig. 1; Table 3, ANOVA, p < 0.001). The percentage of closed
valves was instead 35.5 ± 5.6 % (CTRL), 42.7 ± 4.8 (Tr1), 75.5 ± 5.7 (Tr2) and 87.3 ± 3.1 % (Tr3) (ANOVA, p
< 0.001).

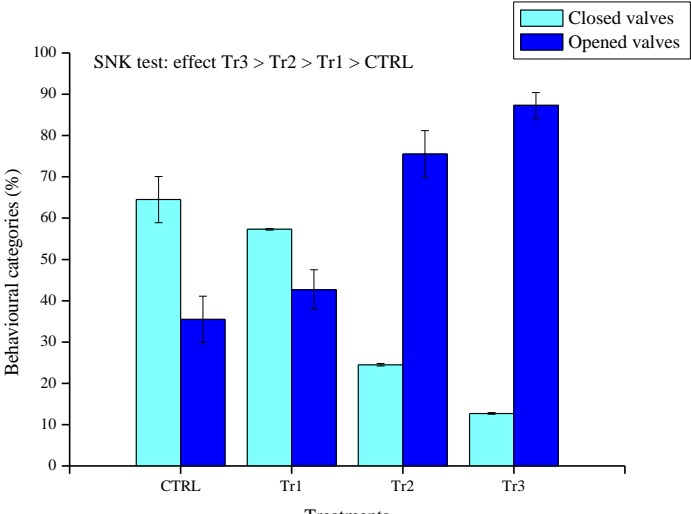


Fig. 1 Behavioural observations of *Mytilus galloprovincialis* under different treatments of oxygen (normoxia –
hypoxia 2ppm) and pH (7.5 – 8.0). The two behavioural categories represented were: closed and opened valves.

**3.3 Oxygen consumption.** Results showed a significant reduction in the oxygen consumption rate by specimens
of *M. galloprovincialis* exposed to treatments (Table 4, ANOVA, p < 0.01), although the SNK test revealed no
significant differences among the various groups (Fig. 2a). No significant effects were highlighted for the time
factor (Table 4, ANOVA, p > 0.05), so in Fig. 2a we reported only results for week 4. The rate of oxygen
consumption was reduced by up to 42% in Tr1, to 35% in Tr2, and to 41% in Tr3, causing a decrease in the $\dot{p}_M$
by up to 29% in Tr1, to 47% in Tr2, and to 49% in Tr3 across the four weeks of exposure.



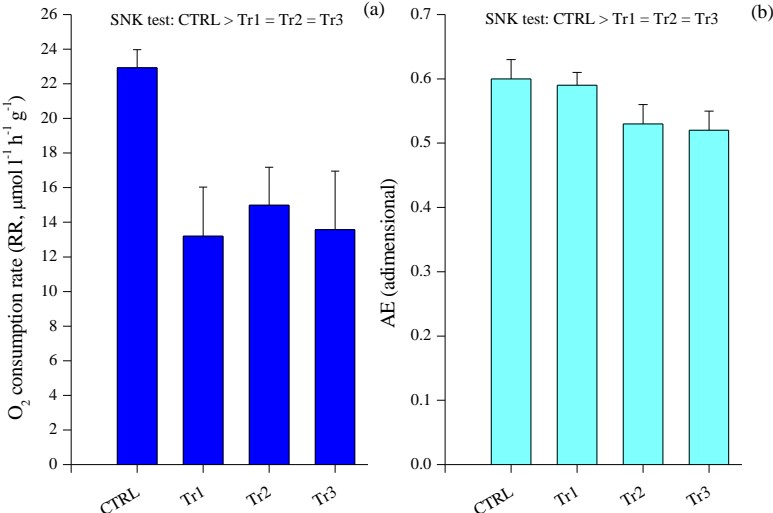

Fig. 2 (a) Oxygen consumption rates (RR) and (b) Assimilation efficiency (AE) of *Mytilus galloprovincialis*

under different treatments of oxygen (normoxia – hypoxia 2ppm) and pH (7.5 – 8.0) at week 4.


**3.4 Assimilation efficiency.** Assimilation efficiency of food (AE) resulted in significantly affected treatments
(Table 4, ANOVA, $p < 0.001$) after four weeks of exposure. No significant effects were highlighted for the time
factor (Table 4, ANOVA, $p > 0.05$), so in Fig. 2b were reported only results for week 4. In particular, AE
decreased of 2.4% in Tr1, of 12.4% in Tr2, and of 14.4% in Tr3, although the SNK test revealed no significant
differences among the various groups (Fig. 2b).

**3.5 DEB simulation results.** Once $\dot{p}_M$ and AE were experimentally estimated, we introduced obtained values
under the different treatments to run DEB models and to obtain the derived effects In terms of LH traits. Thus,
we performed DEB simulations under local thermal conditions (as expressed by the thermal series recorded in
Trieste and Palermo; see M&M for details) and using satellite CHL-a concentrations (2006-2009) as a proxy of
food. Results showed a remarkable effect exerted by hypercapnia and an increasing addictive effect of hypoxia
related to the intensity of disturbance (*i.e.* number of yearly hypoxic events) on LH traits of *M. galloprovincialis*
by the end of 4$^{th}$ year (Table 5). Total length (TL) and total weight (TW) in Trieste and in Palermo were
similarly reduced by hypercapnia (Fig. 3), with a progressive addictive effect of hypoxia (Table 5). The total
number of eggs produced (TRO) and the total number of reproductive events (RE) in Trieste were strongly




reduced by hypercapnia (Fig. 3), with the same progressive addictive effect from hypoxia (Table 5). Maturation
time (TM) was affected both in Trieste and Palermo by hypercapnia, with the same hypoxia contribution
previously shown. Palermo showed no reproductive events in the DEB simulations.

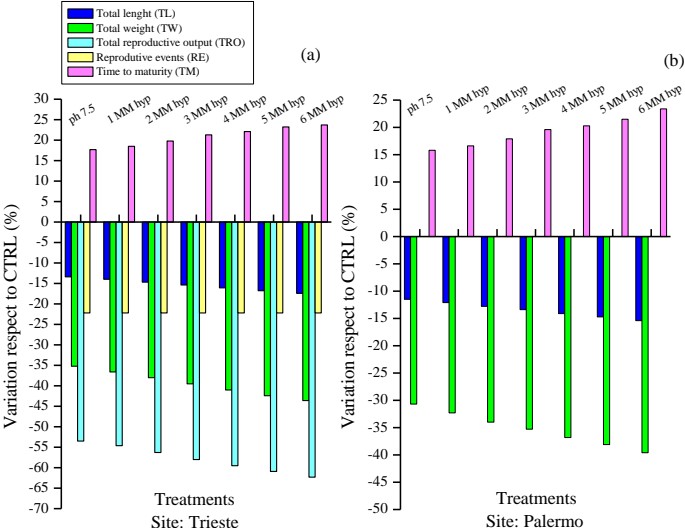


Fig. 3 Results from DEB simulation for (a) Trieste and (b) Palermo sites, percentage variation of DEB outputs
respect to CTRL. TL and TW were reduced by 13.4% and 35.2% in Trieste, and by 11.5% and 30.7% in Palermo
by hypercapnia, with a progressive addictive hypoxia effect up to 8.9%. TRO and RE were reduced by 53.4%
and 66.7% in Trieste by hypercapnia, with a progressive addictive hypoxia effect up to 8.8%. TM increased by
17.8% in Trieste and by 15.7% in Palermo with a similar hypoxia effect (up to 7.6%).

**3.6 Effects on shell.** Specimens of *M. galloprovincialis* collected ranged in age from 1 to 3 years with a mean
age of 1.8 ± 0.04 years (n = 128). Overall, 97% of individuals were > 2 years old. Results from the breaking load
experiment revealed a significant effect of pH (58.8 ± 5 N) and of combined stressors on the breaking load (50 ±
2.7 N), compared to hypoxic (64.4 ± 3.7 N) and CTRL specimens (77.2 ± 2.2 N) (Fig. 4) (Table 3, ANOVA, p <
0.001). In addition, the effect was stronger at week 4 than after one week of exposure (Table 3, ANOVA, p <
0.01). Deeper investigations through scanning electron microscopy validated an effect by showing an increasing
erosion of the shell after exposure to $CO_2$-induced acidification. The external dissolution pattern usually started
from the umbonal region and progressed toward the margin of the shell, usually associated with some degree of
damage to the *periostracum*. The damage was present at differing extensions in all specimens exposed to



treatments, except in the control mussels (Fig. 5 b, c, d). The alteration of the underlying carbonate layer was
instead visible only in Tr1 and Tr3, with details in Fig. 6 (b, d). This kind of alteration was never recorded under
control pH (Fig. 4a).

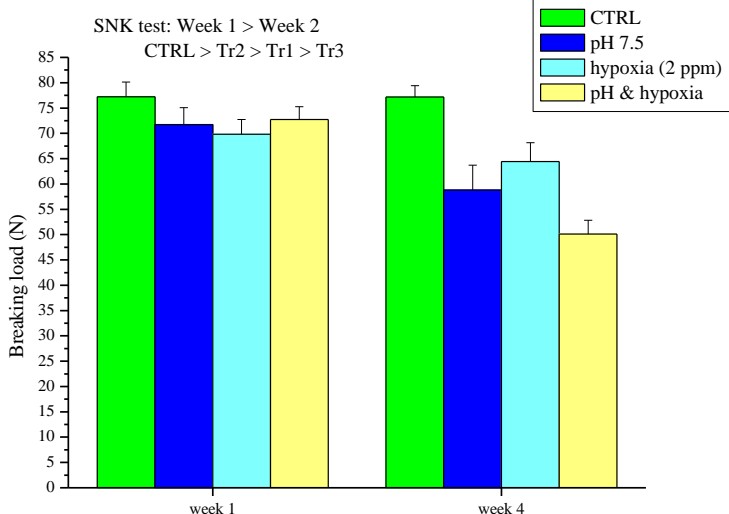


Fig. 4 Breaking load of valves (in Newton, N) exposed to different treatments of oxygen (normoxia – hypoxia

2ppm) and pH (7.5 – 8.0) at week 1 and 4.



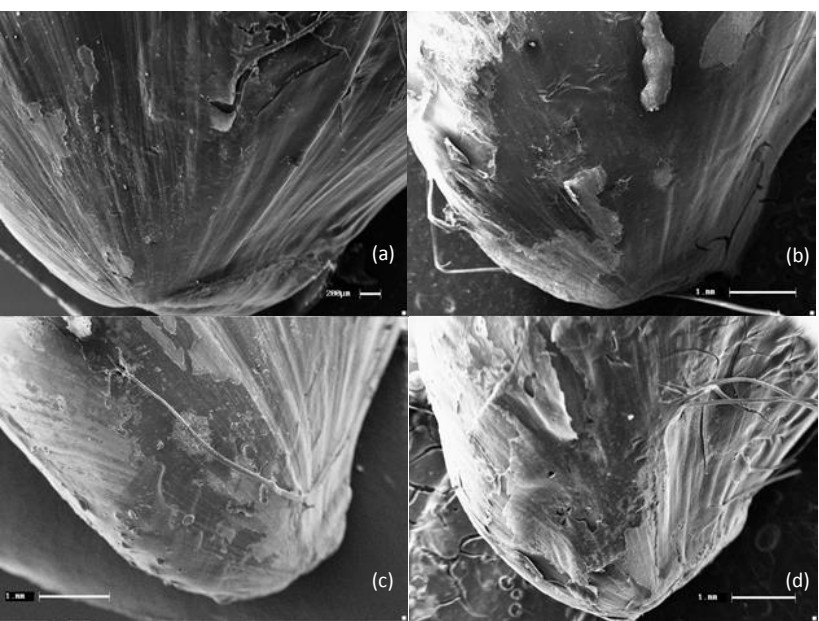


Fig. 5 SEM pictures of different shells exposed to (a) control condition (CTRL); (b) pH 7.5 and normoxia

condition (Tr1); (c) normal pH and hypoxia condition (Tr2); (d) both pH 7.5 and hypoxia conditions (Tr3).


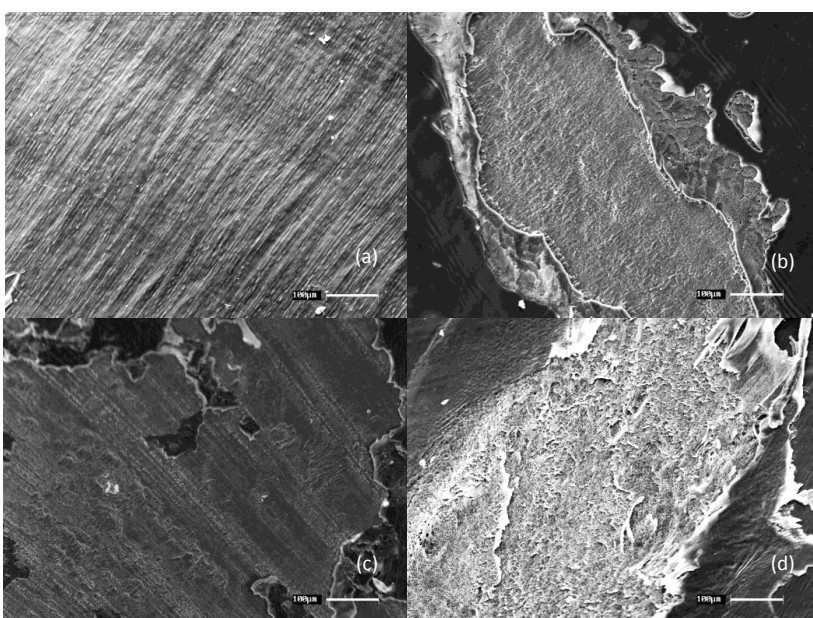


Fig. 6 Details of different shells exposed to (a) control condition (CTRL); (b) pH 7.5 and normoxia condition

(Tr1); (c) normal pH and hypoxia condition (Tr2); (d) both pH 7.5 and hypoxia conditions (Tr3).



## 4 Discussion


Marine organisms, and in particular intertidal species (Montecinos et al., 2009), have been formally recognized as
being equipped with well-developed and conserved compensatory mechanisms to contrast ocean acidification such
as (i) passive buffering of intra- and extracellular fluids; (ii) transport and exchange of relevant ions; (iii) transport
of $CO_2$ in the blood in those species that have respiratory pigments; (iv) metabolic suppression to wait out periods
of elevated $CO_2$ (e.g. Lindinger et al., 1984; Cameron, 1989; Walsh and Milligan, 1989; Hand, 1991; Heisler,
1993; Guppy and Withers, 1999; Pörtner et al., 2004). Several authors recorded suppression of feeding activity and
growth, depressed metabolism, increased N excretion and loss of tissue weight for marine bivalves exposed to
reduced seawater pH (Bamber, 1990; Michaelidis et al., 2005; Berge et al., 2006; Gazeau et al., 2010). Bivalves
are in fact capable of maintaining a constant internal pH by decreasing their metabolic rates and/or dissolving their
shell; the shell acting then as a source of $CO_3^{2-}$ (Bamber, 1990; Michaelidis et al., 2005; Berge et al., 2006)
counterbalancing the crossing effect due to lowering dissolved $CO_2$ through biological membranes (Fabry et al.,
2008). Compensation of low pH through adjustments in ionic composition appears to be a trade-off that is not
likely sustainable on longer time-scales, such as that associated with anthropogenic increases in seawater $pCO_2$
(Fabry et al., 2008). In agreement with current literature showing deleterious effects of $CO_2$-induced acidification
on a wide range of invertebrates (Barnhart & McMahon, 1988; Barnhart, 1989; Rees & Hand, 1990), and similarly
to other studies by *M. galloprovincialis* (Gestoso et al., 2016; Michaelidis et al., 2005), our results showed how
hypercapnia (pH reduced by 0.6 units, relative to the natural pH of the lower Tyrrhenian waters) was able to
induce a decline in metabolic rates of mussels. This kind of decline has already been noticed by other authors as an
adaptive strategy for survival under transiently stressful conditions (Michaelidis et al., 2005). According to Pörtner
et al. (2004), metabolic reduction due to hypercapnia could be a result of acid-base disturbances and therefore be
similar to the response of intertidal individuals to anaerobic conditions. Direct effects of hypoxemia have been
further proven to cause fatal decrements in an organism's performance in growth, reproduction, feeding, immunity
and behaviour (*sensu* Pörtner & Farrell, 2008). Synergistic stressors like ocean acidification and hypoxia are
capable of narrowing the thermal window of functioning according to species-specific sensitivities, modulating
biogeographies, coexistence ranges, community shifts and other interactions (Pörtner & Farrell, 2008). The mussel
*Mytilus edulis* has been proven able to compensate both short- and long-term exposure to hypercapnia by
dissolution of its shell (Lindinger et al.,1984; Michaelidis et al., 2005), resulting in reduced growth and
metabolism. A similar mechanism of release of inorganic molecules into the pallial cavity (as $CaCO_3$ from valves)
has been documented during periods of anaerobic metabolism, to maintain the acid-base balance (Chaparro et al.,





2009), determining further physiological and energetic cost such as decreased growth, respiration rate and protein
synthesis (Pörtner et al., 2005). During periods of environmental oxygen limitation, many organisms are able to
suppress ATP demand, shut down expensive processes, such as protein synthesis (Hand, 1991), but at the same
time limiting growth and reproductive potential. Although suppression of metabolism under short-term
experimental conditions is a "sublethal" reversible process, reductions in growth and reproductive output will
effectively diminish the survival of the species on longer time-scales (Fabry et al., 2008). The contemporary
occurrence in our simulations, of monthly hypoxia events, revealed a growing additive contribution to what was
already elicited by hypercapnia on growth and reproduction. Current literature has not currently explored the
combined effects of multiple stressors on long-term experiments by modulating the intensity and duration of
disturbance. This would probably translate as a very complex experimental set-up which would be hardly
practicable, especially on long-term scales. On the other hand, mechanistic models offer a more sustainable and
reliable alternative to long-term, in-field research when studying the effects of multiple-stressors , with the
advantage of testing, at the same time, the magnitude and duration of disturbance on LH-traits of a model species.
Our results highlighted the general hypoxia growing effect following the increasing duration of disturbance, with a
particular focus in Trieste on TW and TRO, while in Palermo on TW and TM (Table 5). A further important
peculiarity of the DEB simulations deals with the possibility to spatially contextualise the effects of single and
multiple stressors on selected outputs by integrating local thermal conditions and food concentrations. In particular
the DEB model easily allowed the estimation of the fecundity potential of cultivated and natural organisms, that is
often omitted in other ecological studies, but that represents a crucial quantity for resource (e.g. aquaculture) and
conservation purposes. To verify impacts on shellfish fecundity, we contextualised our simulation by introducing
Trieste hourly temperature series after those of Palermo, with the respective local actual CHL-a concentrations, as
long as in the first site no reproductive events came out from our simulations, probably due to food limitations and
temperature threshold. A combined effect of the simultaneous stressors, such as those considered across this study,
has proven, through our experimental and mechanistic integrated approach, to affect the organism's performance
in growth, reproduction and behaviour. Those specific and synergic effects of each stressor seem capable,
especially at extreme temperatures, of narrowing thermal windows, modulating biogeographical distribution,
coexistence ranges, community shifts, food webs and species interactions (*sensu* Pörtner & Farrell, 2008).



**6 Conclusions**

Additional research is still required to improve our knowledge of organismal response to multiple stressors, in
particular, of many marine ectotherms with indeterminate growth amongst invertebrates (e.g. crustaceans,
molluscs). Nevertheless, modelling the growth and reproductive potential (and failure) of species vulnerable to
those stressors with predictive tools, such as bioenergetic models is a useful approach for management and
protection purposes, but also for shellfish culture in general.

**Authors' contributions**
AG and GS conceived the idea and led the writing. AG carried out all experiments in mesocosms, performed
modelling work and analysed data. GS provided lab facilities and research funds. All authors contributed critically
to the drafts and gave final approval for publication.

**Acknowledgements**
PRIN TETRIS 2010 grant n. 2010PBMAXP_003, funded by the Italian Minister of Research and University
(MIUR) supported this research. The authors declare that they have no conflict of interest. We thank and are
especially grateful to all collaborators involved in this paper, in particular to Dr. Alessandro Rinaldi, Matteo
Mercurio and Marco Martinez for their technical support. We also thank Francesco Furnari for the use of the
scanning electron microscope. We deeply thanks Ms. Jan Underwood for the fine-tuning of the English.





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





**Tables**

Table 1. Seawater carbonate chemistry parameters ( mean ± se). Seawater pH on the NBS scale (pH$_{NBS}$),

temperature (T; °C), and salinity were used to calculate $CO_2$ partial pressure ($p$CO$_2$; µatm) as well as

aragonite and calcite saturation states (respectively Ωar and Ωca), for a total alkalinity of 2500 mmol kg$^{-1}$.

| | Measured | | | Calculated | | | |
|---|---|---|---|---|---|---|---|
| | pH$_{NBS}$ | O$_2$ mg/l | Salinity (PSU) | pCO$_2$ (µatm) | CO$_3^-$ | Ωca | Ωar |
| CTRL | 8.01±0.001 | 7.29±0.02 | 37.18±0.11 | 624.31±4.9 | 167.93±0.95 | 3.95±0.02 | 2.58±0.01 |
| Tr1 | 7.53±0.002 | 7.30±0.02 | 37.12±0.05 | 2151.17±22.02 | 62.05±0.73 | 1.46±0.02 | 0.95±0.01 |
| Tr2 | 8.01±0.001 | 2.44±0.02 | 37.07±0.04 | 729.88±18.24 | 152.53±1.51 | 3.59±0.04 | 2.34±0.02 |
| Tr3 | 7.53±0.002 | 2.44±0.02 | 37.21±0.17 | 2238.83±20.72 | 59.59±0.42 | 1.40±0.01 | 0.91±0.01 |

Table 2. ANOVA on seawater chemistry parameters. Comparison between CTRL (normal pH) and TREAT (low

pH and hypoxia) (* = p < 0.05; ** = p < 0.01; *** = p < 0.001; ns = not significant).

| | | pH$_{NBS}$ | | | pO$_2$ | | |
|---|---|---|---|---|---|---|---|
| | df | MS | F | p | MS | F | p |
| TREAT | 3 | 10.73 | 41450.84 | ** | 1083.21 | 18798.36 | ** |
| Residuals | 548 | 0.0003 | | | 0.06 | | |
| Cochran's Snk | | | | * | | | * |

| | | pCO$_2$ | | | CO$_3^-$ | | |
|---|---|---|---|---|---|---|---|
| | df | MS | F | p | MS | F | p |
| TREAT | 3 | 1.06e08 | 2426.84 | ** | 460157.7 | 3433.17 | ** |
| Residuals | 548 | 43851.09 | | | 134.03 | | |
| Cochran's Snk | | | | * | | | * |

| | | Ωca | | | Ωar | | |
|---|---|---|---|---|---|---|---|
| | df | MS | F | p | MS | F | p |
| TREAT | 3 | 254.09 | 3432.44 | ** | 108.26 | 3426.14 | ** |
| Residuals | 548 | 0.07 | | | 0.03 | | |
| Cochran's Snk | | | | * | | | * |



Table 3 ANOVA table of results. Effect on valve gape and breaking load of *Mytilus galloprovincialis* (* = p < 0.05; ** = p < 0.01; *** = p < 0.001; ns = not significant).

| Source | df | Valve gape | | | Source | df | Breaking load | | |
|---|---|---|---|---|---|---|---|---|---|
| | | MS | F | P | | | MS | F | P |
| Treatment (Tr) | 3 | 34.53 | 26.03 | *** | Treatment (Tr) | 3 | 3838.12 | 15.18 | *** |
| Residuals | 84 | 1.33 | | | Time (Ti) | 1 | 777.19 | 9.22 | ** |
| Cochran's C | | | | ns | Tr x Ti | 3 | 132.92 | 1.58 | Ns |
| | | | | | Residuals | 56 | | | |
| | | | | | Cochran's C | | | | ns |

Table 4 ANOVA table of results. Respiration rate (RR) and assimilation efficiency (AE) of *Mytilus galloprovincialis* (* = p < 0.05; ** = p < 0.01; *** = p < 0.001; ns = not significant).

| Source | df | RR st | | | AE | | |
|---|---|---|---|---|---|---|---|
| | | MS | F | P | MS | F | P |
| Treatment (Tr) | 3 | 312.9183 | 6.95 | *** | 0.2783 | 12.21 | *** |
| Time (Ti) | 1 | 205.1325 | 4.56 | * | 0.0424 | 1.86 | ns |
| Tr x Ti | 3 | 40.7752 | 0.91 | ns | 0.0198 | 0.87 | ns |
| Residuals | 120 | 45.0271 | | | 0.0228 | | |
| Cochran's C | | | | * | | | ns |



Table 5 DEB simulation outputs. Percentage variation of treatments from CTRL: Total length (TL), Total weight (WW), Total reproductive output (TRO), Total reproductive events (RE), Time to maturity (TM).

| DEB outputs (CTRL) after 4 years | | | | | | | |
|---|---|---|---|---|---|---|---|
| Site | Stressor | Frequency | TL (cm) | WW (g) | TRO (n° egg) | RE | TM (days) |
| Trieste | CTRL | 0 | 9.55 | 11.19 | 6737889 | 9 | 232 |
| Palermo | CTRL | 0 | 3.08 | 0.31 | 0 | 0 | 739 |

| DEB outputs: percentage variation respect to CTRL after 4 years | | | | | | | |
|---|---|---|---|---|---|---|---|
| Site | Stressor | Frequency | TL (%) | WW (%) | TRO (%) | RE (%) | TM (%) |
| Trieste | pH 7.5 | baseline | -13.44 | -35.20 | -53.49 | -22.22 | 18 |
| Trieste | pH+hypoxia | 1-month | -14.04 | -36.58 | -54.61 | -22.22 | 18 |
| Trieste | pH+hypoxia | 2-month | -14.71 | -38.04 | -56.34 | -22.22 | 20 |
| Trieste | pH+hypoxia | 3-month | -15.40 | -39.52 | -58.01 | -22.22 | 21 |
| Trieste | pH+hypoxia | 4-month | -16.09 | -40.97 | -59.52 | -22.22 | 22 |
| Trieste | pH+hypoxia | 5-month | -16.77 | -42.37 | -60.95 | -22.22 | 23 |
| Trieste | pH+hypoxia | 6-month | -17.38 | -43.62 | -62.26 | -22.22 | 24 |
| Palermo | pH 7.5 | baseline | -11.47 | -30.69 | 0 | 0 | 16 |
| Palermo | pH+hypoxia | 1-month | -12.09 | -32.26 | 0 | 0 | 17 |
| Palermo | pH+hypoxia | 2-month | -12.84 | -33.97 | 0 | 0 | 18 |
| Palermo | pH+hypoxia | 3-month | -13.42 | -35.26 | 0 | 0 | 20 |
| Palermo | pH+hypoxia | 4-month | -14.10 | -36.78 | 0 | 0 | 20 |
| Palermo | pH+hypoxia | 5-month | -14.70 | -38.08 | 0 | 0 | 22 |
| Palermo | pH+hypoxia | 6-month | -15.42 | -39.63 | 0 | 0 | 23 |

| Percentage additive effect of Hypoxia | | | | | | | |
|---|---|---|---|---|---|---|---|
| Trieste | pH+hypoxia | 1-month | -0.6 | -1.4 | -1.1 | 0 | 0.8 |
| Trieste | pH+hypoxia | 2-month | -1.3 | -2.8 | -2.8 | 0 | 2.1 |
| Trieste | pH+hypoxia | 3-month | -2 | -4.3 | -4.5 | 0 | 3.6 |
| Trieste | pH+hypoxia | 4-month | -2.7 | -5.8 | -6 | 0 | 4.4 |
| Trieste | pH+hypoxia | 5-month | -3.4 | -7.2 | -7.4 | 0 | 5.5 |
| Trieste | pH+hypoxia | 6-month | -4 | -8.4 | -8.8 | 0 | 6 |
| Palermo | pH+hypoxia | 1-month | -0.6 | -1.6 | 0 | 0 | 0.8 |
| Palermo | pH+hypoxia | 2-month | -1.3 | -3.3 | 0 | 0 | 2.1 |
| Palermo | pH+hypoxia | 3-month | -1.9 | -4.6 | 0 | 0 | 3.8 |
| Palermo | pH+hypoxia | 4-month | -2.6 | -6.1 | 0 | 0 | 4.5 |
| Palermo | pH+hypoxia | 5-month | -3.2 | -7.4 | 0 | 0 | 5.7 |
| Palermo | pH+hypoxia | 6-month | -3.9 | -8.9 | 0 | 0 | 7.6 |