# Peer review of "Functional spatial contextualisation of the effects of"

_Biogeosciences, 2018_

## Referee Comment (RC1) · Anonymous Referee #1 · 12 Feb 2018

This paper explores the implications of environmental stress (OA and hypoxia), as determined in lab experiments, on the growth and reproductive potential of mussels in two locations of the Mediterranean with simulations based on a dynamic energy budget model. The study capitalizes on the potential of DEB models to integrate the impacts of multiple environmental drivers on organismal level outcomes, including growth, reproduction, time to maturity, rates of feeding and respiration, and so on. This approach is powerful in potential and the application is new. However, there are some important shortcomings, especially in the way the model is parameterized. Also, I find the lack of some form of quality assessment problematic.

[Figure]

It is very annoying that all sections consist of a single paragraph. Did something go wrong with the formatting of the manuscript?

The author's use of respiration measurements as a proxy for DEB maintenance costs is problematic. In the DEB framework, respiration is emphatically not the same as maintenance, but also include energetic overheads, such that of growth. Respiration is a function of the commitment rate in DEB, of which maintenance could be a minor part, depending on size and nutritional status of the animal. In addition, oxygen deprived mussels, and possibly mussels enduring stress of hypercarnia, are able to use anaerobic metabolic pathways to fulfill their maintenance requirements. If stress increases maintenance requirements, one would expect respiration rates to increase with increasing stress intensity. However, we see the opposite happen (see Fig 2). I think this is likely due to the fact that stressed mussels have their shells closed more often than unstressed conspecifics (see Fig. 1), and thus ingest less food. Less food leads to a lower energy reserve buffer and therefore a lower rate at which reserves are committed. I suggest the authors change the maximum assimilation rate parameter of their model based on their behavioral observations and leave the maintenance rate parameter unchanged.

The simulations suggest that unstressed mussels only grow to 3 cm in length and do not reproduce in Palermo. This seems implausible. How long do real mussels get in Palermo? Do they reproduce? How sensitive are the simulation results to the particular choices of parameter values? The authors do not reflect at all on the reliability of their assessments, which I find troublesome, especially given the politicized context of the subject matter.

Title. Functional spatial contextualization sounds impressive but I've no clue what it could mean. Also, the manuscript deals with only a single species; the title is too general.

L27-33 Split up sentence.

L35 (and elsewhere) Put reference in the proper place of the sentence

L40 'lager'?

L68-70. This is a strong statement and should be substantiated with references. BTW, the only 2 papers using DEB in a OA context I'm aware of are 10.1111/gcb.12547 and 10.1016/j.jembe.2015.09.016

L72 the DEB [p_M] parameter does not relate to assimilation

L83 articulated → consisted of

Section 2.4 contains material that should go in 2.3 (or combine the sections).

I didn't get how the authors calculate the assimilation efficiency.

Section 3.1 belongs in the Materials and Methods Section. There is no need for a statistical analysis. Delete Table 2.

Combine Sections 3.2-4. There is no need to duplicate in the text what is already presented in the figures. The percentage of closed valves is simply 100 – percentage opened valves, so don't mention the former. I don't understand why the error measures differ so much, though.

Section 3.7 is incomprehensible for people without DEB modeling background. Include a figure and references to overview texts (e.g. Kooijman's book, Nisbet et al JAE, Sousa et al, and/or most recently Jusup et al Physics of Life Reviews 20:1-39).

L263 addictive → additive. The way the authors use 'additive' is confusing. Additive refers to impacts that can be summed, like 1+1=2, an unlikely situation with nonlinear models, such as DEB.

What are the initial conditions of the simulation runs?

What is the rational for the choices for the frequency of events?

From Table 5 remove data that are already presented in Figure 3. Round off # of eggs
to 6.74e6. Units of frequency should be 1/time

Figure 3 label y axis 'Change relative to control'

L294 delete 'formally'

L295 delete 'compensatory' and change contrast to compensate

L299 suppressed feeding activity

L304 what is crossing effect?

L306 on→ over. 'that' doesn't refer to anything

L333 sustainable and reliable → practical

L337 write out TW, TRO and TM

The readability of the manuscript would improve if there were fewer references. Remove unnecessary repetitive references.
* * *

---

## Referee Comment (RC2) · Anonymous Referee #2 · 12 Mar 2018

This study integrates laboratory-derived parameters of mussel metabolism and assimilation efficiency to run DEB models testing the effect of pH and hypoxia, using environmental data input (temperature, food) from two sites within the mussel's biogeographic range. I appreciate the approach of introducing hypoxia events (although I have a comment on the design these events) as a means of incorporating environmental variability in the model. This literature is sparse with such perspectives, especially in the context of multiple stressors. However, the paper lacks a perspective of the environmental relevance of the experimental design and modeling.

Major comments

- Methods need much more detail (see detailed comments)

- Physiological condition of experimental mussels during the experiment is not quantified. Feeding was ad libitum and it was not assessed if mussels were being fed at conditions of either site used for the modeling. This is problematic as, for example, if the mussels are starving relative to their natural food supply, the derived experimental parameters for the DEB model may be inappropriate. The authors also do not explain the experimental design. Why was the experiment 4 weeks?

- DEB models: As a reader of Biogeosciences, but not an expert on DEB models, it would be helpful if the authors reviewed this approach more clearly (perhaps using a schematic, what program is used to run models, table of input variables etc.). After reading the paper, I am unclear about the exact implementation and conclusions that can be drawn from this simulation based on the following 4 sources of confusion:

1) From what I understand, temperature from 2006-2009 is used as one of the model inputs. However, all the biological parameters taken from the experiments come from 21 degr. C (although this is not stated explicitly for respiration, but I assume it's 21 C). Since environmental data would vary in terms of temperature across the four years, I don't understand how biological performance is scaled across this temperature regime. It would be good to include a figure of the environmental data (means are not great time-series descriptors, especially for biological processes that are seasonal, such as reproduction), as well as a figure on how the biological parameters were scaled for temperature effects over the years. This same argument applies to food concentration (which I assume varies by time of year as well).

2) The authors use hypoxia and acidification, two future stressors, with temperature data from a few years ago. This design ignores the fact that warming is currently the dominant stressors for this species in the Mediterranean Sea and is expected to continue in the future. As the environmental relevance of the study design is not discussed, as written, the results do not match any realistic environmental situation. This counters

the original intent of using DEB models to better "predict organismal functional traits, capturing variation across species to solve a very wide range of problems in ecology and evolutionary biology" (L58).

3) Given that reproduction of this species can be quite seasonal, how does the DEB model handle this in terms of estimating reproductive output?

4) L188-189: Why is the hypoxia event randomized by month of the year? Hypoxia would most likely occur during summer warming and stratification. It seems that varying the duration of summertime hypoxia is a more environmentally relevant exercise rather than randomizing what month the hypoxia event occurred in. How long was each hypoxia event?

- Statistical approach needs to be justified (see detailed comments)

- The choice of using 2500 umol/kg for total alkalinity (TA) based on an oceanographic study for the lab experiments is strange (L106). Especially in static cultures, mussels can alter the TA of a small body of water. I assume the authors did not measure TA during the experiment. In such a case, it would be best to simulate the experiment again, and measure TA so the authors have some idea of the TA variation in their experimental conditions could have been. Either way, the calculated pCO2 parameters will be undefinable without the real TA measurement.

- In addition to lacking an environmental context, the Discussion lacks comments on the non-DEB model functional traits (shell strength, dissolution patterns), their relevance to the study, and by what mechanism hypoxia and pH would differentially or synergistically impact the periostracum and shell quality.

Detailed comments:

The title does not represent the study and reads as if the paper is a literature review. It would behoove the authors include more detail in the title (DEB model, hypoxia, OA). Use of "marine bivalves" is inappropriate given that only one species was assessed.

L27-33: sentence is difficult to follow. Consider breaking this up.

L39: this needs clarification (most functional traits? Which ones?) and references

L47-48: plenty of labs conduct OA experiments for months up to at least one year

L48-54: long sentence, CO2 vents are unrelated to the second half of the sentence. Consider rewriting.

L69: should AE be 'assimilation efficiency'?

Introduction: break text up into paragraphs. Lacks introduction to functional trait-based models; I was expecting this prior to L54.

L92: Mussels were fed ad libitum, but this is an energetics study. So how do the authors know the condition of the mussels used to get model parameters?

L104: dissolution threshold relates to calcium carbonate saturation state, please include the value here, rather than pH.

L109: how was CO2 dissolved? Where their pumps in all the aquaria to ensure mixing?

Section 2.1: Sampling of what (section title)? How often was water replaced in the treatment tanks? Methods need a better description of how carbonate chemistry was calculated. What was the accuracy of the pH measurements? How often was the water sampled for each of the four parameters? How was oxygen maintained and measured in the treatments?

Section 2.2 (L112-120): if there are 25 mussels per tank, and there are 3 tanks, why are only five mussels observed for valve opening and closure? Why were observations made 6 times per day every week rather than fewer times per day but more frequently throughout the experiment? Does time of day matter for this behavior? What about time that food was added? I image that flow rates could affect this behavior, but it's unclear if water motion was the same across all tanks.

L124: define [pm]

L134: this equation results in units of O2 concentration x volume per unit time, oxygen units are not defined and there is no explanation as to how this is converted to [pm], which is in J per cubic cm per hour. What level of oxygen undersaturation was reached by the end of the incubation?

L140: this assumption should be justified

Section 2.3: explain that the same individual was used for the respiration rate followed by AE. It's unclear until the end of section 2.4. Given that the respiration methods continue in the end of Section 2.4, merge Section 2.3 and 2.4.

L141-145: Please explain why AE experiment was not done in treatment water, and justify how AE can then be related to experimental treatments.

L162: Again, if food availability is important at the field sites, food availability during the experiment should be known. Is it closer to that of Trieste or Palermo?

Section 2.7: How are simulations performed (what code or computer program?)?

L180: State what DEB parameters these are.

L181: Is AE the same as [pM]? AE was already defined in the Introduction

L185-186: Was this data from week 1 or 4? How is a 4-week acclimation period determined sufficient enough to extrapolate to 4 years?

Section 2.9: The assumption of normally distributed residuals is not tested for the ANOVA. This needs to be done before moving forward with ANOVAs. A sample size of 16 is not large. The statistical analyses for valve closure does not match the data collection. By using ANOVAs, I assume all the data are pooled across the 4 weeks. This is not appropriate because it does not account for acclimation and it is a repeated measure since there are only 25 mussels in each tank which were observed over 4 weeks. ANOVAs also don't control for the tank replicate per treatment. The authors

need to clarify how the data was pooled (and which behavior was analyzed – open or closed). Since this is binary data, reporting both in the bar graph is duplication the data (Section 3.2), report one, or as a stacked bar graph where each bar graph fills 100%.

Section 3.1: Analysis comparing experimental treatments seems unnecessary, especially given the uncertainty of the calculated parameters using a poor assumption of TA.

L350-352: is this to be expected?

Figures: What is the error bar?

L259: capital I

L261: replace M&M with Section #

Table 1: Include temperature

Table 5: include input parameters

---

## Author Comment (AC1) · 2 Apr 2018

REVIEWER #1 Reviewer wrote: This paper explores the implications of environmental stress (OA and hypoxia), as determined in lab experiments, on the growth and reproductive potential of mussels in two locations of the Mediterranean with simulations based on a dynamic energy budget model. The study capitalizes on the potential of DEB models to integrate the impacts of multiple environmental drivers on organismal level outcomes, including growth, reproduction, time to maturity, rates of feeding and respiration, and so on. This approach is powerful in potential and the application is new. However, there are some important shortcomings, especially in the way the model is

parameterized. Also, I find the lack of some form of quality assessment problematic. Author's reply: First, we thank the Reviewer #1 for the effort in providing his/her suggestions to the original version of our ms. Author's changes: Apart from all possible not clear parts that we accordingly improved, we followed his/her suggestion in the hope to have increased the quality of this ms.

Reviewer wrote: It is very annoying that all sections consist of a single paragraph. Did something go wrong with the formatting of the manuscript? Author's reply: We appreciated the referee's suggestion about the different paragraphs, but we believe that the current structure is already sufficiently sectioned; to increase the number of paragraphs or sub-paragraphs can only increase the text fragmentation which may limit the logical flow of the text. Reviewer wrote: The author's use of respiration measurements as a proxy for DEB maintenance costs is problematic. In the DEB framework, respiration is emphatically not the same as maintenance, but also include energetic overheads, such that of growth. Respiration is a function of the commitment rate in DEB, of which maintenance could be a minor part, depending on size and nutritional status of the animal. Author's reply: We acknowledge that respiration does not include only maintenance, but also include energetic overheads, such as growth. Nevertheless, there is no way - to our knowledge - to measure the different contribution of every energetic components apart from to experimentally measure oxygen consumption as a proxy for metabolism. Also, the proposed approaches measuring indirectly the [áźŮM] values (e.g. van der Veer et al., 2006; Ren and Schiel 2008), are not feasible in the context of the present experimental asset. While this approach is not experimentally feasible when assessing the effect of stressors on the energy budget, the only way to indirectly provide an estimation of the effect of disturbance is through the Jagger et al. (2016) approach which is based on the stress factor "s". Thus, after estimating the effect induced by a treatment on the oxygen consumption, that in the present case study was expressed as a percentage variation, we summed/subtracted the energetic amount due to the effect of a stressor to the species-specific [áźŮM] values of M. galloprovincialis then we run our models. However, we thank the referee for highlighting this point whose importance was addressed in the Discussion section of this ms. as we believe that is crucial to increase our understating on how we can mechanistically assess the effect of disturbance on individual performances through the DEB model. All these limitations show how much is important to date to increase the experimental and theoretical research effort in order to unravel this point, which is increasingly crucial to get realistic answers to management questions in a context of environmental change.

Reviewer wrote: In addition, oxygen deprived mussels, and possibly mussels enduring stress of hypercapnia, are able to use anaerobic metabolic pathways to fulfil their maintenance requirements. If stress increases maintenance requirements, one would expect respiration rates to increase with increasing stress intensity. However, we see the opposite happen (see Fig 2). I think this is likely due to the fact that stressed mussels have their shells closed more often than unstressed conspecifics (see Fig. 1), and thus ingest less food. Less food leads to a lower energy reserve buffer and therefore a lower rate at which reserves are committed. Author's reply: As reported in our results and Fig. 2, maintenance requirements in accordance with respiration rates, decreases with stress in agreement with what is reported in the current literature. We are sorry with Reviewer #1 and with all readers as we made a mistake in writing the text commenting the figure 1 (we wrote wrongly "opened" instead of "closed"). Actually, our mussels increased their openness with the increasing stressful conditions. At the present stage, we are not able to provide information about the amount of ingested food under different treatments and then we are not able to infer on the effect of openness degree on energetic performances. Author's changes: Figure was fixed according to both referee's suggestion, and also the text in the paragraph has been rephrased accordingly.

Reviewer wrote: I suggest the authors change the maximum assimilation rate parameter of their model based on their behavioural observations and leave the maintenance rate parameter unchanged. Author's reply: We appreciated the referee's suggestion but we prefer to focus on both the assimilation efficiency and the metabolic effect (through pM) as i) the main effect of acidification seem to be exerted on metabolism as widely reported in the current literature, and ii) also to show that our mechanistic DEB approach can be really effective in measuring the multiple stressor's effect on LHs. Author's changes: Thus, we enlarged the discussion on these points to include possible shortcomings deriving from the fact that the stressor's effect on maintenance is not still well-experimentally measurable.

Reviewer wrote: The simulations suggest that unstressed mussels only grow to 3 cm in length and do not reproduce in Palermo. This seems implausible. How long do real mussels get in Palermo? Do they reproduce? How sensitive are the simulation results to the particular choices of parameter values? The authors do not reflect at all on the reliability of their assessments, which I find troublesome, especially given the politicized context of the subject matter. Author's reply: Actually, to enlarge the discussion about the magnitude of effects at local level could be not influent for our purposes, although our results are in line with the environmental and trophic conditions reported in section 2.5: "Both sites were chosen as they represent two opposite temperature and food conditions for mussel growth in Italy... etc.". M. galloprovincialis in Sicily is observed to be limited by oligo-trophic conditions although it grows in highly trophic-enriched areas such as harbours or under Integrated Multi-Trophic Aquaculture (IMTA) conditions (Sarà et al 2012; 2013b, Giacoletti et al. 2018 in press JEMA) which supports what we gathered in the present ms. through the DEB simulations.

Reviewer specific comment n. 1 Reviewer wrote: Title. Functional spatial contextualization sounds impressive but I've no clue what it could mean. Also, the manuscript deals with only a single species; the title is too general. Author's changes: We agreed with Reviewer's #1 point and changed the title.

Reviewer specific comment n. 2 Reviewer wrote: L27-33 Split up sentence. Author's changes: Sentence was splitted up accordingly.

Reviewer specific comment n. 3 Reviewer wrote: L35 (and elsewhere) Put reference in the proper place of the sentence Author's changes: All references were checked and put in proper spaces.

Reviewer specific comment n. 4 Reviewer wrote: L40 'lager'? Author's changes: Changed with "larger".

Reviewer specific comment n. 5 Reviewer wrote: L68-70. This is a strong statement and should be substantiated with references. BTW, the only 2 papers using DEB in a OA context I'm aware of are 10.1111/gcb.12547 and 10.1016/j.jembe.2015.09.016 Author's changes: References regarding the effect of OA on functional traits such as feeding and assimilation, and on maintenance costs has been added accordingly.

Reviewer specific comment n. 6 Reviewer wrote: L72 the DEB [p_M] parameter does not relate to assimilation Author's changes: The sentence was rephrased accordingly.

Reviewer specific comment n. 7 Reviewer wrote: L83 articulated ! consisted of Author's changes: Changed accordingly.

Reviewer specific comment n. 8 Reviewer wrote: Section 2.4 contains material that should go in 2.3 (or combine the sections). Author's reply: Section 2.4 refers to assimilation efficiency, while section 2.3 to oxygen consumption measures, so we consider not easy to combine both sections as we may incur in the risk to reduce the readability of this section.

Reviewer specific comment n. 9 Reviewer wrote: I didn't get how the authors calculate the assimilation efficiency. Author's changes: A detailed explanation on how the assimilation efficiency was estimated, was added with supporting references.

Reviewer specific comment n. 10 Reviewer wrote: Section 3.1 belongs in the Materials and Methods Section. There is no need for a statistical analysis. Delete Table 2. Author's changes: Table 2 was deleted according to both referee's suggestion and details were moved in the Materials and Methods Section.

Reviewer specific comment n. 11 Reviewer wrote: Combine Sections 3.2-4. There is no need to duplicate in the text what is already presented in the figures. The percentage of closed valves is simply 100 – percentage opened valves, so don't mention the former. I don't understand why the error measures differ so much, though. Author's changes: We agree not to duplicate in text what is already presented in figures, and we worked to avoid this replication. Following Reviewer's #1 suggestion we also expressed the percentage of closed valves 100 – percentage opened valve. Instead, merging the sections can increase the risk of confusion in the reader as section 3.2 is about behavioural observations, while the other two are about physiological measurements.

Reviewer specific comment n. 12 Reviewer wrote: Section 2.7 is incomprehensible for people without DEB modeling background. Include a figure and references to overview texts (e.g. Kooijman's book, Nisbet et al JAE, Sousa et al, and/or most recently Jusup et al Physics of Life Reviews 20:1-39). Author's changes: We agree with Reviewer #1 that section 2.7 is difficult for someone without a DEB modelling background, and in order to made it more clear we added the suggested references and rephrased some parts.

Reviewer specific comment n. 13 Reviewer wrote: L263 addictive ! additive. The way the authors use 'additive' is confusing. Additive refers to impacts that can be summed, like 1+1=2, an unlikely situation with nonlinear models, such as DEB. Author's changes: Rephrased following suggestions to "with a progressive contribution of hypoxia".

Reviewer specific comment n. 14 Reviewer wrote: What are the initial conditions of the simulation runs? Author's reply: Results of simulation performed with unstressed organism were already reported in Table 5, while model parameters have been reported in Table 1.

Reviewer specific comment n. 15 Reviewer wrote: What is the rational for the choices for the frequency of events? Author's reply: While we are aware that hypoxia events are more frequently during summertime, we decided to not apply any timing and frequency scheme to simulate hypoxia event's occurrence according to many papers published across the recent literature(Crain et al. 2008 Ecology Letters; Miller et al. 2009 PNAS).

Reviewer specific comment n. 16 Reviewer wrote: From Table 5 remove data that are already presented in Figure 3. Round off # of eggs to 6.74e6. Units of frequency should be 1/time Author's changes: Table 5 was corrected accordingly.

Reviewer specific comment n. 17 Reviewer wrote: Figure 3 label y axis 'Change relative to control' Author's changes: Figure was corrected accordingly.

Reviewer specific comment n. 18 Reviewer wrote: L294 delete 'formally' Author's changes: Deleted.

Reviewer specific comment n. 19 Reviewer wrote: L295 delete 'compensatory' and change contrast to compensate Author's changes: Rephrased.

Reviewer specific comment n. 20 Reviewer wrote: L299 suppressed feeding activity Author's changes: Changed.

Reviewer specific comment n. 21 Reviewer wrote: L304 what is crossing effect? Author's changes: Sentence has been rephrased accordingly.

Reviewer specific comment n. 22 Reviewer wrote: L306 on! over. 'that' doesn't refer to anything Author's changes: Rephrased.

Reviewer specific comment n. 23 Reviewer wrote: L333 sustainable and reliable ! practical Author's changes: Changed.

Reviewer specific comment n. 24 Reviewer wrote: L337 write out TW, TRO and TM Author's changes: Written out accordingly.

Reviewer specific comment n. 25 Reviewer wrote: The readability of the manuscript would improve if there were fewer references. Remove unnecessary repetitive references. Author's changes:: All the references has been checked and unnecessary and repetitive ones have been deleted accordingly.

[Figure]

Please also note the supplement to this comment:
https://www.biogeosciences-discuss.net/bg-2018-13/bg-2018-13-AC1-supplement.pdf

**Supplement:**

**REVIEWER #2**

**Reviewer wrote:** This study integrates laboratory-derived parameters of mussel metabolism and assimilation efficiency to run DEB models testing the effect of pH and hypoxia, using environ- mental data input (temperature, food) from two sites within the mussel's biogeographic range. I appreciate the approach of introducing hypoxia events (although I have a comment on the design these events) as a means of incorporating environmental variability in the model. This literature is sparse with such perspectives, especially in the context of multiple stressors. However, the paper lacks a perspective of the environmental relevance of the experimental design and modelling.

**Author's reply:** We thank the Reviewer #2 for helping us improving the readability and the clearness of the ms.

**Author's changes:** In doing so we applied most of the suggestion for the highlighted points, and we discussed them through the specific comments. A perspective of environmental relevance on modelling outputs has been currently added in the discussion section following what suggested by the reviewer.

**Major comments**

Methods need much more detail (see detailed comments)

**Reviewer wrote:** Physiological condition of experimental mussels during the experiment is not quantified. Feeding was ad libitum and it was not assessed if mussels were being fed at conditions of either site used for the modeling. This is problematic as, for example, if the mussels are starving relative to their natural food supply, the derived experimental parameters for the DEB model may be inappropriate. The authors also do not explain the experimental design. Why was the experiment 4 weeks?

**Author's reply:** We thank the reviewer for highlighting this point.

**Author's changes:** We estimated the condition index through the biometric available data and compared it through the experiment, resulting in a not significant variation throughout the study period, and this supports that experimental animals were not stressed by starvation. Further, we used the locution "ad libitum" to indicate a food concentration saturating the feeding processes of animals over time. Such an experimental maintenance condition is commonly used throughout the current literature when bivalves are maintained with ad libitum condition of food in bioenergetic experiments (e.g. Sarà et al., 2013; Montalto et al., 2014; Tagliarolo et al., 2016). However, we adopted a four weeks-period to estimate the effect of OA on functional traits of mussels; such a period is judged to be enough to allow mussels to acclimate to new conditions, as showed in many experimental papers across the current literature(references added in the manuscript).

**Reviewer wrote:** DEB models: As a reader of Biogeosciences, but not an expert on DEB models, it would be helpful if the authors reviewed this approach more clearly (perhaps using a schematic, what program is used to run models, table of input variables etc.). After reading the paper, I am unclear about the exact implementation and conclusions that can be drawn from this simulation based on the following 4 sources of confusion:

1)From what I understand, temperature from 2006-2009 is used as one of the model inputs. However, all the biological parameters taken from the experiments come from 21 degr. C (although this is not stated explicitly for respiration, but I assume it's 21 C). Since environmental data would vary in terms of temperature across the four years, I don't understand how biological performance is scaled across this temperature regime. It would be good to include a figure of the environmental data (means are not great time-series descriptors, especially for biological processes that are seasonal, such as reproduction), as well as a figure on how the biological parameters were scaled for temperature effects over the years. This same argument applies to food concentration (which I assume varies by time of year as well).

**Author's reply:** The 2006-2009 thermal series has been used as a forcing variables inside DEB models in current modelling literature (Sarà et al. 2011, 2013b; Montalto et al. 2014). The most important factors driving changes in energy budgets of ectotherms are body temperature, on which every metabolic rates depends via the Arrhenius relationships (Kooijman 2010), and food. Arrhenius temperature, that is species specific, acts as a correction factors inside DEB models to scale all rates to environmental temperature. At the same time DEB models use the 2006-2009 CHL-a series as a second forcing variables to predict LH-traits of our species. Accordingly, including a new figure of environmental data could make the ms. heavier also as the main object is not to contextualise the effect at that period, but to show that stressor's effect is simulated across a long integrated period.

**Reviewer wrote:** 2) The authors use hypoxia and acidification, two future stressors, with temperature data from a few years ago. This design ignores the fact that warming is currently the dominant stressors for this species in the Mediterranean Sea and is expected to continue in the future. As the environmental relevance of the study design is not discussed, as written, the results do not match any realistic environmental situation. This counters the original intent of using DEB models to better "predict organismal functional traits, capturing variation across species to solve a very wide range of problems in ecology and evolutionary biology" (L58).

**Author's reply:** We did not test the effect of increasing temperature as we are pretty sure that the thermal effect is not manifested on a period so short (only 4-years); however other companion papers (e.g. Montalto et al. 2017) tested the effect of increasing temperature on mussel's performances throughout the whole Basin. To extrapolate the effect only two stressors, we carried out simulations under two different latitudinal temperature patterns (Trieste, north Adriatic and Palermo, Southern MED). Anyway, we included some discussion lines about this issue.

**Reviewer wrote:** 3) Given that reproduction of this species can be quite seasonal, how does the DEB model handle this in terms of estimating reproductive output?

**Author's reply:** DEB manages the seasonal reproduction throughout thresholds based on the temperature-energy relationships.

**Reviewer wrote:** 4) L188-189: Why is the hypoxia event randomized by month of the year? Hypoxia would most likely occur during summer warming and stratification. It seems that varying the duration of summertime hypoxia is a more environmentally relevant exercise rather than randomizing what month the hypoxia event occurred in. How long was each hypoxia event?

**Author's reply:** As we said before, we tested the effect of hypoxia as a stochastic event more than testing is in terms of frequency and timing. Thus, here we adopted a very simple scheme but there is a companion paper still under review (G. Sarà submitted PRS B) whose main aim was to test the effect of duration, frequency and timing of disturbance events on three different invertebrate species through the DEB model.

**Reviewer wrote:** Statistical approach needs to be justified (see detailed comments)
**Author's reply:** Statistical approach has been justified in the detailed comments.

**Reviewer wrote:** The choice of using 2500 umol/kg for total alkalinity (TA) based on an oceanographic study for the lab experiments is strange (L106). Especially in static cultures, mussels can alter the TA of a small body of water. I assume the authors did not measure TA

during the experiment. In such a case, it would be best to simulate the experiment again, and measure TA so the authors have some idea of the TA variation in their experimental conditions could have been. Either way, the calculated pCO2 parameters will be undefinable without the real TA measurement.

**Author's reply:** We did not find any paper reporting such alteration. We tried at the same time to minimize the number of organism for each tank and to perform a sufficient weekly water change in order to maintain a stable environment for our organisms, even if in mesocosm condition. We are perfectly aware that the suggested one is without any reasonable doubt the most appropriate approach to follow, but as soon as our is not a study focused on the chemistry of the shell but it is to provide a proof to test the predictive potential of DEB model about the effect of two stressors; thus, the use of a value from oceanographic study should be considered a minor approximation due to the metabolic and mechanistic nature of the paper.

**Reviewer wrote:** In addition to lacking an environmental context, the Discussion lacks comments on the non-DEB model functional traits (shell strength, dissolution patterns), their relevance to the study, and by what mechanism hypoxia and pH would differentially or synergistically impact the periostracum and shell quality.

**Author's reply:** we didn't analyse the impact on the shell chemistry and ultrastructure in this ms. whose main objective was to predict the effect of two stressors on mussel's LHs.

**Author's changes:** However, to accomplish the referee's suggestion, we discussed shortly shell fragility related to pH and to both combined stressor.

**Detailed comments:**
**Reviewer #2 specific comment n. 1**
**Reviewer wrote:** The title does not represent the study and reads as if the paper is a literature review. It would behove the authors include more detail in the title (DEB model, hypoxia, OA). Use of "marine bivalves" is inappropriate given that only one species was assessed.
**Author's changes:** We agreed with Reviewer's #2 point and changed the title.

**Reviewer #2 specific comment n. 2**
**Reviewer wrote:** L27-33: sentence is difficult to follow. Consider breaking this up.
**Author's changes:** Sentence has been spitted up accordingly.

**Reviewer #2 specific comment n. 3**
**Reviewer wrote:** L39: this needs clarification (most functional traits? Which ones?) and references
**Author's changes:** Although references were already present, following Reviewer #2 suggestion we specified the functional traits which we were referring to.

**Reviewer #2 specific comment n. 4**
**Reviewer wrote:** L47-48: plenty of labs conduct OA experiments for months up to at least one year
**Author's changes:** The sentence has been deleted.

**Reviewer #2 specific comment n. 5**
**Reviewer wrote:** L48-54: long sentence, CO2 vents are unrelated to the second half of the sentence. Consider rewriting.
**Author's changes:** Sentence has been rephrased accordingly.

**Reviewer #2 specific comment n. 6**
**Reviewer wrote:** L69: should AE be 'assimilation efficiency'?
**Author's changes:** Sentence has been rephrased and clarified.

**Reviewer #2 specific comment n. 7**
**Reviewer wrote:** Introduction: break text up into paragraphs. Lacks introduction to functional trait-based models; I was expecting this prior to L54.
**Author's changes:** Introduction was spitted into paragraphs following suggestion and an introduction to functional trait-based models was added.

**Reviewer #2 specific comment n. 8**
**Reviewer wrote:** L92: Mussels were fed ad libitum, but this is an energetics study. So how do the authors know the condition of the mussels used to get model parameters?
**Author's reply:** As in reply to a similar point raised by Rev#1 we used the locution "ad libitum" to indicate a food concentration saturating the feeding processes of animals over time. Such an experimental maintenance condition is commonly used throughout the current literature when bivalves are maintained with ad libitum condition of food in bioenergetic experiments (e.g. Sarà et al., 2013; Montalto et al., 2014; Tagliarolo et al., 2016).

**Reviewer #2 specific comment n. 9**
**Reviewer wrote:** L104: dissolution threshold relates to calcium carbonate saturation state, please include the value here, rather than pH.
**Author's reply:** Unfortunately we do not have such details on calcium carbonate saturation state. We added image details showing the effect of OA on the external shell layer, but a deep analysis on the chemical composition and alteration was out of the purpose of the present investigation.

**Reviewer #2 specific comment n. 10**
**Reviewer wrote:** L109: how was CO2 dissolved? Where their pumps in all the aquaria to ensure mixing?
**Author's changes:** Details on how $CO_2$ was dissolved and of water recirculation were added.

**Reviewer #2 specific comment n. 11**
Section 2.1: Sampling of what (section title)? How often was water replaced in the treatment tanks? Methods need a better description of how carbonate chemistry was calculated. What was the accuracy of the pH measurements? How often was the water sampled for each of the four parameters? How was oxygen maintained and measured in the treatments?
**Author's changes:** Section title has been changed accordingly. Details on water changes, aeration, water recirculation and accuracy of pH measurements has been added, while details on the calculation of water chemistry were already present in section 2.1. Further details on sampling of water (8 time a day for pH), and on frequency of oxygen and temperature measurement has been added in the same section.

**Reviewer #2 specific comment n. 12**
**Reviewer wrote:** Section 2.2 (L112-120): if there are 25 mussels per tank, and there are 3 tanks, why are only five mussels observed for valve opening and closure? Why were observations made 6 times per day every week rather than fewer times per day but more frequently throughout the experiment? Does time of day matter for this behavior? What about time that food was added? I image that flow rates could affect this behavior, but it's unclear if water motion was the same across all tanks.

**Author's reply:** We thank the referee for his/her interest on the behavioural part of our paper. The restricted number of observation was chosen in order to make the behavioural session during as less as possible in order to nor influence with the operator presence mussel's behaviour. Even if it has not proven sometimes mussels suddenly close when moving in front of the tanks. For this reason we decided to observe less individual but more frequently. We didn't notice any difference in the behaviour during the day as mussels were inside a temperature-controlled room, under constant flow through conditions (according to Widdows and Staff 2006 and Sarà et al. 2013) and exposed to automated artificial daylight. The food was added at the end of the day, after all observations were made.

**Reviewer #2 specific comment n. 13**
**Reviewer wrote:** L124: define [pm]
**Author's changes:** [ṗM] has been here defined following suggestion.

**Reviewer #2 specific comment n. 14**
**Reviewer wrote:** L134: this equation results in units of O2 concentration x volume per unit time, oxygen units are not defined and there is no explanation as to how this is converted to [pm], which is in J per cubic cm per hour. What level of oxygen undersaturation was reached by the end of the incubation?
**Author's reply:** Oxygen concentration ($\mu$mol $l^{-1}$ $h^{-1}$) were first converted in J $h^{-1}$ and then in J $cm^{-3}$ $h^{-1}$ using a conversion factor (Kooijman, 2010) and following the current literature (Van der Veer et al., 2006; Ren and Schiel 2008). We never reached lethal oxygen concentration due to the short interval of the measurement as the idea was to simulate a sub-lethal effect as that reach at about 1.5-2.0 mg-l DO.

**Reviewer #2 specific comment n. 15**
**Reviewer wrote:** L140: this assumption should be justified
**Author's changes:** As soon as the first two sentences of the section has the same reference, we moved it at the end of the second sentence.

**Reviewer #2 specific comment n. 16**
**Reviewer wrote:** Section 2.3: explain that the same individual was used for the respiration rate followed by AE. It's unclear until the end of section 2.4. Given that the respiration methods continue in the end of Section 2.4, merge Section 2.3 and 2.4.
**Author's reply:** As answered to Reviewer #1 we do not agree in merging both sections because they represent two different part of metabolic stuff (feeding and respiration). However, we now clear specified that specimens were the same between both measure following Reviewer's #2 suggestion.

**Reviewer #2 specific comment n. 17**
**Reviewer wrote:** L141-145: Please explain why AE experiment was not done in treatment water, and justify how AE can then be related to experimental treatments.
**Author's changes:** We now clearly specified that the experiment was conducted with water specifically treated for each treatment.

**Reviewer #2 specific comment n. 18**
**Reviewer wrote:** L162: Again, if food availability is important at the field sites, food availability during the experiment should be known. Is it closer to that of Trieste or Palermo?
**Author's reply:** As in reply to a similar point raised by Rev#1 we used the locution "ad libitum" to indicate a food concentration saturating the feeding processes of animals over time. Such an experimental maintenance condition is commonly used throughout the current literature when bivalves are maintained with ad libitum condition of food in bioenergetic experiments (e.g. Sarà et al., 2013; Montalto et al., 2014; Tagliarolo et al., 2016).

**Reviewer #2 specific comment n. 19**
**Reviewer wrote:** Section 2.7: How are simulations performed (what code or computer program?)?
**Author's reply:** Our simulation were performed using R routine, and we specified it in the m accordingly.

**Reviewer #2 specific comment n. 20**
**Reviewer wrote:** L180: State what DEB parameters these are.
**Author's changes::** DEB parameters are now reported in Table 1.

**Reviewer #2 specific comment n. 21**
**Reviewer wrote:** L181: Is AE the same as [pM]? AE was already defined in the Introduction
**Author's changes:** According to Reviewer's #1 point we checked and fixed both assimilation efficiency and the somatic maintenance costs definition.

**Reviewer #2 specific comment n. 22**
**Reviewer wrote:** L185-186: Was this data from week 1 or 4? How is a 4-week acclimation period determined sufficient enough to extrapolate to 4 years?
**Author's reply:** The $[\dot{p}_M]$ parameter used inside our simulations was that calculated at week #4. All the species specific parameters derived from one species and freely available online on the Add my pet collection were previously determined either by the covariation method through data present on literature or experimentally by short experimental sessions. Even without considering the effect of a stressor, a parameter estimated for a well-fed organism is then used inside simulation making predictions up to 4, 10 or even 50 years, as parameters are assumed to be specific for each species (Kooijman, 2010). We did not account any possible evolutionary effect whose effect is still far to be assessed in the DEB theory.

**Reviewer #2 specific comment n. 23**
**Reviewer wrote:** Section 2.9: The assumption of normally distributed residuals is not tested for the ANOVA. This needs to be done before moving forward with ANOVAs. A sample size of 16 is not large. The statistical analyses for valve closure does not match the data collection. By using ANOVAs, I assume all the data are pooled across the 4 weeks. This is not appropriate because it does not account for acclimation and it is a repeated measure since there are only 25 mussels in each tank which were observed over 4 weeks. ANOVAs also don't control for the tank replicate per treatment. The authors need to clarify how the data was pooled (and which behavior was analyzed – open or closed). Since this is binary data, reporting both in the bar graph is duplication the data (Section 3.2), report one, or as a stacked bar graph where each bar graph fills 100%.
**Author's reply:** The assumption of normal distribution has been tested through the Anderson–Darling test using Past® software. We are aware that the sample size is not large, but pooling the six observation per week we obtain a sample of 24, and we believe it is sufficient for the purpose of the present paper.
**Author's changes:** We repeated the data analysis and accordingly to the suggestion we compared week1 and week4 using two levels of the factor time and 4 levels of the factor treatment. We analysed the open valve behaviour and following suggestion we decided to use only one category in the graph. The paragraph, the table and the figure has been modified accordingly.

**Reviewer #2 specific comment n. 24**
**Reviewer wrote:** Section 3.1: Analysis comparing experimental treatments seems unnecessary, especially given the uncertainty of the calculated parameters using a poor assumption of TA.
**Author's changes:** Analysis comparing experimental treatments were removed accordingly to both referee's suggestions.

**Reviewer #2 specific comment n. 25**
**Reviewer wrote:** L350-352: is this to be expected?
**Author's reply:** We thank the referee for the question and we have already answered to this point being highlighted by Referee#1. *M. galloprovincialis* in Sicily is observed to be limited by oligo-trophic conditions although it grows in highly trophic-enriched areas such as harbours or under Integrated Multi-Trophic Aquaculture (IMTA) conditions (Sarà et al 2012; 2013b, Giacoletti et al. 2018 in press JEMA) which supports what we gathered in the present ms. through the DEB simulations.

**Reviewer #2 specific comment n. 26**
**Reviewer wrote:** Figures: What is the error bar?
**Author's reply:** The error bar indicated standard errors for means.
**Author's changes:** Details were added in each figure following Reviewer's suggestions.

**Reviewer #2 specific comment n. 27**
**Reviewer wrote:** L259: capital I
**Author's changes::** Replaced.

**Reviewer #2 specific comment n. 28**
**Reviewer wrote:** L261: replace M&M with Section #
**Author's changes::** Replaced accordingly.

**Reviewer #2 specific comment n. 29**
**Reviewer wrote:** Table 1: Include temperature
**Author's changes:** Temperature included inside Table 1.

**Reviewer #2 specific comment n. 30**
**Reviewer wrote:** Table 5: include input parameters
**Author's changes:** DEB parameters were included in Table 1.

**Predicting the multiple effects of acidification and hypoxia on *Mytilus galloprovincialis* (Bivalvia, Mollusca) life history traits**
* * *
Antonio Giacoletti* and Gianluca Sarà

[revised manuscript text omitted]

Recent insights obtained by the experimental research have shown that OA mainly affects feeding rates (FR), assimilation efficiency (AE) and maintenance cost  of marine organisms (Appelhans, 2012; Navarro et al. 2013; Kroeker et al. 2014; Zhang et al., 2015; Jager et al. 2016). Here, we translated the combined effects of hypoxia and hypercapnia on  assimilation and oxygen consumption rates as measured under different treatments into effects on assimilation AE and somatic maintenance cost  . Somatic maintenance is a crucial suite  of functional trait used in recent bioenergetics based on the DEB theory that mechanistically can be used to investigate the role played by multiple stressors on LH traits of organisms by using first principles (Sarà et al., 2014). We further documented the effects of those stressors on *M. galloprovincialis* shells through the use of a scanning electron microscope (SEM), and compared the maximum shell breaking load of treated *vs.* control specimens. A behavioural analysis completed the frame concerning the individual's response to both single and combined stressors. Carried out in a context of OA, this exercise comprises a first step in linking the fields of ecomechanics and climate change ecology, which should yield a more mechanistic understanding of how biodiversity will respond to environmental change (*sensu* Buckley et al., 2012).

                         **2 Materials and methods**

This study  consisted of three steps: 1) laboratory investigation on the effects of pH and hypoxia on functional  (both behavioural and physiological) traits of *Mytilus galloprovincialis*; 2) collection of water temperature data and CHL-a data from two Mediterranean sites (Trieste and Palermo), as a further forcing variable in the DEB model and lastly 3) model running to simulate growth and fitness of *M.*

*galloprovincialis* under stressful conditions by using  DEB parameters estimated  by activities in the first step.

**2.1 Experimental set-up.** Specimens of *M. galloprovincialis* (45 - 55 mm) were provided by the Ittica Alimentare Soc. Coop. Arl. (Palermo) and transferred within 30 minutes to the laboratory. Mussels were then carefully cleaned and placed in a 300L tank filled with natural seawater at room temperature (18-

20°C), field salinity (37-38 ) and fed *ad libitum* with cultured *Isochrysis galbana* (Sarà et al., 2011).

Mussels were acclimated for two weeks to reduce stress generated by manipulation and transport (Sarà et al., 2013a)  once acclimated, 200 specimens were randomly divided in groups of 25

organisms, transferred to 8 independent rectangular glass tanks of 120L capacity (100 cm long, 30 cm deep, 40

cm wide) and kept in a conditioned room at 21°C for 4 weeks according to common protocol with bivalves (Braby & Somero 2006; Fields et al., 2012; Kittner and Riisgård 2005). Tanks 1 to 4 were filled with sea water and continuously  aerated through air pumps, while Tanks 5 to 8 were not aerated and covered with a plastic film disposed on the water surface, in order to avoid gas-exchanges between air and water. Tanks 1-2 were used as a control (CTRL), while hypercapnia was imposed in Tanks 3-4 (Tr1), hypoxia (2

ppm) in Tanks 5-6 (Tr2), and both factor (pH 7.5 and hypoxia) in Tanks 7-8 (Tr3) (see Table 2). Mussels were acclimated to two different nominal pH treatments: (i) pH 8.0 in Tanks 1-2 (CTRL) and 5-6 (Tr2), corresponding to present average pH at the sampling site; and (ii) pH 7.5 in Tanks 2-3 (Tr1) and 7-8 (Tr3), deviating from present range of natural variability and relevant for 2100 ocean acidification scenarios.

(Melzner et al., 2011; Gazeau et al., 2013). The carbonate system speciation ($pCO2$, $HCO_3^-$, $CO_3^{2-}$, $\Omega Ca$ and

$\Omega Ar$) was calculated from $pH_{NBS}$, temperature, salinity and alkalinity ($T_A$ = 2.5 mM; Rivaro et al., 2010) using

CO2SYS (see Table 2; Lewis and Wallace, 1998) with dissociation constants from Dickson & Millero (1987).

The pH was manually controlled 8 times a day by an electronic pH-meter (Cyberscan 510, Eutech Instruments; accuracy = ± 0.01 pH) and gaseous $CO_2$ was injected directly into the aquarium through a commercial ceramic diffusor, when required. Oxygen concentration and temperature were monitored with the same frequency through the PiroScience FirestingO2 oxygen logger equipped with a dedicated temperature sensor. Water movement and recirculation were assured by water pumps. Tanks were siphoned at the end of each  day, removing all the faecal material in order to avoid the accumulation of waste products, and 20% of water was weekly changed with specific pre-conditioned sea water for each treatment.

**2.2 Behavioural observations.** The valve gape of mussels was recorded by means of the two simplest behavioural categories reported in Jørgensen et al. (1988): closed valves and opened valves. Each observation was carried out by an operator with the aim to record changes in the behavioural repertoire of bivalves in response to the exposure to a single stressor (pH or hypoxia) and to both pH and hypoxia, compared to individuals kept in normal environmental conditions. All experiments were conducted at environmental (37-38

) salinity and with well-aerated sea water through a gentle flow (Ameyaw-Akumfi & Naylor, 1987), except for specimens of Tank 5-6 and 7-8, that were not aerated in order to maintain the hypoxia level set through the gaseous nitrogen. Behavioural observations were repeated six times a day at week 1 and 4 of exposure, involving 5 random specimens for each treatment.

**2.3 Oxygen consumption.** The rate of oxygen consumption was determined twice (week 1 and week 4) in a respirometric glass chamber (0.3L) inside a temperature-controlled water bath, in order to  investigate the effects of multiple stressors  on metabolic somatic maintenance costs and to integrate it in the standard DEB model. Volume-specific somatic maintenance costs, as expressed by the $[\dot{p}_M]$ parameter (J cm$^{-3}$ h$^{-1}$), represent the amount of energy needed to fuel basal metabolism ($\dot{p}_M$) scaled with the organisms' volume,  ($[\dot{p}_M] = \dot{p}_M/V$). All determinations were performed at 21°C using filtered seawater with the same pH and oxygen content as that of the respective treatment, stirred with a magnetic stirrer bar beneath a perforated glass plate supporting each individual ( Ezgeta-Balic et al., 2011). The decline in oxygen concentration was measured by a PiroScience FirestingO2 respirometer, capable of four sensor connections. We used a total of n = 64 mussels per week, 16 for each treatment (8 for each tank) acclimated as above, fed *ad libitum* until the day before the experiment. The decline was continuously recorded for at least 1 h, excluding an initial period (~ 10 min) when usually there is a more rapid decline in oxygen caused by a disturbance of the sensor's temperature equilibration. Respiration rate (RR, µmol O$_2$ h$^{-1}$) was calculated according to ( Sarà et al.,  2013b): $RR = (C_{t0} - C_{t1}) x\ Vol_r x 60(t_1 - t_0)^{-1}$, where $C_{t0}$ is oxygen concentration at the beginning of the measurement, $C_{t1}$ is the oxygen concentration at the end of the measurement, and $Vol_r$ is the volume of water in the respirometric chamber. Volume-specific somatic maintenance costs were then calculated by converting oxygen consumption rates expressed in µmol h$^{-1}$ in J h$^{-1}$ through a conversion factor (Kooijman 2010) and then in J cm$^{-3}$ (van der Veer et al., 2006; Ren and Schiel 2008) (for the calculation of dry weights refer to the end of section 2.4).

**2.4 Assimilation efficiency.**  Assimilation efficiency (AE) was measured through the Conover ratio (1966) AE = (F − E)/[(1 − E)F], where F is the ratio between ash-free dry weight (AFDW) and dry weight (DW) for food, and E is the same ratio for the faeces; this represents the efficiency with which organic material is absorbed from the ingested food material. Here, after oxygen consumption measurement, the same  specimens  were  placed into separate beakers containing 1L of filtered seawater (specific for each treatment) and a magnetic stirrer bar. In order to allow the mussels to open their valves and start their filtration activity, they were given 15 minutes before the introduction of food with an initial concentration of ~ 15,000 *Isochrysis galbana* cells ml$^{-1}$. After a period of 2 h mussels were moved to cleaned 1L glass beakers with filtered seawater for a period of 12 h, after that the water contained in each beaker was filtered on pre-ashed and weighted GF/C fibreglass filters. Once filtered, filters were washed with 0.5 M ammonium formate (purest grade) to remove adventitious salts (Sarà et al., 2013a), dried in the oven (95°C for 24 h) and then incinerated in a muffle furnace (450°C for 4 h). After each step, the samples were weighted using a balance (Sartorius BL 120S ± 1µg). For the calculation of AE, together with the faeces collected from the mussels, filters containing algal food were dried and incinerated as above. After respirometric measurement and the collection of faeces each animal was killed by gentle freezing and dissected, and the shells were separated from the body tissue  to calculate the condition index according to Davenport & Chen (1987) (CI = (body weight/shell weight) × 100), and their individual dry weights  to standardize respiration rates.

**2.5 Water temperature data.** The main forcing driver of shellfish LH inside DEB models is represented by  seawater temperature (Pouvreau et al., 2006; Kearney et al., 2010; Kooijman, 2010; Sarà et al., 2011; 2013). DEB simulations were run under subtidal conditions (body temperature was expressed by the mean seawater temperature; Montalto et al., 2014) with 4 years-hourly data (Jan 2006 - Dec 2009) of seawater temperature measured about 1 m below the surface at the closest meteo-oceanographic station held in Trieste (LAT 45° 38′ 57.81″; LONG 13° 45′ 28.58″) and Palermo (LAT 38° 07′ 17.08″; LONG 13° 22′ 16.79″). The period of 4 years is consistent with the normal life span of most Mediterranean shellfishes (Sarà et al., 2012; 2013b). Both sites were chosen as they represent two opposite temperature and food conditions for mussel growth in Italy, with Trieste as representative of lower temperature (average 16.98 ± 6.19 °C) and higher food levels (average 1.36 ± 0.37 CHL-a), and Palermo of higher temperatures (average 20.19 ± 4.64 °C) and lower food (average 0.19 ± 0.09 CHL-a). Data are available online from the Italian Institute of Environmental Research (ISPRA) web page (http://www.mareografico.it/).

**2.6 CHL-a dataset.**  (CHL-a was derived from satellite imageries (µg L$^{-1}$; http://emis.jrc.ec.europa.eu/))  and adopted as a reliable food quantifier for suspension feeders (Kearney et al., 2010; Sarà et al., 2011; 2012.

**2.7 Model description.** The Dynamic Energy Budget (DEB) Theory provides a general framework that allows to describe how physiological mechanisms are driven by temperature and food availability, and influences growth and the reproductive performances in marine organisms (Sousa et al., 2010; Monaco et al., 2014; Jusup et al., 2017). Following the κ-rule (DEB theory; Kooijman, 2010) a fixed energy fraction (κ) is allocated to growth and somatic maintenance, while the remaining fraction (1-κ) is allocated to maturity maintenance plus maturation or reproduction. If the general environmental conditions deviates from common natural patterns (i.e. changes in temperature, food availability etc.) reproduction and growth are consequently affected. According to the DEB theory, a reduction in growth can be caused either by reduced food assimilation ($\dot{p}_A$), enhanced maintenance costs [$\dot{p}_M$]($\dot{p}_M$), or enhanced growth costs ($\dot{p}_G$). Using this approach, and through the DEB parameters derived from Sarà et al. (2012)reported in Table 1, except for the variation in the maintenance costs [$\dot{p}_M$]($\dot{p}_M$) and in the assimilation efficiency of food (AE) which were experimentally estimated throughout this study, we performed simulations using thea sStandard version of the DEB model (Nisbet et al., 2010) aimed at investigating the potential variations in growth and fecundity of our model species. To run the DEB simulations, local thermal series of selected sites were used together with satellite CHL-a concentrations, obtaining a first model with environmental conditions. A second model was run with the [$\dot{p}_M$]$\dot{p}_M$ calculated from the oxygen measurements on specimens of *M. galloprovincialis* from Tanks 3-4 (pH 7.5) simulating a chronic hypercapnia condition for the full cycle (4 years) and the relative estimated AE. Subsequently, further models were run by simulating one random hypoxia event (duration = 30 days) for each of the four years of the cycle, then simulating two yearly events, and so on up to six monthly hypoxia events. The starting month of each event was randomly chosen for every year with the use of a table of random digits. The [$\dot{p}_M$]$\dot{p}_M$ calculated from the oxygen consumption rate measurements on specimens from Tanks 7-8 (pH 7.5 and hypoxia) was used in substitution to [$\dot{p}_M$]$\dot{p}_M$ from pH 7.5 tanks 3-4, coupled with the relative estimated AE, when simulating both stressors. Simulations were performed using the R routine for Standard DEB model developed by M. Kearney (2012), and further modified (for use in bivalve modelling) by Sarà et al. (2013). Outputs of the DEB models (Sarà et al., 2014) were: the maximum theoretical total length of shellfish (TL), the maximum total weight (TW), the total number of eggs (TRO) produced during a life-span of 4 years, the total number of reproductive events (RE) and the time needed to reach gonadic maturity (TM) for each treatment.

**2.7.1 Model limitation.** DEB models are particularly useful to quantitatively assess the effects of multiple stressors on LH-traits in an integrated manner, leading to test the hypothesis on how OA may affect the maintenance costs of living organisms (Jager et al., 2016). Maintenance costs, as defined by Dynamic Energy Budget Theory (Kooijman, 2010), represent the energy requirement of an organism to survive, excluding investments in growth, reproduction and development. The volume-specific somatic maintenance costs parameter [$\dot{p}_M$] within the standard DEB model has been up to date estimated only by indirect approaches through changes in energy content by starvation over time (van der Meer, 2006) or measurements of respiration rate of starved organisms (van der Veer et al., 2006; Ren and Schiel 2008). The idea of quantitatively assess the effect of a stressor including it as a modification of a specific parameter was first introduced by Jager et al. (2016) with the *stress factor* "s" applied to assimilation, maintenance and cost of growth. Thus, after estimating the effect induced by a treatment on the oxygen consumption, in our case expressed as percentage variation, we summed/subtracted the energetic amount due to the effect of a stressor to the species-specific [$\dot{p}M$] parameter of *M. galloprovincialis* (Sarà et al. 2012) then we run our models. Previous proposed approaches, taking into account starvation for [$\dot{p}_M$] estimation, wouldn't be realistically applicable for testing and quantifying the effect of a stressor on the energy budget, without adding a further stressor. Jager et al. (2016) was therefore the first to adopt this concept, although using a simplified DEB model (DEBkiss; Jager et al. 2013) that did not involve the concepts of reserve and maturity that play a central role in DEB theory. Although this may not be considered a reliable measure of maintenance costs but a simpler proxy of metabolic effect, negligible costs for growth and gonadic development stand on the assumption of constant protein turnover throughout the experimental range (Hawkins et al. 1989).

[revised manuscript text omitted]

Berge et al., 2006) counterbalancing the  effect  of dissolved $CO_2$ crossing biological membranes . Compensation of low pH, associated with anthropogenic increases in seawater $pCO_2$ (Fabry et al., 2008), through adjustments in ionic composition appears to be a trade-off that is not likely sustainable  over longer time-scales.

From our behavioural observations mussels exposed to low pH resulted in a higher, even if not significant, percentage of opened valves respect to CTRL individuals, with the highest significant difference relative to hypoxia exposition. The effect of low pH on the adductor muscle of bivalves has been already documented by Pynnönen & Huebner (1995) and the same effect has been reported after exposition to hypoxia (Sheldon & Walker 1989). In agreement with current literature showing deleterious effects of $CO_2$-induced acidification on a wide range of invertebrates (Barnhart & McMahon, 1988; Barnhart, 1989; Rees & Hand, 1990)

and similarly to other studies by *M. galloprovincialis* (Gestoso et al., 2016; Michaelidis et al., 2005), our results showed how hypercapnia (pH reduced by 0.6 units, relative to the natural pH of the lower Tyrrhenian waters) was able to induce a decline in metabolic rates of mussels. This kind of decline has already been noticed by other authors as an adaptive strategy for survival under transiently stressful conditions (Michaelidis et al., 2005).

According to Pörtner et al. (2004), metabolic reduction due to hypercapnia could be a result of acid-base disturbances and therefore be similar to the response of intertidal individuals to anaerobic conditions. Direct effects of hypoxemia have been further proven to cause fatal decrements in an organism's performance in growth, reproduction, feeding, immunity and behaviour (*sensu* Pörtner & Farrell, 2008).  Combined effects by stressors  such as ocean acidification and hypoxia narrow  the thermal window of functioning according to species-specific sensitivities, modulating biogeographices, coexistence ranges, community shifts and in general ecological interactions (Pörtner & Farrell, 2008). The mussel *Mytilus*

*edulis* has been proven able to compensate both short- and long-term exposure to hypercapnia by dissolution of shell (Lindinger et al., 1984; Michaelidis et al., 2005), resulting in reduced growth and metabolism. A similar mechanism of release of inorganic molecules into the pallial cavity (as $CaCO_3$ from valves) has been documented during periods of anaerobic metabolism, to maintain the acid-base balance (Chaparro et al., 2009), determining further physiological and energetic cost such as decreased growth, respiration rate and protein synthesis (Pörtner et al., 2005). During periods of environmental oxygen limitation, many organisms are able to suppress ATP demand, shut down expensive processes, such as protein synthesis (Hand, 1991), but at the same time limiting growth and the reproductive potential. Although suppression of metabolism under short-term experimental conditions is a

"sub-lethal" reversible process, reductions in growth and reproductive output will effectively diminish the survival of the species on longer time-scales (Fabry et al., 2008). The contemporary occurrence in our simulations, of monthly hypoxia events, revealed a growing  contribution to what was already elicited by hypercapnia alone on growth and reproduction. Current literature has not currently explored the combined effects of multiple stressors on long-term experiments by modulating the intensity and duration of disturbance. This would probably translate as a very complex experimental set-up which would be hardly practicable, especially on long-term scales.

On the other hand, mechanistic models offer a more practical alternative to long-term, in- field research when studying the effects of multiple-stressors, with the advantage of testing, at the same time, the magnitude and the duration of disturbance on LH _traits of a model species. Our results highlighted the general hypoxia growing effect following the increasing duration of disturbance, with a particular focus in Trieste on TW

and TRO, while in Palermo on TW and TM (Fig. 3). A further important peculiarity of the mechanistic modelling  deals with the possibility to spatially contextualise the effects of single and multiple stressors on selected outputs by integrating local thermal conditions and food concentrations (Sarà et al.

2018c). Comparing the effect of hypoxia across frequencies (Fig. 4), total length (TL) resulted unaffected up to a frequency of 2 hypoxia events in both sites, then the highest effect was recorded in the eutrophic site (Trieste).

Trieste, between the two chosen sites, had  the lowest temperature. On the contrary a smaller effect of hypoxia was detected  on  in

Trieste, suggesting  a sort of food compensation capacity on the effect of environmental stressor (Mackenzie et al., 2014). Also the DEB model easily allowed the estimation of the fecundity potential of  organisms, that is often omitted in other ecological studies, but that represents a crucial quantity for resource (e.g. aquaculture) (Sarà et al. 2018c) and conservation purposes. To verify impacts on shellfish fecundity, we contextualised our simulation by introducing Trieste hourly temperature series after those of Palermo, with the respective local actual CHL-a concentrations, as long as in the first site no reproductive events came out from our simulations, probably due to food limitations and temperature threshold. This is reflected by natural populations in Palermo colonising only substrates in  trophic- enriched sites. A combined effect of the simultaneous stressors, such as those considered across this study, has proven in the present study to affect the organism's performance in growth, reproduction and behaviour. Our results highlighted an effect of pH alone and when combined with hypoxia on the breaking load of shells of our experimental mussels. Through a similar approach, Martinez et al. (2018) showed that temperature was a primary factor driving shell's fragility along a latitudinal gradient. Present findings corroborates that idea that fragility can be affected by both stressors through a combined effect. Multiple stressors can narrow, especially  when organisms are on the edge of their thermal tolerance range, the thermal window and this has a potential for generating repercussions on  biogeographical distribution, coexistence ranges, community shifts, food webs and species interactions (*sensu* Pörtner & Farrell,

2008). Moreover, an appropriate knowledge of species' biological traits, and a mechanistic understanding of the effect of each stressor, reached through an FT-based approach, will allow the translation of the effects of environmental change into realistic management measures taking into account the optimisation of the species' biological traits (Sarà et al. 2018a,b).

**6 Conclusions**

Additional research is still required to improve our knowledge of organismal response to multiple stressors, in particular, of many marine ectotherms with indeterminate growth amongst invertebrates (e.g. crustaceans, molluscs). Nevertheless, modelling the growth and reproductive potential (and failure) of species vulnerable to those stressors with predictive tools, such as bioenergetic models is a useful approach for management and protection purposes, but also for shellfish culture in general.

**Authors' contributions**
Both authors contributed to all phases of this ms. AG and GS conceived the idea and led the writing. AG carried out all experiments in mesocosms, performed modelling work and analysed data. GS provided lab facilities and research funds. All authors contributed critically to the drafts and gave final approval for publication.

**Acknowledgements**
PRIN TETRIS 2010 grant n. 2010PBMAXP_003, funded to GSARA by the Italian Minister of Research and University (MIUR) supported this research. The authors declare that they have no conflict of interest. We thank and are especially grateful to all collaborators involved in this paper, in particular to Dr. Alessandro Rinaldi, Matteo Mercurio and Marco Martinez for their technical support. We also thank Francesco Furnari for the use of the scanning electron microscope. We deeply thanks and Ms. Jan Underwood for the fine-tuning of the English.

**Tables**

Table 1. DEB parameters for *Mytilus galloprovincialis* (1 = Kooijman, 2010; 2= van der Veer et al., 2006; 3 = Sarà et al., 2011; 4 = Thomas et al., 2006; 5 = Schneider, 2008); Lb, Lp, Ls = length at birth, puberty and seeding respectively; dVw = specific density to convert weight into volume; $f$ = functional response type II f = X/(XK + X); µx = chemical potential to convert moles into food energy; SMI is the somatic mass index of both starved and well-fed animals, expressed as somatic ash free dry mass (AFDM, mg); $X_K$ = saturation coefficient expressed as a concentration of chlorophyll a (µg CHL-a $l^{-1}$), where the ingestion rate is half of the maximum.

| Symbol | Description | Formulation | Units | *Mytilus galloprovincialis* Value | Ref |
|---|---|---|---|---|---|
| $V_b$ | Structural volume at birth | $V_b = (L_b \times \delta_m)^3$ | $cm^3$ | 0.0000013 | 1 |
| $V_s$ | Structural volume at seeding | $V_s = (L_s \times \delta_m)^3$ | $cm^3$ | - | - |
| $V_p$ | Structural volume at puberty | $V_p = (L_p \times \delta_m)^3$ | $cm^3$ | 0.06 | 2 |
| $\delta_m$ | Shape coefficient | $\delta_m = (Ww \times d_{Vw}^{-1}) \times L^{-1}$ | - | 0.2254 | 3 |
| $\{J_{Xm}\}$ | Maximum surface area-specific ingestion rate | $\{J_{Xm}\} = J_X /(f \times V^{\frac{2}{3}})$ | $J\ cm^{-2}h^{-1}$ | 8.2 | 4 |
| ae | Assimilation efficiency | $ae = (\mu_x \times J_X)/p_A$ | - | 0.88 | 3 |
| $X_K$ | Saturation coefficient | - | $\mu g\ l^{-1}$ | 2.1 | 3 |
| $[E_G]$ | Volume-specific cost of growth | $[E_G] = SMI_{starved} \times 23 \times (\delta^3_m)^{-1}$ | $J\ cm^3$ | 5,993 | 5 |
| $[E_m]$ | Maximum storage density | $[E_m] = (SMI_{fed} - SMI_{starved}) \times 23 \times (\delta^3_m)^{-1}$ | $J\ cm^3$ | 2,190 | 2 |
| $[p_M]$ | Volume-specific maintenance cost | $[p_M] = p_M/V$ | $J\ cm^{-3}\ h^{-1}$ | 1 | 2 |
| $\kappa$ | Fraction of utilized energy spent on maintenance and growth | - | - | 0.7 | 2 |
| $K_R$ | Fraction of reproductive energy | - | - | 0.8 | 3 |
| $T_A$ | Arrhenius temperature | $T_A = \ln |K_{(T_0)}/K_{(T_1)}| x \frac{(T_1 x T_0)}{(T_0 - T_1)}$ | °K | 7,022 | 2 |
| $T_L$ | Lower boundary of tolerance range | | °K | 275 | 2 |
| $T_H$ | Upper boundary of tolerance range | | °K | 296 | 2 |
| $T_{AL}$ | Rate of decrease at lower boundary | | °K | 45,430 | 2 |
| $T_{AH}$ | Rate of decrease at upper Boundary | | °K | 31,376 | 2 |

Table 12. Seawater carbonate chemistry parameters (mean ± se). Seawater pH on the NBS scale (pHNBS), temperature (T; °C), and salinity were used to calculate $CO_2$ partial pressure ($p$CO2; µatm) as well as aragonite and calcite saturation states (respectively Ωar and Ωca), for a total alkalinity of 2500 mmol $kg^{-1}$.

| | Measured | Calculated |
|---|---|---|
| | | |

| | Temperature (°C) | pH$_{NBS}$ | O$_2$ mg/l | Salinity (PSU) | pCO$_2$ (μatm) | CO$_3^-$ | Ωca | Ωar |
|---|---|---|---|---|---|---|---|---|
| CTRL | 20.77 ± 0.01 | 8.01±0.001 | 7.29±0.02 | 37.18±0.11 | 624.31±4.9 | 167.93±0.95 | 3.95±0.02 | 2.58±0.01 |
| Tr1 | 20.77 ± 0.01 | 7.53±0.002 | 7.30±0.02 | 37.12±0.05 | 2151.17±22.02 | 62.05±0.73 | 1.46±0.02 | 0.95±0.01 |
| Tr2 | 20.77 ± 0.01 | 8.01±0.001 | 2.44±0.02 | 37.07±0.04 | 729.88±18.24 | 152.53±1.51 | 3.59±0.04 | 2.34±0.02 |
| Tr3 | 20.77 ± 0.01 | 7.53±0.002 | 2.44±0.02 | 37.21±0.17 | 2238.83±20.72 | 59.59±0.42 | 1.40±0.01 | 0.91±0.01 |

Table 2. ANOVA on seawater chemistry parameters. Comparison between CTRL (normal pH) and TREAT (low pH and hypoxia) (* = p < 0.05; ** = p < 0.01; *** = p < 0.001; ns = not significant).

| | | pH$_{NBS}$ | | | pO$_2$ | | |
|---|---|---|---|---|---|---|---|
| | df | MS | F | p | MS | F | p |
| TREAT | 3 | 10.73 | 41450.84 | ** | 1083.21 | 18798.36 | ** |
| Residuals | 548 | 0.0003 | | | 0.06 | | |
| Cochran's Snk | | | | * | | | * |

| | | pCO$_2$ | | | CO$_3^-$ | | |
|---|---|---|---|---|---|---|---|
| | df | MS | F | p | MS | F | p |
| TREAT | 3 | 1.06e08 | 2426.84 | ** | 460157.7 | 3433.17 | ** |
| Residuals | 548 | 43851.09 | | | 134.03 | | |
| Cochran's Snk | | | | * | | | * |

| | | Ωca | | | Ωar | | |
|---|---|---|---|---|---|---|---|
| | df | MS | F | p | MS | F | p |
| TREAT | 3 | 254.09 | 3432.44 | ** | 108.26 | 3426.14 | ** |
| Residuals | 548 | 0.07 | | | 0.03 | | |
| Cochran's Snk | | | | * | | | * |

Table 3 ANOVA table of results. Effect on valve gape and breaking load of *Mytilus galloprovincialis* (* = p < 0.05; ** = p < 0.01; *** = p < 0.001; ns = not significant).

| Source | df | Valve gape | | | Source | df | Breaking load | | |
|---|---|---|---|---|---|---|---|---|---|
| | | MS | F | P | | | MS | F | P |

| Treatment (Tr) | 3 | 17.41 | 15.60 | *** | Treatment (Tr) | 3 | 3838.12 | 15.18 | *** |
|---|---|---|---|---|---|---|---|---|---|
| Time (Ti) | 84 | 2.08 | 1.87 | ns | Time (Ti) | 1 | 777.19 | 9.22 | ** |
| Tr x Ti | | 0.3056 | 0.27 | ns | Tr x Ti | 3 | 132.92 | 1.58 | ns |
| Residuals | 40 | | | | Residuals | 56 | | | |
| Cochran's C | | | | ns | Cochran's C | | | | ns |

Table 4 ANOVA table of results. Respiration rate (RR) and assimilation efficiency (AE) of *Mytilus galloprovincialis* (* = p < 0.05; ** = p < 0.01; *** = p < 0.001; ns = not significant).

| Source | df | RR st | | | AE | | |
|---|---|---|---|---|---|---|---|
| | | MS | F | P | MS | F | P |
| Treatment (Tr) | 3 | 312.9183 | 6.95 | *** | 0.2783 | 12.21 | *** |
| Time (Ti) | 1 | 205.1325 | 4.56 | * | 0.0424 | 1.86 | ns |
| Tr x Ti | 3 | 40.7752 | 0.91 | ns | 0.0198 | 0.87 | ns |
| Residuals | 120 | 45.0271 | | | 0.0228 | | |
| Cochran's C | | | | * | | | ns |

Table 5 DEB simulation outputs. Percentage variation of treatments from CTRL: Total length (TL), Total weight (WW), Total reproductive output (TRO), Total reproductive events (RE), Time to maturity (TM).

| DEB outputs (CTRL) after 4 years | | | | | | | | |
|---|---|---|---|---|---|---|---|---|
| Site | Stressor | Hypoxia events (days) | Frequency (1/Time) | TL (cm) | WW (g) | TRO (n° egg) | RE | TM (days) |
| Trieste | CTRL | 0 | 0 | 9.55 | 11.19 | 6.74e6  | 9 | 232 |
| Palermo | CTRL | 0 | 0 | 3.08 | 0.31 | 0 | 0 | 739 |

|  | | | | | | | |
|---|---|---|---|---|---|---|---|
|  |  |  |  |  |  |  |  |
|  |  |  |  |  |  |  |  |
|  |  |  |  |  |  |  |  |
|  |  |  |  |  |  |  |  |
|  |  |  |  |  |  |  |  |
|  |  |  |  |  |  |  |  |
|  |  |  |  |  |  |  |  |
|  |  |  |  |  |  |  |  |
|  |  |  |  |  |  |  |  |
|  |  |  |  |  |  |  |  |
|  |  |  |  |  |  |  |  |
|  |  |  |  |  |  |  |  |
|  |  |  |  |  |  |  |  |
|  |  |  |  |  |  |  |  |
|  |  |  |  |  |  |  |  |

| Percentage  contributing effect of Hypoxia | | | | | | | | |
|---|---|---|---|---|---|---|---|---|
| Trieste | pH+hypoxia | 30 | 0.08  | -0.6 | -1.4 | -1.1 | 0 | 0.8 |
| Trieste | pH+hypoxia | 60 | 0.17  | -1.3 | -2.8 | -2.8 | 0 | 2.1 |
| Trieste | pH+hypoxia | 90 | 0.25  | -2 | -4.3 | -4.5 | 0 | 3.6 |
| Trieste | pH+hypoxia | 120 | 0.33  | -2.7 | -5.8 | -6 | 0 | 4.4 |
| Trieste | pH+hypoxia | 150 | 0.42  | -3.4 | -7.2 | -7.4 | 0 | 5.5 |
| Trieste | pH+hypoxia | 180 | 0.50  | -4 | -8.4 | -8.8 | 0 | 6 |
| Palermo | pH+hypoxia | 30 | 0.08  | -0.6 | -1.6 | 0 | 0 | 0.8 |
| Palermo | pH+hypoxia | 60 | 0.17  | -1.3 | -3.3 | 0 | 0 | 2.1 |
| Palermo | pH+hypoxia | 90 | 0.25  | -1.9 | -4.6 | 0 | 0 | 3.8 |
| Palermo | pH+hypoxia | 120 | 0.33  | -2.6 | -6.1 | 0 | 0 | 4.5 |
| Palermo | pH+hypoxia | 150 | 0.42  | -3.2 | -7.4 | 0 | 0 | 5.7 |

| | | | month | | | | | |
|---|---|---|---|---|---|---|---|---|
| Palermo | pH+hypoxia | 180 | 0.50 | -3.9 | -8.9 | 0 | 0 | 7.6 |

---

## Author Comment (AC2) · 2 Apr 2018

REVIEWER #1 Reviewer wrote: This paper explores the implications of environmental stress (OA and hypoxia), as determined in lab experiments, on the growth and reproductive potential of mussels in two locations of the Mediterranean with simulations based on a dynamic energy budget model. The study capitalizes on the potential of DEB models to integrate the impacts of multiple environmental drivers on organismal level outcomes, including growth, reproduction, time to maturity, rates of feeding and respiration, and so on. This approach is powerful in potential and the application is new. However, there are some important shortcomings, especially in the way the model is parameterized. Also, I find the lack of some form of quality assessment problematic. Author's reply: First, we thank the Reviewer #1 for the effort in providing his/her suggestions to the original version of our ms. Author's changes: Apart from all possible not clear parts that we accordingly improved, we followed his/her suggestion in the hope to have increased the quality of this ms.

Reviewer wrote: It is very annoying that all sections consist of a single paragraph. Did something go wrong with the formatting of the manuscript? Author's reply: We appreciated the referee's suggestion about the different paragraphs, but we believe that the current structure is already sufficiently sectioned; to increase the number of paragraphs or sub-paragraphs can only increase the text fragmentation which may limit the logical flow of the text. Reviewer wrote: The author's use of respiration measurements as a proxy for DEB maintenance costs is problematic. In the DEB framework, respiration is emphatically not the same as maintenance, but also include energetic overheads, such that of growth. Respiration is a function of the commitment rate in DEB, of which maintenance could be a minor part, depending on size and nutritional status of the animal. Author's reply: We acknowledge that respiration does not include only maintenance, but also include energetic overheads, such as growth. Nevertheless, there is no way - to our knowledge - to measure the different contribution of every energetic components apart from to experimentally measure oxygen consumption as a proxy for metabolism. Also, the proposed approaches measuring indirectly the [áźŮM] values (e.g. van der Veer et al., 2006; Ren and Schiel 2008), are not feasible in the context of the present experimental asset. While this approach is not experimentally feasible when assessing the effect of stressors on the energy budget, the only way to indirectly provide an estimation of the effect of disturbance is through the Jagger et al. (2016) approach which is based on the stress factor "s". Thus, after estimating the effect induced by a treatment on the oxygen consumption, that in the present case study was expressed as a percentage variation, we summed/subtracted the energetic amount due to the effect of a stressor to the species-specific [áźŮM] values of M. galloprovincialis then we run our models. However, we thank the referee for highlighting this point whose importance was addressed in the Discussion section of this ms. as we believe that is crucial to increase our understating on how we can mechanistically assess the effect of disturbance on individual performances through the DEB model. All these limitations show how much is important to date to increase the experimental and theoretical research effort in order to unravel this point, which is increasingly crucial to get realistic answers to management questions in a context of environmental change.

Reviewer wrote: In addition, oxygen deprived mussels, and possibly mussels enduring stress of hypercapnia, are able to use anaerobic metabolic pathways to fulfil their maintenance requirements. If stress increases maintenance requirements, one would expect respiration rates to increase with increasing stress intensity. However, we see the opposite happen (see Fig 2). I think this is likely due to the fact that stressed mussels have their shells closed more often than unstressed conspecifics (see Fig. 1), and thus ingest less food. Less food leads to a lower energy reserve buffer and therefore a lower rate at which reserves are committed. Author's reply: As reported in our results and Fig. 2, maintenance requirements in accordance with respiration rates, decreases with stress in agreement with what is reported in the current literature. We are sorry with Reviewer #1 and with all readers as we made a mistake in writing the text commenting the figure 1 (we wrote wrongly "opened" instead of "closed"). Actually, our mussels increased their openness with the increasing stressful conditions. At the present stage, we are not able to provide information about the amount of ingested food under different treatments and then we are not able to infer on the effect of openness degree on energetic performances. Author's changes: Figure was fixed according to both referee's suggestion, and also the text in the paragraph has been rephrased accordingly.

Reviewer wrote: I suggest the authors change the maximum assimilation rate parameter of their model based on their behavioural observations and leave the maintenance rate parameter unchanged. Author's reply: We appreciated the referee's suggestion but we prefer to focus on both the assimilation efficiency and the metabolic effect

(through pM) as i) the main effect of acidification seem to be exerted on metabolism as widely reported in the current literature, and ii) also to show that our mechanistic DEB approach can be really effective in measuring the multiple stressor's effect on LHs. Author's changes: Thus, we enlarged the discussion on these points to include possible shortcomings deriving from the fact that the stressor's effect on maintenance is not still well-experimentally measurable.

Reviewer wrote: The simulations suggest that unstressed mussels only grow to 3 cm in length and do not reproduce in Palermo. This seems implausible. How long do real mussels get in Palermo? Do they reproduce? How sensitive are the simulation results to the particular choices of parameter values? The authors do not reflect at all on the reliability of their assessments, which I find troublesome, especially given the politicized context of the subject matter. Author's reply: Actually, to enlarge the discussion about the magnitude of effects at local level could be not influent for our purposes, although our results are in line with the environmental and trophic conditions reported in section 2.5: "Both sites were chosen as they represent two opposite temperature and food conditions for mussel growth in Italy... etc.". M. galloprovincialis in Sicily is observed to be limited by oligo-trophic conditions although it grows in highly trophic-enriched areas such as harbours or under Integrated Multi-Trophic Aquaculture (IMTA) conditions (Sarà et al 2012; 2013b, Giacoletti et al. 2018 in press JEMA) which supports what we gathered in the present ms. through the DEB simulations.

Reviewer specific comment n. 1 Reviewer wrote: Title. Functional spatial contextualization sounds impressive but I've no clue what it could mean. Also, the manuscript deals with only a single species; the title is too general. Author's changes: We agreed with Reviewer's #1 point and changed the title.

Reviewer specific comment n. 2 Reviewer wrote: L27-33 Split up sentence. Author's changes: Sentence was splitted up accordingly.

Reviewer specific comment n. 3 Reviewer wrote: L35 (and elsewhere) Put reference in the proper place of the sentence Author's changes: All references were checked and put in proper spaces.

Reviewer specific comment n. 4 Reviewer wrote: L40 'lager'? Author's changes: Changed with "larger".

Reviewer specific comment n. 5 Reviewer wrote: L68-70. This is a strong statement and should be substantiated with references. BTW, the only 2 papers using DEB in a OA context I'm aware of are 10.1111/gcb.12547 and 10.1016/j.jembe.2015.09.016 Author's changes: References regarding the effect of OA on functional traits such as feeding and assimilation, and on maintenance costs has been added accordingly.

Reviewer specific comment n. 6 Reviewer wrote: L72 the DEB [p_M] parameter does not relate to assimilation Author's changes: The sentence was rephrased accordingly.

Reviewer specific comment n. 7 Reviewer wrote: L83 articulated ! consisted of Author's changes: Changed accordingly.

Reviewer specific comment n. 8 Reviewer wrote: Section 2.4 contains material that should go in 2.3 (or combine the sections). Author's reply: Section 2.4 refers to assimilation efficiency, while section 2.3 to oxygen consumption measures, so we consider not easy to combine both sections as we may incur in the risk to reduce the readability of this section.

Reviewer specific comment n. 9 Reviewer wrote: I didn't get how the authors calculate the assimilation efficiency. Author's changes: A detailed explanation on how the assimilation efficiency was estimated, was added with supporting references.

Reviewer specific comment n. 10 Reviewer wrote: Section 3.1 belongs in the Materials and Methods Section. There is no need for a statistical analysis. Delete Table 2. Author's changes: Table 2 was deleted according to both referee's suggestion and details were moved in the Materials and Methods Section.

Reviewer specific comment n. 11 Reviewer wrote: Combine Sections 3.2-4. There is no need to duplicate in the text what is already presented in the figures. The percentage of closed valves is simply 100 – percentage opened valves, so don't mention the former. I don't understand why the error measures differ so much, though. Author's changes: We agree not to duplicate in text what is already presented in figures, and we worked to avoid this replication. Following Reviewer's #1 suggestion we also expressed the percentage of closed valves 100 – percentage opened valve. Instead, merging the sections can increase the risk of confusion in the reader as section 3.2 is about behavioural observations, while the other two are about physiological measurements.

Reviewer specific comment n. 12 Reviewer wrote: Section 2.7 is incomprehensible for people without DEB modeling background. Include a figure and references to overview texts (e.g. Kooijman's book, Nisbet et al JAE, Sousa et al, and/or most recently Jusup et al Physics of Life Reviews 20:1-39). Author's changes: We agree with Reviewer #1 that section 2.7 is difficult for someone without a DEB modelling background, and in order to made it more clear we added the suggested references and rephrased some parts.

Reviewer specific comment n. 13 Reviewer wrote: L263 addictive ! additive. The way the authors use 'additive' is confusing. Additive refers to impacts that can be summed, like 1+1=2, an unlikely situation with nonlinear models, such as DEB. Author's changes: Rephrased following suggestions to "with a progressive contribution of hypoxia".

Reviewer specific comment n. 14 Reviewer wrote: What are the initial conditions of the simulation runs? Author's reply: Results of simulation performed with unstressed organism were already reported in Table 5, while model parameters have been reported in Table 1.

Reviewer specific comment n. 15 Reviewer wrote: What is the rational for the choices for the frequency of events? Author's reply: While we are aware that hypoxia events are more frequently during summertime, we decided to not apply any timing and frequency scheme to simulate hypoxia event's occurrence according to many papers published across the recent literature(Crain et al. 2008 Ecology Letters; Miller et al. 2009 PNAS).

Reviewer specific comment n. 16 Reviewer wrote: From Table 5 remove data that are already presented in Figure 3. Round off # of eggs to 6.74e6. Units of frequency should be 1/time Author's changes: Table 5 was corrected accordingly.

Reviewer specific comment n. 17 Reviewer wrote: Figure 3 label y axis 'Change relative to control' Author's changes: Figure was corrected accordingly.

Reviewer specific comment n. 18 Reviewer wrote: L294 delete 'formally' Author's changes: Deleted.

Reviewer specific comment n. 19 Reviewer wrote: L295 delete 'compensatory' and change contrast to compensate Author's changes: Rephrased.

Reviewer specific comment n. 20 Reviewer wrote: L299 suppressed feeding activity Author's changes: Changed.

Reviewer specific comment n. 21 Reviewer wrote: L304 what is crossing effect? Author's changes: Sentence has been rephrased accordingly.

Reviewer specific comment n. 22 Reviewer wrote: L306 on! over. 'that' doesn't refer to anything Author's changes: Rephrased.

Reviewer specific comment n. 23 Reviewer wrote: L333 sustainable and reliable ! practical Author's changes: Changed.

Reviewer specific comment n. 24 Reviewer wrote: L337 write out TW, TRO and TM Author's changes: Written out accordingly.

Reviewer specific comment n. 25 Reviewer wrote: The readability of the manuscript would improve if there were fewer references. Remove unnecessary repetitive references. Author's changes:: All the references has been checked and unnecessary and repetitive ones have been deleted accordingly.

**BGD**

Please also note the supplement to this comment:
https://www.biogeosciences-discuss.net/bg-2018-13/bg-2018-13-AC2-supplement.pdf

**Supplement:**

**REVIEWER #1**
**Reviewer wrote:** This paper explores the implications of environmental stress (OA and hypoxia), as determined in lab experiments, on the growth and reproductive potential of mussels in two locations of the Mediterranean with simulations based on a dynamic energy budget model. The study capitalizes on the potential of DEB models to integrate the impacts of multiple environmental drivers on organismal level outcomes, including growth, reproduction, time to maturity, rates of feeding and respiration, and so on. This approach is powerful in potential and the application is new. However, there are some important shortcomings, especially in the way the model is parameterized. Also, I find the lack of some form of quality assessment problematic.
**Author's reply:** First, we thank the Reviewer #1 for the effort in providing his/her suggestions to the original version of our ms.
**Author's changes:** Apart from all possible not clear parts that we accordingly improved, we followed his/her suggestion in the hope to have increased the quality of this ms.

**Reviewer wrote:** It is very annoying that all sections consist of a single paragraph. Did something go wrong with the formatting of the manuscript?
**Author's reply:** We appreciated the referee's suggestion about the different paragraphs, but we believe that the current structure is already sufficiently sectioned; to increase the number of paragraphs or sub-paragraphs can only increase the text fragmentation which may limit the logical flow of the text.
**Reviewer wrote:** The author's use of respiration measurements as a proxy for DEB maintenance costs is problematic. In the DEB framework, respiration is emphatically not the same as maintenance, but also include energetic overheads, such that of growth. Respiration is a function of the commitment rate in DEB, of which maintenance could be a minor part, depending on size and nutritional status of the animal.
**Author's reply:** We acknowledge that respiration does not include only maintenance, but also include energetic overheads, such as growth. Nevertheless, there is no way - to our knowledge - to measure the different contribution of every energetic components apart from to experimentally measure oxygen consumption as a proxy for metabolism. Also, the proposed approaches measuring indirectly the $[\dot{p}_M]$ values (e.g. van der Veer et al., 2006; Ren and Schiel 2008), are not feasible in the context of the present experimental asset. While this approach is not experimentally feasible when assessing the effect of stressors on the energy budget, the only way to indirectly provide an estimation of the effect of disturbance is through the Jagger et al. (2016) approach which is based on the stress factor "s". Thus, after estimating the effect induced by a treatment on the oxygen consumption, that in the present case study was expressed as a percentage variation, we summed/subtracted the energetic amount due to the effect of a stressor to the species-specific $[\dot{p}_M]$ values of *M. galloprovincialis* then we run our models. However, we thank the referee for highlighting this point whose importance was addressed in the Discussion section of this ms. as we believe that is crucial to increase our understating on how we can mechanistically assess the effect of disturbance on individual performances through the DEB model. All these limitations show how much is important to date to increase the experimental and theoretical research effort in order to unravel this point, which is increasingly crucial to get realistic answers to management questions in a context of environmental change.

**Reviewer wrote:** In addition, oxygen deprived mussels, and possibly mussels enduring stress of hypercapnia, are able to use anaerobic metabolic pathways to fulfil their maintenance requirements. If stress increases maintenance requirements, one would expect respiration rates to increase with increasing stress intensity. However, we see the opposite happen (see Fig 2).

I think this is likely due to the fact that stressed mussels have their shells closed more often than unstressed conspecifics (see Fig. 1), and thus ingest less food. Less food leads to a lower energy reserve buffer and therefore a lower rate at which reserves are committed.

**Author's reply:** As reported in our results and Fig. 2, maintenance requirements in accordance with respiration rates, decreases with stress in agreement with what is reported in the current literature. We are sorry with Reviewer #1 and with all readers as we made a mistake in writing the text commenting the figure 1 (we wrote wrongly "opened" instead of "closed"). Actually, our mussels increased their openness with the increasing stressful conditions. At the present stage, we are not able to provide information about the amount of ingested food under different treatments and then we are not able to infer on the effect of openness degree on energetic performances.

**Author's changes:** Figure was fixed according to both referee's suggestion, and also the text in the paragraph has been rephrased accordingly.

**Reviewer wrote:** I suggest the authors change the maximum assimilation rate parameter of their model based on their behavioural observations and leave the maintenance rate parameter unchanged.

**Author's reply:** We appreciated the referee's suggestion but we prefer to focus on both the assimilation efficiency and the metabolic effect (through pM) as i) the main effect of acidification seem to be exerted on metabolism as widely reported in the current literature, and ii) also to show that our mechanistic DEB approach can be really effective in measuring the multiple stressor's effect on LHs.

**Author's changes:** Thus, we enlarged the discussion on these points to include possible shortcomings deriving from the fact that the stressor's effect on maintenance is not still well-experimentally measurable.

**Reviewer wrote:** The simulations suggest that unstressed mussels only grow to 3 cm in length and do not reproduce in Palermo. This seems implausible. How long do real mussels get in Palermo? Do they reproduce? How sensitive are the simulation results to the particular choices of parameter values? The authors do not reflect at all on the reliability of their assessments, which I find troublesome, especially given the politicized context of the subject matter.

**Author's reply:** Actually, to enlarge the discussion about the magnitude of effects at local level could be not influent for our purposes, although our results are in line with the environmental and trophic conditions reported in section 2.5: "Both sites were chosen as they represent two opposite temperature and food conditions for mussel growth in Italy… etc.". *M. galloprovincialis* in Sicily is observed to be limited by oligo-trophic conditions although it grows in highly trophic-enriched areas such as harbours or under Integrated Multi-Trophic Aquaculture (IMTA) conditions (Sarà et al 2012; 2013b, Giacoletti et al. 2018 in press JEMA) which supports what we gathered in the present ms. through the DEB simulations.

**Reviewer specific comment n. 1**
**Reviewer wrote:** Title. Functional spatial contextualization sounds impressive but I've no clue what it could mean. Also, the manuscript deals with only a single species; the title is too general.

**Author's changes:** We agreed with Reviewer's #1 point and changed the title.

**Reviewer specific comment n. 2**
**Reviewer wrote:** L27-33 Split up sentence.

**Author's changes:** Sentence was splitted up accordingly.

**Reviewer specific comment n. 3**
**Reviewer wrote:** L35 (and elsewhere) Put reference in the proper place of the sentence
**Author's changes:** All references were checked and put in proper spaces.

**Reviewer specific comment n. 4**
**Reviewer wrote:** L40 'lager'?
**Author's changes:** Changed with "larger".

**Reviewer specific comment n. 5**
**Reviewer wrote:** L68-70. This is a strong statement and should be substantiated with references. BTW, the only 2 papers using DEB in a OA context I'm aware of are 10.1111/gcb.12547 and 10.1016/j.jembe.2015.09.016
**Author's changes:** References regarding the effect of OA on functional traits such as feeding and assimilation, and on maintenance costs has been added accordingly.

**Reviewer specific comment n. 6**
**Reviewer wrote:** L72 the DEB [p_M] parameter does not relate to assimilation
**Author's changes:** The sentence was rephrased accordingly.

**Reviewer specific comment n. 7**
**Reviewer wrote:** L83 articulated ! consisted of
**Author's changes:** Changed accordingly.

**Reviewer specific comment n. 8**
**Reviewer wrote:** Section 2.4 contains material that should go in 2.3 (or combine the sections).
**Author's reply:** Section 2.4 refers to assimilation efficiency, while section 2.3 to oxygen consumption measures, so we consider not easy to combine both sections as we may incur in the risk to reduce the readability of this section.

**Reviewer specific comment n. 9**
**Reviewer wrote:** I didn't get how the authors calculate the assimilation efficiency.
**Author's changes:** A detailed explanation on how the assimilation efficiency was estimated, was added with supporting references.

**Reviewer specific comment n. 10**
**Reviewer wrote:** Section 3.1 belongs in the Materials and Methods Section. There is no need for a statistical analysis. Delete Table 2.
**Author's changes:** Table 2 was deleted according to both referee's suggestion and details were moved in the Materials and Methods Section.

**Reviewer specific comment n. 11**
**Reviewer wrote:** Combine Sections 3.2-4. There is no need to duplicate in the text what is already presented in the figures. The percentage of closed valves is simply 100 – percentage opened valves, so don't mention the former. I don't understand why the error measures differ so much, though.
**Author's changes:** We agree not to duplicate in text what is already presented in figures, and we worked to avoid this replication. Following Reviewer's #1 suggestion we also expressed the percentage of closed valves 100 – percentage opened valve. Instead, merging the sections can increase the risk of confusion in the reader as section 3.2 is about behavioural observations, while the other two are about physiological measurements.

**Reviewer specific comment n. 12**
**Reviewer wrote:** Section 2.7 is incomprehensible for people without DEB modeling background. Include a figure and references to overview texts (e.g. Kooijman's book, Nisbet et al JAE, Sousa et al, and/or most recently Jusup et al Physics of Life Reviews 20:1-39).
**Author's changes:** We agree with Reviewer #1 that section 2.7 is difficult for someone without a DEB modelling background, and in order to made it more clear we added the suggested references and rephrased some parts.

**Reviewer specific comment n. 13**
**Reviewer wrote:** L263 addictive ! additive. The way the authors use 'additive' is confusing. Additive refers to impacts that can be summed, like 1+1=2, an unlikely situation with nonlinear models, such as DEB.
**Author's changes:** Rephrased following suggestions to "with a progressive contribution of hypoxia".

**Reviewer specific comment n. 14**
**Reviewer wrote:** What are the initial conditions of the simulation runs?
**Author's reply:** Results of simulation performed with unstressed organism were already reported in Table 5, while model parameters have been reported in Table 1.

**Reviewer specific comment n. 15**
**Reviewer wrote:** What is the rational for the choices for the frequency of events?
**Author's reply:** While we are aware that hypoxia events are more frequently during summertime, we decided to not apply any timing and frequency scheme to simulate hypoxia event's occurrence according to many papers published across the recent literature(Crain et al. 2008 Ecology Letters; Miller et al. 2009 PNAS).

**Reviewer specific comment n. 16**
**Reviewer wrote:** From Table 5 remove data that are already presented in Figure 3. Round off # of eggs to 6.74e6. Units of frequency should be 1/time
**Author's changes:** Table 5 was corrected accordingly.

**Reviewer specific comment n. 17**
**Reviewer wrote:** Figure 3 label y axis 'Change relative to control'
**Author's changes:** Figure was corrected accordingly.

**Reviewer specific comment n. 18**
**Reviewer wrote:** L294 delete 'formally'
**Author's changes:** Deleted.

**Reviewer specific comment n. 19**
**Reviewer wrote:** L295 delete 'compensatory' and change contrast to compensate
**Author's changes:** Rephrased.

**Reviewer specific comment n. 20**
**Reviewer wrote:** L299 suppressed feeding activity

**Author's changes:** Changed.

**Reviewer specific comment n. 21**
**Reviewer wrote:** L304 what is crossing effect?
**Author's changes:** Sentence has been rephrased accordingly.

**Reviewer specific comment n. 22**
**Reviewer wrote:** L306 on! over. 'that' doesn't refer to anything
**Author's changes:** Rephrased.

**Reviewer specific comment n. 23**
**Reviewer wrote:** L333 sustainable and reliable ! practical
**Author's changes:** Changed.

**Reviewer specific comment n. 24**
**Reviewer wrote:** L337 write out TW, TRO and TM
**Author's changes:** Written out accordingly.

**Reviewer specific comment n. 25**
**Reviewer wrote:** The readability of the manuscript would improve if there were fewer references. Remove unnecessary repetitive references.
**Author's changes::** All the references has been checked and unnecessary and repetitive ones have been deleted accordingly.

**Predicting the multiple effects of acidification and hypoxia**

**on *Mytilus galloprovincialis* (Bivalvia, Mollusca) life**

**history traits**

Antonio Giacoletti* and Gianluca Sarà

[revised manuscript text omitted]

Recent insights obtained by the experimental research have shown that OA mainly affects feeding rates (FR), assimilation efficiency (AE) and maintenance cost of marine organisms (Appelhans, 2012; Navarro et al. 2013; Kroeker et al. 2014; Zhang et al., 2015; Jager et al. 2016). Here, we translated the combined effects of hypoxia and hypercapnia on  assimilation and oxygen consumption rates as measured under different treatments into effects on assimilation AE and somatic maintenance cost . Somatic maintenance is a crucial suite  of functional traits used in recent bioenergetics based on the DEB theory that mechanistically can be used to investigate the role played by multiple stressors on LH traits of organisms by using first principles (Sarà et al., 2014). We further documented the effects of those stressors on *M. galloprovincialis* shells through the use of a scanning electron microscope (SEM), and compared the maximum shell breaking load of treated *vs.* control specimens. A behavioural analysis completed the frame concerning the individual's response to both single and combined stressors. Carried out in a context of OA, this exercise comprises a first step in linking the fields of ecomechanics and climate change ecology, which should yield a more mechanistic understanding of how biodiversity will respond to environmental change (*sensu* Buckley et al., 2012).

**2 Materials and methods**

This study  consisted of three steps: 1) laboratory investigation on the effects of pH and hypoxia on functional  (both behavioural and physiological) traits of *Mytilus galloprovincialis*; 2) collection of water temperature data and  (CHL-a) data from two Mediterranean sites (Trieste and Palermo), as a further forcing variable in the DEB model and lastly 3) model running to simulate growth and fitness of *M.*

*galloprovincialis* under stressful conditions by using  DEB parameters estimated  by the activities in the first step.

**2.1 Experimental set-up.** Specimens of *M. galloprovincialis* (45 - 55 mm) were provided by the Ittica Alimentare Soc. Coop. Arl. (Palermo) and transferred within 30 minutes to the laboratory. Mussels were then carefully cleaned and placed in a 300L tank filled with natural seawater at room temperature (18-

20°C), field salinity (37-38 ) and fed *ad libitum* with cultured *Isochrysis galbana* (Sarà et al., 2011).

Mussels were acclimated for two weeks to reduce stress generated by manipulation and transport (Sarà et al., 2013a)  once acclimated, 200 specimens were randomly divided in groups of 25

organisms, transferred to 8 independent rectangular glass tanks of 120L capacity (100 cm long, 30 cm deep, 40

cm wide) and kept in a conditioned room at 21°C for 4 weeks according to common protocol with bivalves (Braby & Somero 2006; Fields et al., 2012; Kittner and Riisgård 2005). Tanks 1 to 4 were filled with sea water and continuously  aerated through air pumps, while Tanks 5 to 8 were not aerated and covered with a plastic film disposed on the water surface, in order to avoid gas-exchanges between air and water. Tanks 1-2 were used as a control (CTRL), while hypercapnia was imposed in Tanks 3-4 (Tr1), hypoxia (2

ppm) in Tanks 5-6 (Tr2), and both factor (pH 7.5 and hypoxia) in Tanks 7-8 (Tr3) (see Table 2). Mussels were acclimated to two different nominal pH treatments: (i) pH 8.0 in Tanks 1-2 (CTRL) and 5-6 (Tr2), corresponding to present average pH at the sampling site; and (ii) pH 7.5 in Tanks 2-3 (Tr1) and 7-8 (Tr3), deviating from present range of natural variability and relevant for 2100 ocean acidification scenarios.

(Melzner et al., 2011; Gazeau et al., 2013). The carbonate system speciation ($p$CO2, $HCO_3^-$, $CO_3^{2-}$, $\Omega$Ca and

$\Omega$Ar) was calculated from $pH_{NBS}$, temperature, salinity and alkalinity ($T_A$ = 2.5 mM; Rivaro et al., 2010) using

CO2SYS (see Table 2; Lewis and Wallace, 1998) with dissociation constants from Dickson & Millero (1987).

The pH was manually controlled 8 times a day by an electronic pH-meter (Cyberscan 510, Eutech Instruments; accuracy = ± 0.01 pH) and gaseous $CO_2$ was injected directly into the aquarium through a commercial ceramic diffusor, when required. Oxygen concentration and temperature were monitored with the same frequency through the PiroScience FirestingO2 oxygen logger equipped with a dedicated temperature sensor. Water movement and recirculation were assured by water pumps. Tanks were siphoned at the end of each  day, removing all the faecal material in order to avoid the accumulation of waste products, and 20% of water was weekly changed with specific pre-conditioned sea water for each treatment.

**2.2 Behavioural observations.** The valve gape of mussels was recorded by means of the two simplest behavioural categories reported in Jørgensen et al. (1988): closed valves and opened valves. Each observation was carried out by an operator with the aim to record changes in the behavioural repertoire of bivalves in response to the exposure to a single stressor (pH or hypoxia) and to both pH and hypoxia, compared to individuals kept in normal environmental conditions. All experiments were conducted at environmental (37-38

) salinity and with well-aerated sea water through a gentle flow (Ameyaw-Akumfi & Naylor, 1987), except for specimens of Tank 5-6 and 7-8, that were not aerated in order to maintain the hypoxia level set through the gaseous nitrogen. Behavioural observations were repeated six times a day at week 1 and 4 of exposure, involving 5 random specimens for each treatment.

**2.3 Oxygen consumption.** The rate of oxygen consumption was determined twice (week 1 and week 4) in a respirometric glass chamber (0.3L) inside a temperature-controlled water bath, in order to  investigate the effects of multiple stressors  on metabolic somatic maintenance costs and to integrate it in the standard DEB model. Volume-specific somatic maintenance costs, as expressed by the $[\dot{p}_M]$ parameter (J cm$^{-3}$ h$^{-1}$), represent the amount of energy needed to fuel basal metabolism ($\dot{p}_M$) scaled with the organisms' volume,  ($[\dot{p}_M] = \dot{p}_M/V$). All determinations were performed at 21°C using filtered seawater with the same pH and oxygen content as that of the respective treatment, stirred with a magnetic stirrer bar beneath a perforated glass plate supporting each individual ( Ezgeta-Balic et al., 2011). The decline in oxygen concentration was measured by a PiroScience FirestingO2 respirometer, capable of four sensor connections. We used a total of n = 64 mussels per week, 16 for each treatment (8 for each tank) acclimated as above, fed *ad libitum* until the day before the experiment. The decline was continuously recorded for at least 1 h, excluding an initial period (~ 10 min) when usually there is a more rapid decline in oxygen caused by a disturbance of the sensor's temperature equilibration. Respiration rate (RR, µmol O$_2$ h$^{-1}$) was calculated according to ( Sarà et al.,  2013b): $RR = (C_{t0} - C_{t1}) \, x \, Vol_r \, x \, 60(t_1 - t_0)^{-1}$, where $C_{t0}$ is oxygen concentration at the beginning of the measurement, $C_{t1}$ is the oxygen concentration at the end of the measurement, and $Vol_r$ is the volume of water in the respirometric chamber. Volume-specific somatic maintenance costs were then calculated by converting oxygen consumption rates expressed in µmol h$^{-1}$ in J h$^{-1}$ through a conversion factor (Kooijman 2010) and then in J cm$^{-3}$ (van der Veer et al., 2006; Ren and Schiel 2008) (for the calculation of dry weights refer to the end of section 2.4).

**2.4 Assimilation efficiency.**  Assimilation efficiency (AE) was measured through the Conover ratio (1966) $AE = (F - E)/[(1 - E)F]$, where F is the ratio between ash-free dry weight (AFDW) and dry weight (DW) for food, and E is the same ratio for the faeces; this represents the efficiency with which organic material is absorbed from the ingested food material. Here, after oxygen consumption measurement, the same  specimens  were  placed into separate beakers containing 1L of filtered seawater (specific for each treatment) and a magnetic stirrer bar. In order to allow the mussels to open their valves and start their filtration activity, they were given 15 minutes before the introduction of food with an initial concentration of ~ 15,000 *Isochrysis galbana* cells ml$^{-1}$. After a period of 2 h mussels were moved to cleaned 1L glass beakers with filtered seawater for a period of 12 h, after that the water contained in each beaker was filtered on pre-ashed and weighted GF/C fibreglass filters. Once filtered, filters were washed with 0.5 M ammonium formate (purest grade) to remove adventitious salts (Sarà et al., 2013a), dried in the oven (95°C for 24 h) and then incinerated in a muffle furnace (450°C for 4 h). After each step, the samples were weighted using a balance (Sartorius BL 120S ± 1µg). For the calculation of AE, together with the faeces collected from the mussels, filters containing algal food were dried and incinerated as above. After respirometric measurement and the collection of faeces each animal was killed by gentle freezing and dissected, and the shells were separated from the body tissue  to calculate the condition index according to Davenport & Chen (1987) (CI = (body weight/shell weight) × 100), and their individual dry weights  to standardize respiration rates.

**2.5 Water temperature data.** The main forcing driver of shellfish LH inside DEB models is represented by  seawater temperature (Pouvreau et al., 2006; Kearney et al., 2010; Kooijman, 2010; Sarà et al., 2011; 2013). DEB simulations were run under subtidal conditions (body temperature was expressed by the mean seawater temperature; Montalto et al., 2014) with 4 years-hourly data (Jan 2006 - Dec 2009) of seawater temperature measured about 1 m below the surface at the closest meteo-oceanographic station held in Trieste (LAT 45° 38′ 57.81″; LONG 13° 45′ 28.58″) and Palermo (LAT 38° 07′ 17.08″; LONG 13° 22′ 16.79″). The period of 4 years is consistent with the normal life span of most Mediterranean shellfishes (Sarà et al., 2012; 2013b). Both sites were chosen as they represent two opposite temperature and food conditions for mussel growth in Italy, with Trieste as representative of lower temperature (average 16.98 ± 6.19 °C) and higher food levels (average 1.36 ± 0.37 CHL-a), and Palermo of higher temperatures (average 20.19 ± 4.64 °C) and lower food (average 0.19 ± 0.09 CHL-a). Data are available online from the Italian Institute of Environmental Research (ISPRA) web page (http://www.mareografico.it/).

**2.6 CHL-a dataset.**  (CHL-a) was derived from satellite imageries (µg L$^{-1}$; http://emis.jrc.ec.europa.eu/))  and adopted as a reliable food quantifier for suspension feeders (Kearney et al., 2010; Sarà et al., 2011; 2012)

**2.7 Model description.** The Dynamic Energy Budget (DEB) Theory provides a general framework that allows to describe how physiological mechanisms are driven by temperature and food availability, and influences growth and the reproductive performances in marine organisms (Sousa et al., 2010; Monaco et al., 2014; Jusup et al., 2017). Following the κ-rule ( Kooijman, 2010) a fixed energy fraction (κ) is allocated to growth and somatic maintenance, while the remaining fraction (1-κ) is allocated to maturity maintenance plus maturation or reproduction. If the general environmental condition deviate from common natural patterns (i.e.

changes in temperature, food availability etc.) reproduction and growth are consequently affected. According to the DEB theory, a reduction in growth can be caused either by reduced food assimilation ($\dot{p}_A$), enhanced maintenance costs $[\dot{p}_M]$, or enhanced growth costs ($\dot{p}_G$). Using this approach, and through the DEB

parameters reported in Table 1, except for the variation in the maintenance costs

$[\dot{p}_M]$ and in the assimilation efficiency of food (AE) which were experimentally estimated throughout this study, we performed simulations using theStandard version of the DEB model (Nisbet et al., 2010) aimed at investigating the potential variations in growth and fecundity of our model species. To run the DEB simulations, local thermal series of selected sites were used together with satellite CHL-a concentrations, obtaining a first model with environmental conditions. A second model was run with the $[\dot{p}_M]$$\dot{p}_M$ calculated from the oxygen measurements on specimens of *M. galloprovincialis* from Tanks 3-4 (pH 7.5) simulating a chronic hypercapnia condition for the full cycle (4 years) and the relative estimated AE. Subsequently, further models were run by simulating one random hypoxia event (duration = 30 days) for each of the four years of the cycle, then simulating two yearly events, and so on up to six  hypoxia events. The starting month of each event was randomly chosen for every year with the use of a table of random digits. The $[\dot{p}_M]$$\dot{p}_M$ calculated from the oxygen consumption rate measurements on specimens from Tanks 7-8 (pH 7.5 and hypoxia) was used in substitution to

$[\dot{p}_M]$$\dot{p}_M$ from pH 7.5 tanks 3-4, coupled with the relative estimated AE, when simulating both stressors.

Simulations were performed using the R routine for Standard DEB model developed by M. Kearney (2012), and further modified (for use in bivalve modelling) by Sarà et al. (2013). Outputs of the DEB models were: the maximum theoretical total length of shellfish (TL), the maximum total weight (TW), the total number of eggs (TRO) produced during a life-span of 4 years, the total number of reproductive events (RE) and the time needed to reach gonadic maturity (TM) for each treatment.

**2.7.1 Model limitation.** DEB models are particularly useful to quantitatively assess the effects of multiple stressors on LH-traits in an integrated manner, leading to test the hypothesis on how OA may affect the maintenance costs of living organisms (Jager et al., 2016). Maintenance costs, as defined by Dynamic Energy

Budget Theory (Kooijman, 2010), represent the energy requirement of an organism to survive, excluding investments in growth, reproduction and development. The volume-specific somatic maintenance costs parameter [$\dot{p}_M$] within the standard DEB model has been up to date estimated only by indirect approaches through changes in energy content by starvation over time (van der Meer, 2006) or measurements of respiration rate of starved organisms (van der Veer et al., 2006; Ren and Schiel 2008). The idea of quantitatively assess the effect of a stressor including it as a modification of a specific parameter was first introduced by Jager et al.

(2016) with the *stress factor* "s" applied to assimilation, maintenance and cost of growth. Thus, after estimating the effect induced by a treatment on the oxygen consumption, in our case expressed as percentage variation, we summed/subtracted the energetic amount due to the effect of a stressor to the species-specific [$\dot{p}M$] parameter of

*M. galloprovincialis* (Sarà et al. 2012) then we run our models. Previous proposed approaches, taking into account starvation for [$\dot{p}_M$] estimation, wouldn't be realistically applicable for testing and quantifying the effect of a stressor on the energy budget, without adding a further stressor. Jager et al. (2016) was therefore the first to adopt this concept, although using a simplified DEB model (DEBkiss; Jager et al. 2013) that did not involve the concepts of reserve and maturity that play a central role in DEB theory. Although this may not be considered a reliable measure of maintenance costs but a simpler proxy of metabolic effect, negligible costs for growth and gonadic development stand on the assumption of constant protein turnover throughout the experimental range (Hawkins et al. 1989).

**2.8 Effects on shell: mechanical strength and SEM pictures.** The functional impact of exposure to pH and to validate the pH effect on morphological structure of valves, was tested on mussels exposed to the two nominal pHs for 4 weeks. Twice (week 1 and week 4), 16 mussels for each treatment were collected and dissected, and both valves were cleaned and dried with absorbent paper. The left valve was then sliced transversely using a circular saw (Dremel® 300 series) to section the whole length of the shell. Age was estimated using the analysis of shell rings proposed by Peharda et al. (2011) by counting the number of rings with the use of a stereo microscope (Leica EZ4). The right valves were instead evaluated for their breaking properties through crushing tests (maximum breaking load in N)  as previously done in Martinez et al. (2018)

. The effect of low pH exposure  was also documented by the use of a scanning electron microscope (SEM; Zeiss LEO 440) that led to a thorough investigation on the integrity of the mussels' external protein layer (*periostracum*) and on the underlying mineral layer, rich in calcite and aragonite.

**2.9 Statistical analysis.** The assumption of normal distribution has been tested through the Anderson–Darling test using Past® software. In order to test for significant differences in respiration rate and the assimilation efficiency, ANOVAs were performed using Treatment (CTRL, Tr1, Tr2, Tr3) and Time (Week 1 and Week 4) as fixed factors, with respectively four and two levels. In order to test for significant differences in behavioural categories, ANOVAs were performed using Treatment (CTRL, Tr1, Tr2, Tr3) as fixed factors, while Breaking load was tested with Treatment (CTRL, Tr1, Tr2, Tr3) and Time (Week 1 and Week 4) as fixed factors. When significant differences were detected, the Student-Newman-Keuls (SNK) post-hoc pair wise comparison of means was used (Underwood, 1997). Cochran's test was used prior to ANOVA to test the assumption of homogeneity of variance (Underwood, 1997). When no homogeneous variances were rendered with any type of transformation, the significance level was set at 0.01 instead of 0.05, as ANOVA can withstand variance heterogeneity, particularly in large balanced experiments, thereby reducing the possibility of a Type I error (Underwood, 1997).

**3 Results**

**3. 1 Valve gaping.** During behavioural observations on *M. galloprovincialis*, specimens showed a significant difference in the behavioural categories, showing respectively

33.3 ± 11.2 (CTRL), 50 ±4.5 (Tr1), 80 ± 8.9 (Tr2) and 83.3 ± 6.1 of opened valves (Fig. 1; Table 3,

ANOVA, p < 0.001). The percentage of closed valves can be easily calculated as 100 – open valves. No significant differences resulted between week 1 and 4 (ANOVA, p > 0.05), between CTRL and Tr1 and between

Tr2 and Tr3.

[revised manuscript text omitted]

Fig. 4 Percentage effect of Hypoxia from DEB simulations on TL and TW considering the two different trophic conditions represented by Trieste (a) and Palermo (b).

**3.6 Effects on shell.**  Results from the breaking load experiment revealed a significant effect of pH (58.8 ± 5 N) and of combined stressors on the breaking load (50 ± 2.7 N), compared to hypoxic (64.4 ± 3.7 N) and CTRL specimens (77.2 ± 2.2 N) (Fig. 5) (Table 3, ANOVA, p < 0.001). In addition, the effect was stronger at week 4 than after one week of exposure (Table 3, ANOVA, p < 0.01). Deeper investigations through scanning electron microscopy validated an effect by showing an increasing erosion of the shell after exposure to $CO_2$-induced acidification. The external dissolution pattern usually started from the umbonal region and progressed toward the margin of the shell, usually associated with some degree of damage to the *periostracum*. The damage was present at differing extensions in all specimens exposed to treatments, except in the control mussels (Fig. 6 b, c, d). The alteration of the underlying carbonate layer was instead visible only in Tr1 and Tr3, with details in Fig. 7 (b, d). This kind of alteration was never  observed under control pH (Fig. 4a).

[Figure]

Fig. 4 5 Breaking load of valves (in Newton, N ± se) exposed to different treatments of oxygen (normoxia –

hypoxia 2ppm) and pH (7.5 – 8.0) at week 1 and 4.

[Figure]

Fig. 5 6 SEM pictures of different shells exposed to (a) control condition (CTRL); (b) pH 7.5 and normoxia condition (Tr1); (c) normal pH and hypoxia condition (Tr2); (d) both pH 7.5 and hypoxia conditions (Tr3).

[Figure]

Fig. 6 7 Details of different shells exposed to (a) control condition (CTRL); (b) pH 7.5 and normoxia condition (Tr1); (c) normal pH and hypoxia condition (Tr2); (d) both pH 7.5 and hypoxia conditions (Tr3).

**4 Discussion**

Marine organisms, and in particular intertidal species (Montecinos et al., 2009), have been  recognized as being equipped with well-developed and conserved  mechanisms to  compensate ocean acidification such as (i) passive buffering of intra- and extracellular fluids; (ii) transport and exchange of relevant ions; (iii) transport of $CO_2$ in the blood in those species that have respiratory pigments; (iv) metabolic suppression to wait out periods of elevated $CO_2$ (e.g. Lindinger et al., 1984; Cameron, 1989; Walsh and Milligan, 1989; Hand,

1991; Heisler, 1993; Guppy and Withers, 1999; Pörtner et al., 2004). Several authors recorded suppressed of feeding activity and growth, depressed metabolism, increased N excretion and loss of tissue weight for marine bivalves exposed to reduced seawater pH (Bamber, 1990; Michaelidis et al., 2005; Berge et al., 2006; Gazeau et al., 2010). Bivalves are in fact capable of maintaining a constant internal pH by decreasing their metabolic rates and/or dissolving their shell; the shell acting then as a source of $CO_3^{2-}$ (Bamber, 1990; Michaelidis et al., 2005;

Berge et al., 2006) counterbalancing the  effect of dissolved $CO_2$ crossing biological membranes . Compensation of low pH, associated with anthropogenic increases in seawater pCO$_2$ (Fabry et al., 2008), through adjustments in ionic composition appears to be a trade-off that is not likely sustainable  over longer time-scales., pCO₂ (Fabry et al., 2008) From our behavioural observations mussels exposed to low pH resulted in a higher, even if not significant, percentage of opened valves respect to CTRL individuals, with the highest significant difference relative to hypoxia exposition. The effect of low pH on the adductor muscle of bivalves has been already documented by Pynnönen & Huebner (1995) and the same effect has been reported after exposition to hypoxia (Sheldon & Walker 1989). In agreement with current literature showing deleterious effects of $CO_2$-induced acidification on a wide range of invertebrates (Barnhart & McMahon, 1988; Barnhart, 1989; Rees & Hand, 1990), and similarly to other studies by *M. galloprovincialis* (Gestoso et al., 2016; Michaelidis et al., 2005), our results showed how hypercapnia (pH reduced by 0.6 units, relative to the natural pH of the lower Tyrrhenian waters) was able to induce a decline in metabolic rates of mussels. This kind of decline has already been noticed by other authors as an adaptive strategy for survival under transiently stressful conditions (Michaelidis et al., 2005).

According to Pörtner et al. (2004), metabolic reduction due to hypercapnia could be a result of acid-base disturbances and therefore be similar to the response of intertidal individuals to anaerobic conditions. Direct effects of hypoxemia have been further proven to cause fatal decrements in an organism's performance in growth, reproduction, feeding, immunity and behaviour (*sensu* Pörtner & Farrell, 2008). Synergistic Combined effects by sstressors like such as ocean acidification and hypoxia narrow are capable of narrowing the thermal window of functioning according to species-specific sensitivities, modulating biogeographical distributionses, coexistence ranges, community shifts and in general ecologicalother interactions (Pörtner & Farrell, 2008). The mussel *Mytilus*

*edulis* has been proven able to compensate both short- and long-term exposure to hypercapnia by dissolution of its shells (Lindinger et al., 1984; Michaelidis et al., 2005), resulting in reduced growth and metabolism. A similar mechanism of release of inorganic molecules into the pallial cavity (as $CaCO_3$ from valves) has been documented during periods of anaerobic metabolism, to maintain the acid-base balance (Chaparro et al., 2009), determining further physiological and energetic cost such as decreased growth, respiration rate and protein synthesis (Pörtner et al., 2005). During periods of environmental oxygen limitation, many organisms are able to suppress ATP demand, shut down expensive processes, such as protein synthesis (Hand, 1991), but at the same time limiting growth and the reproductive potential. Although suppression of metabolism under short-term experimental conditions is a

"sub-lethal" reversible process, reductions in growth and reproductive output will effectively diminish the survival of the species on longer time-scales (Fabry et al., 2008). The contemporary occurrence in our simulations, of monthly hypoxia events, revealed a growing additive contribution to what was already elicited by hypercapnia alone on growth and reproduction. Current literature has not currently explored the combined effects of multiple stressors on long-term experiments by modulating the intensity and duration of disturbance. This would probably translate as a very complex experimental set-up which would be hardly practicable, especially on long-term scales.

On the other hand, mechanistic models offer a more practical alternative to long-term, in- field research when studying the effects of multiple-stressors, with the advantage of testing, at the same time, the magnitude and the duration of disturbance on LH _traits of a model species. Our results highlighted the general hypoxia growing effect following the increasing duration of disturbance, with a particular focus in Trieste on TW

and TRO, while in Palermo on TW and TM (Fig. 3). A further important peculiarity of the mechanistic modelling  deals with the possibility to spatially contextualise the effects of single and multiple stressors on selected outputs by integrating local thermal conditions and food concentrations (Sarà et al.

2018c). Comparing the effect of hypoxia across frequencies (Fig. 4), total length (TL) resulted unaffected up to a frequency of 2 hypoxia events in both sites, then the highest effect was recorded in the eutrophic site (Trieste).

Trieste, between the two chosen sites, had  the lowest temperature. On the contrary a smaller effect of hypoxia was detected  on  in

Trieste, suggesting  a sort of food compensation capacity on the effect of environmental stressor (Mackenzie et al., 2014). Also the DEB model easily allowed the estimation of the fecundity potential of  organisms, that is often omitted in other ecological studies, but that represents a crucial quantity for resource (e.g. aquaculture) (Sarà et al. 2018c) and conservation purposes. To verify impacts on shellfish fecundity, we contextualised our simulation by introducing Trieste hourly temperature series after those of Palermo, with the respective local actual CHL-a concentrations, as long as in the first site no reproductive events came out from our simulations, probably due to food limitations and temperature threshold. This is reflected by natural populations in Palermo colonising  only substrates in  trophic- enriched sites. A combined effect of the simultaneous stressors, such as those considered across this study, has proven in the present study to affect the organism's performance in growth, reproduction and behaviour. Our results highlighted an effect of pH alone and when combined with hypoxia on the breaking load of shells of our experimental mussels. Through a similar approach, Martinez et al. (2018) showed that temperature was a primary factor driving shell's fragility along a latitudinal gradient. Present findings corroborates that idea that fragility can be affected by both stressors through a combined effect. Multiple stressors can narrow, especially  when organisms are on the edge of their thermal tolerance range, the thermal window and this has a potential for generating repercussions on  biogeographical distribution, coexistence ranges, community shifts, food webs and species interactions (*sensu* Pörtner & Farrell,

2008). Moreover, an appropriate knowledge of species' biological traits, and a mechanistic understanding of the effect of each stressor, reached through an FT-based approach, will allow the translation of the effects of environmental change into realistic management measures taking into account the optimisation of the species'

biological traits (Sarà et al. 2018a,b).

6 Conclusions

Additional research is still required to improve our knowledge of organismal response to multiple stressors, in particular, of many marine ectotherms with indeterminate growth amongst invertebrates (e.g. crustaceans, molluscs). Nevertheless, modelling the growth and reproductive potential (and failure) of species vulnerable to those stressors with predictive tools, such as bioenergetic models is a useful approach for management and protection purposes, but also for shellfish culture in general.

**Authors' contributions**
Both authors contributed to all phases of this ms. AG and GS conceived the idea and led the writing. AG carried
out all experiments in mesocosms, performed modelling work and analysed data. GS provided lab facilities and
research funds. All authors contributed critically to the drafts and gave final approval for publication.

**Acknowledgements**
PRIN TETRIS 2010 grant n. 2010PBMAXP_003, funded to GSARA by the Italian Minister of Research and
University (MIUR) supported this research. The authors declare that they have no conflict of interest. We thank
and are especially grateful to all collaborators involved in this paper, in particular to Dr. Alessandro Rinaldi,
Matteo Mercurio and Marco Martinez for their technical support. We also thank Francesco Furnari for the use of
the scanning electron microscope. We deeply thanks and Ms. Jan Underwood for the fine-tuning of the English.

**Tables**

Table 1. DEB parameters for *Mytilus galloprovincialis* (1 = Kooijman, 2010; 2= van der Veer et al., 2006; 3 = Sarà et al., 2011; 4 = Thomas et al., 2006; 5 = Schneider, 2008); Lb, Lp, Ls = length at birth, puberty and seeding respectively; dVw = specific density to convert weight into volume; $f$ = functional response type II f = X/(XK + X); μx = chemical potential to convert moles into food energy; SMI is the somatic mass index of both starved and well-fed animals, expressed as somatic ash free dry mass (AFDM, mg); $X_K$ = saturation coefficient expressed as a concentration of chlorophyll a (μg CHL-a l$^{-1}$), where the ingestion rate is half of the maximum.

| Symbol | Description | Formulation | Units | *Mytilus galloprovincialis* Value | Ref |
|--------|-------------|-------------|-------|--------|-----|
| $V_b$ | Structural volume at birth | $V_b = (L_b \times \delta_m)^3$ | cm$^3$ | 0.0000013 | 1 |
| $V_s$ | Structural volume at seeding | $V_s = (L_s \times \delta_m)^3$ | cm$^3$ | - | - |
| $V_p$ | Structural volume at puberty | $V_p = (L_p \times \delta_m)^3$ | cm$^3$ | 0.06 | 2 |
| $\delta_m$ | Shape coefficient | $\delta_m = (Ww \times d_{Vw}^{-1}) \times L^{-1}$ | - | 0.2254 | 3 |
| $\{J_{Xm}\}$ | Maximum surface area-specific ingestion rate | $\{J_{Xm}\} = J_X /(f \times V^{\frac{2}{3}})$ | J cm$^{-2}$h$^{-1}$ | 8.2 | 4 |
| ae | Assimilation efficiency | ae = $(\mu_x \times J_X)/p_A$ | - | 0.88 | 3 |
| $X_K$ | Saturation coefficient | - | μg l$^{-1}$ | 2.1 | 3 |
| $[E_G]$ | Volume-specific cost of growth | $[E_G] = SMI_{starved} \times 23 \times (\delta^3_m)^{-1}$ | J cm$^3$ | 5,993 | 5 |
| $[E_m]$ | Maximum storage density | $[E_m] = (SMI_{fed} - SMI_{starved}) \times 23 \times (\delta^3_m)^{-1}$ | J cm$^3$ | 2,190 | 2 |
| $[p_M]$ | Volume-specific maintenance cost | $[p_M] = p_M/V$ | J cm$^{-3}$ h$^{-1}$ | 1 | 2 |
| κ | Fraction of utilized energy spent on maintenance and growth | - | - | 0.7 | 2 |
| $K_R$ | Fraction of reproductive energy | - | - | 0.8 | 3 |
| $T_A$ | Arrhenius temperature | $T_A = \ln|K_{(T_0)}/K_{(T_1)}|x\frac{(T_1 x T_0)}{(T_0 - T_1)}$ | °K | 7,022 | 2 |
| $T_L$ | Lower boundary of tolerance range | | °K | 275 | 2 |
| $T_H$ | Upper boundary of tolerance range | | °K | 296 | 2 |
| $T_{AL}$ | Rate of decrease at lower boundary | | °K | 45,430 | 2 |
| $T_{AH}$ | Rate of decrease at upper Boundary | | °K | 31,376 | 2 |

Table 2. Seawater carbonate chemistry parameters (mean ± se). Seawater pH on the NBS scale (pH$_{NBS}$), temperature (T; °C), and salinity were used to calculate $CO_2$ partial pressure ($p$CO$_2$; μatm) as well as aragonite and calcite saturation states (respectively Ω$_{ar}$ and Ω$_{ca}$), for a total alkalinity of 2500 mmol kg$^{-1}$.

| | Measured | Calculated |
|--|----------|------------|
| | | |

| | Temperature (°C) | pH$_{NBS}$ | O$_2$ mg/l | Salinity (PSU) | pCO$_2$ (µatm) | CO$_3^-$ | Ωca | Ωar |
|---|---|---|---|---|---|---|---|---|
| CTRL | 20.77 ± 0.01 | 8.01±0.001 | 7.29±0.02 | 37.18±0.11 | 624.31±4.9 | 167.93±0.95 | 3.95±0.02 | 2.58±0.01 |
| Tr1 | 20.77 ± 0.01 | 7.53±0.002 | 7.30±0.02 | 37.12±0.05 | 2151.17±22.02 | 62.05±0.73 | 1.46±0.02 | 0.95±0.01 |
| Tr2 | 20.77 ± 0.01 | 8.01±0.001 | 2.44±0.02 | 37.07±0.04 | 729.88±18.24 | 152.53±1.51 | 3.59±0.04 | 2.34±0.02 |
| Tr3 | 20.77 ± 0.01 | 7.53±0.002 | 2.44±0.02 | 37.21±0.17 | 2238.83±20.72 | 59.59±0.42 | 1.40±0.01 | 0.91±0.01 |

Table 2. ANOVA on seawater chemistry parameters. Comparison between CTRL (normal pH) and TREAT (low pH and hypoxia) (* = p < 0.05; ** = p < 0.01; *** = p < 0.001; ns = not significant).

| | | pH$_{NBS}$ | | | pO$_2$ | | |
|---|---|---|---|---|---|---|---|
| | df | MS | F | p | MS | F | p |
| TREAT | 3 | 10.73 | 41450.84 | ** | 1083.21 | 18798.36 | ** |
| Residuals | 548 | 0.0003 | | | 0.06 | | |
| Cochran's Snk | | | | * | | | * |

| | | pCO$_2$ | | | CO$_3^-$ | | |
|---|---|---|---|---|---|---|---|
| | df | MS | F | p | MS | F | p |
| TREAT | 3 | 1.06e08 | 2426.84 | ** | 460157.7 | 3433.17 | ** |
| Residuals | 548 | 43851.09 | | | 134.03 | | |
| Cochran's Snk | | | | * | | | * |

| | | Ωca | | | Ωar | | |
|---|---|---|---|---|---|---|---|
| | df | MS | F | p | MS | F | p |
| TREAT | 3 | 254.09 | 3432.44 | ** | 108.26 | 3426.14 | ** |
| Residuals | 548 | 0.07 | | | 0.03 | | |
| Cochran's Snk | | | | * | | | * |

Table 3  ANOVA table of results. Effect on valve gape and breaking load of *Mytilus galloprovincialis* (* = p < 0.05; ** = p < 0.01; *** = p < 0.001; ns = not significant).

| Source | df | Valve gape | | | Source | df | Breaking load | | |
|---|---|---|---|---|---|---|---|---|---|
| | | MS | F | P | | | MS | F | P |

| Source | df | MS | F | P | Source | df | MS | F | P |
|---|---|---|---|---|---|---|---|---|---|
| Treatment (Tr) | 3 | 17.41 | 15.60 | *** | Treatment (Tr) | 3 | 3838.12 | 15.18 | *** |
| Time (Ti) | 84 | 2.08 | 1.87 | ns | Time (Ti) | 1 | 777.19 | 9.22 | ** |
| Tr x Ti | | 0.3056 | 0.27 | ns | Tr x Ti | 3 | 132.92 | 1.58 | ns |
| Residuals | 40 | | | | Residuals | 56 | | | |
| Cochran's C | | | | ns | Cochran's C | | | | ns |

Table 4 ANOVA table of results. Respiration rate (RR) and assimilation efficiency (AE) of *Mytilus galloprovincialis* (* = $p < 0.05$; ** = $p < 0.01$; *** = $p < 0.001$; ns = not significant).

| Source | df | RR st | | | AE | | |
|---|---|---|---|---|---|---|---|
| | | MS | F | P | MS | F | P |
| Treatment (Tr) | 3 | 312.9183 | 6.95 | *** | 0.2783 | 12.21 | *** |
| Time (Ti) | 1 | 205.1325 | 4.56 | * | 0.0424 | 1.86 | ns |
| Tr x Ti | 3 | 40.7752 | 0.91 | ns | 0.0198 | 0.87 | ns |
| Residuals | 120 | 45.0271 | | | 0.0228 | | |
| Cochran's C | | | | * | | | ns |

Table 5 DEB simulation outputs. Percentage variation of treatments from CTRL: Total length (TL), Total weight (WW), Total reproductive output (TRO), Total reproductive events (RE), Time to maturity (TM).

| DEB outputs (CTRL) after 4 years | | | | | | | | |
|---|---|---|---|---|---|---|---|---|
| Site | Stressor | Hypoxia events (days) | Frequency (1/Time) | TL (cm) | WW (g) | TRO (n° egg) | RE | TM (days) |
| Trieste | CTRL | 0 | 0 | 9.55 | 11.19 | 6.74e6  | 9 | 232 |
| Palermo | CTRL | 0 | 0 | 3.08 | 0.31 | 0 | 0 | 739 |

|  | | | | | | | |
|---|---|---|---|---|---|---|---|
|  |  |  |  |  |  |  |  |
|  |  |  |  |  |  |  |  |
|  |  |  |  |  |  |  |  |
|  |  |  |  |  |  |  |  |
|  |  |  |  |  |  |  |  |
|  |  |  |  |  |  |  |  |
|  |  |  |  |  |  |  |  |
|  |  |  |  |  |  |  |  |
|  |  |  |  |  |  |  |  |
|  |  |  |  |  |  |  |  |
|  |  |  |  |  |  |  |  |
|  |  |  |  |  |  |  |  |
|  |  |  |  |  |  |  |  |
|  |  |  |  |  |  |  |  |
|  |  |  |  |  |  |  |  |

| Percentage  contributing effect of Hypoxia | | | | | | | | |
|---|---|---|---|---|---|---|---|---|
| Trieste | pH+hypoxia | 30 | 0.08  | -0.6 | -1.4 | -1.1 | 0 | 0.8 |
| Trieste | pH+hypoxia | 60 | 0.17  | -1.3 | -2.8 | -2.8 | 0 | 2.1 |
| Trieste | pH+hypoxia | 90 | 0.25  | -2 | -4.3 | -4.5 | 0 | 3.6 |
| Trieste | pH+hypoxia | 120 | 0.33  | -2.7 | -5.8 | -6 | 0 | 4.4 |
| Trieste | pH+hypoxia | 150 | 0.42  | -3.4 | -7.2 | -7.4 | 0 | 5.5 |
| Trieste | pH+hypoxia | 180 | 0.50  | -4 | -8.4 | -8.8 | 0 | 6 |
| Palermo | pH+hypoxia | 30 | 0.08  | -0.6 | -1.6 | 0 | 0 | 0.8 |
| Palermo | pH+hypoxia | 60 | 0.17  | -1.3 | -3.3 | 0 | 0 | 2.1 |
| Palermo | pH+hypoxia | 90 | 0.25  | -1.9 | -4.6 | 0 | 0 | 3.8 |
| Palermo | pH+hypoxia | 120 | 0.33  | -2.6 | -6.1 | 0 | 0 | 4.5 |
| Palermo | pH+hypoxia | 150 | 0.42  | -3.2 | -7.4 | 0 | 0 | 5.7 |

| | | | | | | | | |
|---|---|---|---|---|---|---|---|---|
| | | | month | | | | | |
| Palermo | pH+hypoxia | 180 | 0.506 month | -3.9 | -8.9 | 0 | 0 | 7.6 |

---

## Author Comment (AC3) · 2 Apr 2018

Sorry we wrongly sent the reply for REF 1. There's the reply for you:

REVIEWER #2 Reviewer wrote: This study integrates laboratory-derived parameters of mussel metabolism and assimilation efficiency to run DEB models testing the effect of pH and hypoxia, using environ- mental data input (temperature, food) from two sites within the mussel's biogeographic range. I appreciate the approach of introducing hypoxia events (although I have a comment on the design these events) as a means of incorporating environmental variability in the model. This literature is sparse with such perspectives, especially in the context of multiple stressors. However, the paper lacks a perspective of the environmental relevance of the experimental design and modelling. Author's reply: We thank the Reviewer #2 for helping us improving the readability and the clearness of the ms. Author's changes: In doing so we applied most of the suggestion for the highlighted points, and we discussed them through the specific comments. A perspective of environmental relevance on modelling outputs has been currently added in the discussion section following what suggested by the reviewer.

Major comments Methods need much more detail (see detailed comments) Reviewer wrote: Physiological condition of experimental mussels during the experiment is not quantified. Feeding was ad libitum and it was not assessed if mussels were being fed at conditions of either site used for the modeling. This is problematic as, for example, if the mussels are starving relative to their natural food supply, the derived experimental parameters for the DEB model may be inappropriate. The authors also do not explain the experimental design. Why was the experiment 4 weeks? Author's reply: We thank the reviewer for highlighting this point. Author's changes: We estimated the condition index through the biometric available data and compared it through the experiment, resulting in a not significant variation throughout the study period, and this supports that experimental animals were not stressed by starvation. Further, we used the locution "ad libitum" to indicate a food concentration saturating the feeding processes of animals over time. Such an experimental maintenance condition is commonly used throughout the current literature when bivalves are maintained with ad libitum condition of food in bioenergetic experiments (e.g. Sarà et al., 2013; Montalto et al., 2014; Tagliarolo et al., 2016). However, we adopted a four weeks-period to estimate the effect of OA on functional traits of mussels; such a period is judged to be enough to allow mussels to acclimate to new conditions, as showed in many experimental papers across the current literature(references added in the manuscript).

Reviewer wrote: DEB models: As a reader of Biogeosciences, but not an expert on DEB models, it would be helpful if the authors reviewed this approach more clearly (perhaps using a schematic, what program is used to run models, table of input variables etc.). After reading the paper, I am unclear about the exact implementation and conclusions that can be drawn from this simulation based on the following 4 sources of confusion: 1)From what I understand, temperature from 2006-2009 is used as one of the model inputs. However, all the biological parameters taken from the experiments come from 21 degr. C (although this is not stated explicitly for respiration, but I assume it's 21 C). Since environmental data would vary in terms of temperature across the four years, I don't understand how biological performance is scaled across this temperature regime. It would be good to include a figure of the environmental data (means are not great time-series descriptors, especially for biological processes that are seasonal, such as reproduction), as well as a figure on how the biological parameters were scaled for temperature effects over the years. This same argument applies to food concentration (which I assume varies by time of year as well). Author's reply: The 2006-2009 thermal series has been used as a forcing variables inside DEB models in current modelling literature (Sarà et al. 2011, 2013b; Montalto et al. 2014). The most important factors driving changes in energy budgets of ectotherms are body temperature, on which every metabolic rates depends via the Arrhenius relationships (Kooijman 2010), and food. Arrhenius temperature, that is species specific, acts as a correction factors inside DEB models to scale all rates to environmental temperature. At the same time DEB models use the 2006-2009 CHL-a series as a second forcing variables to predict LH-traits of our species. Accordingly, including a new figure of environmental data could make the ms. heavier also as the main object is not to contextualise the effect at that period, but to show that stressor's effect is simulated across a long integrated period.

Reviewer wrote: 2) The authors use hypoxia and acidification, two future stressors, with temperature data from a few years ago. This design ignores the fact that warming is currently the dominant stressors for this species in the Mediterranean Sea and is expected to continue in the future. As the environmental relevance of the study design is not discussed, as written, the results do not match any realistic environmental situation. This counters the original intent of using DEB models to better "predict organismal functional traits, capturing variation across species to solve a very wide range of problems in ecology and evolutionary biology" (L58). Author's reply: We did not test the effect of increasing temperature as we are pretty sure that the thermal effect is not manifested on a period so short (only 4-years); however other companion papers (e.g. Montalto et al. 2017) tested the effect of increasing temperature on mussel's performances throughout the whole Basin. To extrapolate the effect only two stressors, we carried out simulations under two different latitudinal temperature patterns (Trieste, north Adriatic and Palermo, Southern MED). Anyway, we included some discussion lines about this issue.

Reviewer wrote: 3) Given that reproduction of this species can be quite seasonal, how does the DEB model handle this in terms of estimating reproductive output? Author's reply: DEB manages the seasonal reproduction throughout thresholds based on the temperature-energy relationships.

Reviewer wrote: 4) L188-189: Why is the hypoxia event randomized by month of the year? Hypoxia would most likely occur during summer warming and stratification. It seems that varying the duration of summertime hypoxia is a more environmentally relevant exercise rather than randomizing what month the hypoxia event occurred in. How long was each hypoxia event? Author's reply: As we said before, we tested the effect of hypoxia as a stochastic event more than testing is in terms of frequency and timing. Thus, here we adopted a very simple scheme but there is a companion paper still under review (G. Sarà submitted PRS B) whose main aim was to test the effect of duration, frequency and timing of disturbance events on three different invertebrate species through the DEB model.

Reviewer wrote: Statistical approach needs to be justified (see detailed comments) Author's reply: Statistical approach has been justified in the detailed comments.

Reviewer wrote: The choice of using 2500 umol/kg for total alkalinity (TA) based on an oceanographic study for the lab experiments is strange (L106). Especially in static

cultures, mussels can alter the TA of a small body of water. I assume the authors did not measure TA during the experiment. In such a case, it would be best to simulate the experiment again, and measure TA so the authors have some idea of the TA variation in their experimental conditions could have been. Either way, the calculated pCO2 parameters will be undefinable without the real TA measurement. Author's reply: We did not find any paper reporting such alteration. We tried at the same time to minimize the number of organism for each tank and to perform a sufficient weekly water change in order to maintain a stable environment for our organisms, even if in mesocosm condition. We are perfectly aware that the suggested one is without any reasonable doubt the most appropriate approach to follow, but as soon as our is not a study focused on the chemistry of the shell but it is to provide a proof to test the predictive potential of DEB model about the effect of two stressors; thus, the use of a value from oceanographic study should be considered a minor approximation due to the metabolic and mechanistic nature of the paper.

Reviewer wrote: In addition to lacking an environmental context, the Discussion lacks comments on the non-DEB model functional traits (shell strength, dissolution patterns), their relevance to the study, and by what mechanism hypoxia and pH would differentially or synergistically impact the periostracum and shell quality. Author's reply:. we didn't analyse the impact on the shell chemistry and ultrastructure in this ms. whose main objective was to predict the effect of two stressors on mussel's LHs. Author's changes: However, to accomplish the referee's suggestion, we discussed shortly shell fragility related to pH and to both combined stressor.

Detailed comments: Reviewer #2 specific comment n. 1 Reviewer wrote: The title does not represent the study and reads as if the paper is a literature review. It would behove the authors include more detail in the title (DEB model, hypoxia, OA). Use of "marine bivalves" is inappropriate given that only one species was assessed. Author's changes: We agreed with Reviewer's #2 point and changed the title.

Reviewer #2 specific comment n. 2 Reviewer wrote: L27-33: sentence is difficult to

follow. Consider breaking this up. Author's changes: Sentence has been spitted up accordingly.

Reviewer #2 specific comment n. 3 Reviewer wrote: L39: this needs clarification (most functional traits? Which ones?) and references Author's changes: Although references were already present, following Reviewer #2 suggestion we specified the functional traits which we were referring to.

Reviewer #2 specific comment n. 4 Reviewer wrote: L47-48: plenty of labs conduct OA experiments for months up to at least one year Author's changes: The sentence has been deleted.

Reviewer #2 specific comment n. 5 Reviewer wrote: L48-54: long sentence, CO2 vents are unrelated to the second half of the sentence. Consider rewriting. Author's changes: Sentence has been rephrased accordingly.

Reviewer #2 specific comment n. 6 Reviewer wrote: L69: should AE be 'assimilation efficiency'? Author's changes: Sentence has been rephrased and clarified.

Reviewer #2 specific comment n. 7 Reviewer wrote: Introduction: break text up into paragraphs. Lacks introduction to functional trait-based models; I was expecting this prior to L54. Author's changes: Introduction was spitted into paragraphs following suggestion and an introduction to functional trait-based models was added.

Reviewer #2 specific comment n. 8 Reviewer wrote: L92: Mussels were fed ad libitum, but this is an energetics study. So how do the authors know the condition of the mussels used to get model parameters? Author's reply: As in reply to a similar point raised by Rev#1 we used the locution "ad libitum" to indicate a food concentration saturating the feeding processes of animals over time. Such an experimental maintenance condition is commonly used throughout the current literature when bivalves are maintained with ad libitum condition of food in bioenergetic experiments (e.g. Sarà et al., 2013; Montalto et al., 2014; Tagliarolo et al., 2016).

Reviewer #2 specific comment n. 9 Reviewer wrote: L104: dissolution threshold relates to calcium carbonate saturation state, please include the value here, rather than pH. Author's reply: Unfortunately we do not have such details on calcium carbonate saturation state. We added image details showing the effect of OA on the external shell layer, but a deep analysis on the chemical composition and alteration was out of the purpose of the present investigation.

Reviewer #2 specific comment n. 10 Reviewer wrote: L109: how was CO2 dissolved? Where their pumps in all the aquaria to ensure mixing? Author's changes: Details on how CO2 was dissolved and of water recirculation were added.

Reviewer #2 specific comment n. 11 Section 2.1: Sampling of what (section title)? How often was water replaced in the treatment tanks? Methods need a better description of how carbonate chemistry was calculated. What was the accuracy of the pH measurements? How often was the water sampled for each of the four parameters? How was oxygen maintained and measured in the treatments? Author's changes: Section title has been changed accordingly. Details on water changes, aeration, water recirculation and accuracy of pH measurements has been added, while details on the calculation of water chemistry were already present in section 2.1. Further details on sampling of water (8 time a day for pH), and on frequency of oxygen and temperature measurement has been added in the same section.

Reviewer #2 specific comment n. 12 Reviewer wrote: Section 2.2 (L112-120): if there are 25 mussels per tank, and there are 3 tanks, why are only five mussels observed for valve opening and closure? Why were observations made 6 times per day every week rather than fewer times per day but more frequently throughout the experiment? Does time of day matter for this behavior? What about time that food was added? I image that flow rates could affect this behavior, but it's unclear if water motion was the same across all tanks. Author's reply: We thank the referee for his/her interest on the behavioural part of our paper. The restricted number of observation was chosen in order to make the behavioural session during as less as possible in order to nor influence

with the operator presence mussel's behaviour. Even if it has not proven sometimes mussels suddenly close when moving in front of the tanks. For this reason we decided to observe less individual but more frequently. We didn't notice any difference in the behaviour during the day as mussels were inside a temperature-controlled room, under constant flow through conditions (according to Widdows and Staff 2006 and Sarà et al. 2013) and exposed to automated artificial daylight. The food was added at the end of the day, after all observations were made.

Reviewer #2 specific comment n. 13 Reviewer wrote: L124: define [pm] Author's changes: [áźŮM] has been here defined following suggestion.

Reviewer #2 specific comment n. 14 Reviewer wrote: L134: this equation results in units of O2 concentration x volume per unit time, oxygen units are not defined and there is no explanation as to how this is converted to [pm], which is in J per cubic cm per hour. What level of oxygen undersaturation was reached by the end of the incubation? Author's reply: Oxygen concentration ($\mu$mol l-1 h-1) were first converted in J h-1 and then in J cm-3 h-1 using a conversion factor (Kooijman, 2010) and following the current literature (Van der Veer et al., 2006; Ren and Schiel 2008). We never reached lethal oxygen concentration due to the short interval of the measurement as the idea was to simulate a sub-lethal effect as that reach at about 1.5-2.0 mg-l DO.

Reviewer #2 specific comment n. 15 Reviewer wrote: L140: this assumption should be justified Author's changes: As soon as the first two sentences of the section has the same reference, we moved it at the end of the second sentence.

Reviewer #2 specific comment n. 16 Reviewer wrote: Section 2.3: explain that the same individual was used for the respiration rate followed by AE. It's unclear until the end of section 2.4. Given that the respiration methods continue in the end of Section 2.4, merge Section 2.3 and 2.4. Author's reply: As answered to Reviewer #1 we do not agree in merging both sections because they represent two different part of metabolic stuff (feeding and respiration). However, we now clear specified that specimens were

the same between both measure following Reviewer's #2 suggestion.

Reviewer #2 specific comment n. 17 Reviewer wrote: L141-145: Please explain why AE experiment was not done in treatment water, and justify how AE can then be related to experimental treatments. Author's changes: We now clearly specified that the experiment was conducted with water specifically treated for each treatment.

Reviewer #2 specific comment n. 18 Reviewer wrote: L162: Again, if food availability is important at the field sites, food availability during the experiment should be known. Is it closer to that of Trieste or Palermo? Author's reply: As in reply to a similar point raised by Rev#1 we used the locution "ad libitum" to indicate a food concentration saturating the feeding processes of animals over time. Such an experimental maintenance condition is commonly used throughout the current literature when bivalves are maintained with ad libitum condition of food in bioenergetic experiments (e.g. Sarà et al., 2013; Montalto et al., 2014; Tagliarolo et al., 2016).

Reviewer #2 specific comment n. 19 Reviewer wrote: Section 2.7: How are simulations performed (what code or computer program?)? Author's reply: Our simulation were performed using R routine, and we specified it in the m accordingly.

Reviewer #2 specific comment n. 20 Reviewer wrote: L180: State what DEB parameters these are. Author's changes:: DEB parameters are now reported in Table 1.

Reviewer #2 specific comment n. 21 Reviewer wrote: L181: Is AE the same as [pM]? AE was already defined in the Introduction Author's changes: According to Reviewer's #1 point we checked and fixed both assimilation efficiency and the somatic maintenance costs definition.

Reviewer #2 specific comment n. 22 Reviewer wrote: L185-186: Was this data from week 1 or 4? How is a 4-week acclimation period determined sufficient enough to extrapolate to 4 years? Author's reply: The [áźŮM] parameter used inside our simulations was that calculated at week #4. All the species specific parameters derived from

one species and freely available online on the Add my pet collection were previously determined either by the covariation method through data present on literature or experimentally by short experimental sessions. Even without considering the effect of a stressor, a parameter estimated for a well-fed organism is then used inside simulation making predictions up to 4, 10 or even 50 years, as parameters are assumed to be specific for each species (Kooijman, 2010). We did not account any possible evolutionary effect whose effect is still far to be assessed in the DEB theory.

Reviewer #2 specific comment n. 23 Reviewer wrote: Section 2.9: The assumption of normally distributed residuals is not tested for the ANOVA. This needs to be done before moving forward with ANOVAs. A sample size of 16 is not large. The statistical analyses for valve closure does not match the data collection. By using ANOVAs, I assume all the data are pooled across the 4 weeks. This is not appropriate because it does not account for acclimation and it is a repeated measure since there are only 25 mussels in each tank which were observed over 4 weeks. ANOVAs also don't control for the tank replicate per treatment. The authors need to clarify how the data was pooled (and which behavior was analyzed – open or closed). Since this is binary data, reporting both in the bar graph is duplication the data (Section 3.2), report one, or as a stacked bar graph where each bar graph fills 100%. Author's reply: The assumption of normal distribution has been tested through the Anderson–Darling test using Past$^{®}$ software. We are aware that the sample size is not large, but pooling the six observation per week we obtain a sample of 24, and we believe it is sufficient for the purpose of the present paper. Author's changes: We repeated the data analysis and accordingly to the suggestion we compared week1 and week4 using two levels of the factor time and 4 levels of the factor treatment. We analysed the open valve behaviour and following suggestion we decided to use only one category in the graph. The paragraph, the table and the figure has been modified accordingly.

Reviewer #2 specific comment n. 24 Reviewer wrote: Section 3.1: Analysis comparing experimental treatments seems unnecessary, especially given the uncertainty of

the calculated parameters using a poor assumption of TA. Author's changes: Analysis comparing experimental treatments were removed accordingly to both referee's suggestions.

Reviewer #2 specific comment n. 25 Reviewer wrote: L350-352: is this to be expected? Author's reply: We thank the referee for the question and we have already answered to this point being highlighted by Referee#1. M. galloprovincialis in Sicily is observed to be limited by oligo-trophic conditions although it grows in highly trophic-enriched areas such as harbours or under Integrated Multi-Trophic Aquaculture (IMTA) conditions (Sarà et al 2012; 2013b, Giacoletti et al. 2018 in press JEMA) which supports what we gathered in the present ms. through the DEB simulations.

Reviewer #2 specific comment n. 26 Reviewer wrote: Figures: What is the error bar? Author's reply: The error bar indicated standard errors for means. Author's changes: Details were added in each figure following Reviewer's suggestions.

Reviewer #2 specific comment n. 27 Reviewer wrote: L259: capital I Author's changes:: Replaced.

Reviewer #2 specific comment n. 28 Reviewer wrote: L261: replace M&M with Section # Author's changes:: Replaced accordingly.

Reviewer #2 specific comment n. 29 Reviewer wrote: Table 1: Include temperature Author's changes: Temperature included inside Table 1.

Reviewer #2 specific comment n. 30 Reviewer wrote: Table 5: include input parameters Author's changes: DEB parameters were included in Table 1.

Please also note the supplement to this comment:
https://www.biogeosciences-discuss.net/bg-2018-13/bg-2018-13-AC3-supplement.pdf

[Figure]

**Supplement:**

**REVIEWER #2**

**Reviewer wrote:** This study integrates laboratory-derived parameters of mussel metabolism and assimilation efficiency to run DEB models testing the effect of pH and hypoxia, using environ- mental data input (temperature, food) from two sites within the mussel's biogeographic range. I appreciate the approach of introducing hypoxia events (although I have a comment on the design these events) as a means of incorporating environmental variability in the model. This literature is sparse with such perspectives, especially in the context of multiple stressors. However, the paper lacks a perspective of the environmental relevance of the experimental design and modelling.

**Author's reply:** We thank the Reviewer #2 for helping us improving the readability and the clearness of the ms.

**Author's changes:** In doing so we applied most of the suggestion for the highlighted points, and we discussed them through the specific comments. A perspective of environmental relevance on modelling outputs has been currently added in the discussion section following what suggested by the reviewer.

**Major comments**

Methods need much more detail (see detailed comments)

**Reviewer wrote:** Physiological condition of experimental mussels during the experiment is not quantified. Feeding was ad libitum and it was not assessed if mussels were being fed at conditions of either site used for the modeling. This is problematic as, for example, if the mussels are starving relative to their natural food supply, the derived experimental parameters for the DEB model may be inappropriate. The authors also do not explain the experimental design. Why was the experiment 4 weeks?

Author's reply: We thank the reviewer for highlighting this point.

**Author's changes:** We estimated the condition index through the biometric available data and compared it through the experiment, resulting in a not significant variation throughout the study period, and this supports that experimental animals were not stressed by starvation. Further, we used the locution "ad libitum" to indicate a food concentration saturating the feeding processes of animals over time. Such an experimental maintenance condition is commonly used throughout the current literature when bivalves are maintained with ad libitum condition of food in bioenergetic experiments (e.g. Sarà et al., 2013; Montalto et al., 2014; Tagliarolo et al., 2016). However, we adopted a four weeks-period to estimate the effect of OA on functional traits of mussels; such a period is judged to be enough to allow mussels to acclimate to new conditions, as showed in many experimental papers across the current literature(references added in the manuscript).

**Reviewer wrote:** DEB models: As a reader of Biogeosciences, but not an expert on DEB models, it would be helpful if the authors reviewed this approach more clearly (perhaps using a schematic, what program is used to run models, table of input variables etc.). After reading the paper, I am unclear about the exact implementation and conclusions that can be drawn from this simulation based on the following 4 sources of confusion:

1)From what I understand, temperature from 2006-2009 is used as one of the model inputs. However, all the biological parameters taken from the experiments come from 21 degr. C (although this is not stated explicitly for respiration, but I assume it's 21 C). Since environmental data would vary in terms of temperature across the four years, I don't understand how biological performance is scaled across this temperature regime. It would be good to include a figure of the environmental data (means are not great time-series descriptors, especially for biological parameters were scaled for temperature effects over the years. This same argument applies to food concentration (which I assume varies by time of year as well).

**Author's reply:** The 2006-2009 thermal series has been used as a forcing variables inside DEB models in current modelling literature (Sarà et al. 2011, 2013b; Montalto et al. 2014). The most important factors driving changes in energy budgets of ectotherms are body temperature, on which every metabolic rates depends via the Arrhenius relationships (Kooijman 2010), and food. Arrhenius temperature, that is species specific, acts as a correction factors inside DEB models to scale all rates to environmental temperature. At the same time DEB models use the 2006-2009 CHL-a series as a second forcing variables to predict LH-traits of our species. Accordingly, including a new figure of environmental data could make the ms. heavier also as the main object is not to contextualise the effect at that period, but to show that stressor's effect is simulated across a long integrated period.

**Reviewer wrote:** 2) The authors use hypoxia and acidification, two future stressors, with temperature data from a few years ago. This design ignores the fact that warming is currently the dominant stressors for this species in the Mediterranean Sea and is expected to continue in the future. As the environmental relevance of the study design is not discussed, as written, the results do not match any realistic environmental situation. This counters the original intent of using DEB models to better "predict organismal functional traits, capturing variation across species to solve a very wide range of problems in ecology and evolutionary biology" (L58).

**Author's reply:** We did not test the effect of increasing temperature as we are pretty sure that the thermal effect is not manifested on a period so short (only 4-years); however other companion papers (e.g. Montalto et al. 2017) tested the effect of increasing temperature on mussel's performances throughout the whole Basin. To extrapolate the effect only two stressors, we carried out simulations under two different latitudinal temperature patterns (Trieste, north Adriatic and Palermo, Southern MED). Anyway, we included some discussion lines about this issue.

**Reviewer wrote:** 3) Given that reproduction of this species can be quite seasonal, how does the DEB model handle this in terms of estimating reproductive output?

**Author's reply:** DEB manages the seasonal reproduction throughout thresholds based on the temperature-energy relationships.

**Reviewer wrote:** 4) L188-189: Why is the hypoxia event randomized by month of the year? Hypoxia would most likely occur during summer warming and stratification. It seems that varying the duration of summertime hypoxia is a more environmentally relevant exercise rather than randomizing what month the hypoxia event occurred in. How long was each hypoxia event?

**Author's reply:** As we said before, we tested the effect of hypoxia as a stochastic event more than testing is in terms of frequency and timing. Thus, here we adopted a very simple scheme but there is a companion paper still under review (G. Sarà submitted PRS B) whose main aim was to test the effect of duration, frequency and timing of disturbance events on three different invertebrate species through the DEB model.

**Reviewer wrote:** Statistical approach needs to be justified (see detailed comments) **Author's reply:** Statistical approach has been justified in the detailed comments.

**Reviewer wrote:** The choice of using 2500 umol/kg for total alkalinity (TA) based on an oceanographic study for the lab experiments is strange (L106). Especially in static cultures, mussels can alter the TA of a small body of water. I assume the authors did not measure TA

during the experiment. In such a case, it would be best to simulate the experiment again, and measure TA so the authors have some idea of the TA variation in their experimental conditions could have been. Either way, the calculated pCO2 parameters will be undefinable without the real TA measurement.

**Author's reply:** We did not find any paper reporting such alteration. We tried at the same time to minimize the number of organism for each tank and to perform a sufficient weekly water change in order to maintain a stable environment for our organisms, even if in mesocosm condition. We are perfectly aware that the suggested one is without any reasonable doubt the most appropriate approach to follow, but as soon as our is not a study focused on the chemistry of the shell but it is to provide a proof to test the predictive potential of DEB model about the effect of two stressors; thus, the use of a value from oceanographic study should be considered a minor approximation due to the metabolic and mechanistic nature of the paper.

**Reviewer wrote:** In addition to lacking an environmental context, the Discussion lacks comments on the non-DEB model functional traits (shell strength, dissolution patterns), their relevance to the study, and by what mechanism hypoxia and pH would differentially or synergistically impact the periostracum and shell quality.

**Author's reply:** we didn't analyse the impact on the shell chemistry and ultrastructure in this ms. whose main objective was to predict the effect of two stressors on mussel's LHs. **Author's changes:** However, to accomplish the referee's suggestion, we discussed shortly shell fragility related to pH and to both combined stressor.

**Detailed comments:**

**Reviewer #2 specific comment n. 1**

**Reviewer wrote:** The title does not represent the study and reads as if the paper is a literature review. It would behave the authors include more detail in the title (DEB model, hypoxia, OA). Use of "marine bivalves" is inappropriate given that only one species was assessed. **Author's changes:** We agreed with Reviewer's #2 point and changed the title.

**Reviewer #2 specific comment n. 2**

**Reviewer wrote:** L27-33: sentence is difficult to follow. Consider breaking this up. **Author's changes:** Sentence has been spitted up accordingly.

**Reviewer #2 specific comment n. 3**

**Reviewer wrote:** L39: this needs clarification (most functional traits? Which ones?) and references

**Author's changes:** Although references were already present, following Reviewer #2 suggestion we specified the functional traits which we were referring to.

**Reviewer #2 specific comment n. 4**

**Reviewer wrote:** L47-48: plenty of labs conduct OA experiments for months up to at least one year

Author's changes: The sentence has been deleted.

**Reviewer #2 specific comment n. 5**

**Reviewer wrote:** L48-54: long sentence, CO2 vents are unrelated to the second half of the sentence. Consider rewriting.

Author's changes: Sentence has been rephrased accordingly.

**Reviewer #2 specific comment n. 6**

**Reviewer wrote:** L69: should AE be 'assimilation efficiency'? **Author's changes:** Sentence has been rephrased and clarified.

**Reviewer #2 specific comment n. 7**

**Reviewer wrote:** Introduction: break text up into paragraphs. Lacks introduction to functional trait-based models; I was expecting this prior to L54.

Author's changes: Introduction was spitted into paragraphs following suggestion and an introduction to functional trait-based models was added.

**Reviewer #2 specific comment n. 8**

**Reviewer wrote:** L92: Mussels were fed ad libitum, but this is an energetics study. So how do the authors know the condition of the mussels used to get model parameters?

**Author's reply:** As in reply to a similar point raised by Rev#1 we used the locution "ad libitum" to indicate a food concentration saturating the feeding processes of animals over time. Such an experimental maintenance condition is commonly used throughout the current literature when bivalves are maintained with ad libitum condition of food in bioenergetic experiments (e.g. Sarà et al., 2013; Montalto et al., 2014; Tagliarolo et al., 2016).

**Reviewer #2 specific comment n. 9**

**Reviewer wrote:** L104: dissolution threshold relates to calcium carbonate saturation state, please include the value here, rather than pH.

**Author's reply:** Unfortunately we do not have such details on calcium carbonate saturation state. We added image details showing the effect of OA on the external shell layer, but a deep analysis on the chemical composition and alteration was out of the purpose of the present investigation.

**Reviewer #2 specific comment n. 10**

**Reviewer wrote:** L109: how was CO2 dissolved? Where their pumps in all the aquaria to ensure mixing?

Author's changes: Details on how CO2 was dissolved and of water recirculation were added.

**Reviewer #2 specific comment n. 11**

Section 2.1: Sampling of what (section title)? How often was water replaced in the treatment tanks? Methods need a better description of how carbonate chemistry was calculated. What was the accuracy of the pH measurements? How often was the water sampled for each of the four parameters? How was oxygen maintained and measured in the treatments?

**Author's changes:** Section title has been changed accordingly. Details on water changes, aeration, water recirculation and accuracy of pH measurements has been added, while details on the calculation of water chemistry were already present in section 2.1. Further details on sampling of water (8 time a day for pH), and on frequency of oxygen and temperature measurement has been added in the same section.

**Reviewer #2 specific comment n. 12**

**Reviewer wrote:** Section 2.2 (L112-120): if there are 25 mussels per tank, and there are 3 tanks, why are only five mussels observed for valve opening and closure? Why were observations made 6 times per day every week rather than fewer times per day but more frequently throughout the experiment? Does time of day matter for this behavior? What about time that food was added? I image that flow rates could affect this behavior, but it's unclear if water motion was the same across all tanks.

**Author's reply:** We thank the referee for his/her interest on the behavioural part of our paper. The restricted number of observation was chosen in order to make the behavioural session during as less as possible in order to nor influence with the operator presence mussel's behaviour. Even if it has not proven sometimes mussels suddenly close when moving in front of the tanks. For this reason we decided to observe less individual but more frequently. We didn't notice any difference in the behaviour during the day as mussels were inside a temperature-controlled room, under constant flow through conditions (according to Widdows and Staff 2006 and Sarà et al. 2013) and exposed to automated artificial daylight. The food was added at the end of the day, after all observations were made.

**Reviewer #2 specific comment n. 13**

**Reviewer wrote:** L124: define [pm] **Author's changes:** [pm] has been here defined following suggestion.

**Reviewer #2 specific comment n. 14**

**Reviewer wrote:** L134: this equation results in units of O2 concentration x volume per unit time, oxygen units are not defined and there is no explanation as to how this is converted to [pm], which is in J per cubic cm per hour. What level of oxygen undersaturation was reached by the end of the incubation?

**Author's reply:** Oxygen concentration ( $\mu$ mol l-1 h-1) were first converted in J h-1 and then in J cm-3 h-1 using a conversion factor (Kooijman, 2010) and following the current literature (Van der Veer et al., 2006; Ren and Schiel 2008). We never reached lethal oxygen concentration due to the short interval of the measurement as the idea was to simulate a sub-lethal effect as that reach at about 1.5-2.0 mg-l DO.

**Reviewer #2 specific comment n. 15**

Reviewer wrote: L140: this assumption should be justified

Author's changes: As soon as the first two sentences of the section has the same reference, we moved it at the end of the second sentence.

**Reviewer #2 specific comment n. 16**

**Reviewer wrote:** Section 2.3: explain that the same individual was used for the respiration rate followed by AE. It's unclear until the end of section 2.4. Given that the respiration methods continue in the end of Section 2.4, merge Section 2.3 and 2.4.

**Author's reply:** As answered to Reviewer #1 we do not agree in merging both sections because they represent two different part of metabolic stuff (feeding and respiration). However, we now clear specified that specimens were the same between both measure following Reviewer's #2 suggestion.

**Reviewer #2 specific comment n. 17**

**Reviewer wrote:** L141-145: Please explain why AE experiment was not done in treatment water, and justify how AE can then be related to experimental treatments.

Author's changes: We now clearly specified that the experiment was conducted with water specifically treated for each treatment.

**Reviewer #2 specific comment n. 18**

**Reviewer wrote:** L162: Again, if food availability is important at the field sites, food availability during the experiment should be known. Is it closer to that of Trieste or Palermo? **Author's reply:** As in reply to a similar point raised by Rev#1 we used the locution "ad libitum" to indicate a food concentration saturating the feeding processes of animals over

time. Such an experimental maintenance condition is commonly used throughout the current literature when bivalves are maintained with ad libitum condition of food in bioenergetic experiments (e.g. Sarà et al., 2013; Montalto et al., 2014; Tagliarolo et al., 2016).

**Reviewer #2 specific comment n. 19**

**Reviewer wrote:** Section 2.7: How are simulations performed (what code or computer program?)?

Author's reply: Our simulation were performed using R routine, and we specified it in the m accordingly.

**Reviewer #2 specific comment n. 20**

**Reviewer wrote:** L180: State what DEB parameters these are. **Author's changes::** DEB parameters are now reported in Table 1.

**Reviewer #2 specific comment n. 21**

**Reviewer wrote:** L181: Is AE the same as [pM]? AE was already defined in the Introduction **Author's changes:** According to Reviewer's #1 point we checked and fixed both assimilation efficiency and the somatic maintenance costs definition.

**Reviewer #2 specific comment n. 22**

**Reviewer wrote:** L185-186: Was this data from week 1 or 4? How is a 4-week acclimation period determined sufficient enough to extrapolate to 4 years?

**Author's reply:** The  $[\dot{p}_M]$  parameter used inside our simulations was that calculated at week #4. All the species specific parameters derived from one species and freely available online on the Add my pet collection were previously determined either by the covariation method through data present on literature or experimentally by short experimental sessions. Even without considering the effect of a stressor, a parameter estimated for a well-fed organism is then used inside simulation making predictions up to 4, 10 or even 50 years, as parameters are assumed to be specific for each species (Kooijman, 2010). We did not account any possible evolutionary effect whose effect is still far to be assessed in the DEB theory.

**Reviewer #2 specific comment n. 23**

**Reviewer wrote:** Section 2.9: The assumption of normally distributed residuals is not tested for the ANOVA. This needs to be done before moving forward with ANOVAs. A sample size of 16 is not large. The statistical analyses for valve closure does not match the data collection. By using ANOVAs, I assume all the data are pooled across the 4 weeks. This is not appropriate because it does not account for acclimation and it is a repeated measure since there are only 25 mussels in each tank which were observed over 4 weeks. ANOVAs also don't control for the tank replicate per treatment. The authors need to clarify how the data was pooled (and which behavior was analyzed – open or closed). Since this is binary data, reporting both in the bar graph is duplication the data (Section 3.2), report one, or as a stacked bar graph where each bar graph fills 100%.

**Author's reply:** The assumption of normal distribution has been tested through the Anderson–Darling test using Past® software. We are aware that the sample size is not large, but pooling the six observation per week we obtain a sample of 24, and we believe it is sufficient for the purpose of the present paper.

**Author's changes:** We repeated the data analysis and accordingly to the suggestion we compared week1 and week4 using two levels of the factor time and 4 levels of the factor treatment. We analysed the open valve behaviour and following suggestion we decided to use

only one category in the graph. The paragraph, the table and the figure has been modified accordingly.

**Reviewer #2 specific comment n. 24**

**Reviewer wrote:** Section 3.1: Analysis comparing experimental treatments seems unnecessary, especially given the uncertainty of the calculated parameters using a poor assumption of TA.

Author's changes: Analysis comparing experimental treatments were removed accordingly to both referee's suggestions.

**Reviewer #2 specific comment n. 25**

Reviewer wrote: L350-352: is this to be expected?

**Author's reply:** We thank the referee for the question and we have already answered to this point being highlighted by Referee#1. *M. galloprovincialis* in Sicily is observed to be limited by oligo-trophic conditions although it grows in highly trophic-enriched areas such as harbours or under Integrated Multi-Trophic Aquaculture (IMTA) conditions (Sarà et al 2012; 2013b, Giacoletti et al. 2018 in press JEMA) which supports what we gathered in the present ms. through the DEB simulations.

**Reviewer #2 specific comment n. 26**

**Reviewer wrote:** Figures: What is the error bar? **Author's reply:** The error bar indicated standard errors for means. **Author's changes:** Details were added in each figure following Reviewer's suggestions.

Reviewer #2 specific comment n. 27 Reviewer wrote: L259: capital I Author's changes:: Replaced.

**Reviewer #2 specific comment n. 28**

**Reviewer wrote:** L261: replace M&M with Section # **Author's changes::** Replaced accordingly.

**Reviewer #2 specific comment n. 29**

**Reviewer wrote:** Table 1: Include temperature **Author's changes:** Temperature included inside Table 1.

**Reviewer #2 specific comment n. 30**

**Reviewer wrote:** Table 5: include input parameters **Author's changes:** DEB parameters were included in Table 1.

**Predicting the multiple effects of acidification and hypoxia on Mytilus galloprovincialis (Bivalvia, Mollusca) life history traits Functional spatial contextualisation of the effects of multiple stressors in marine bivalves**

**6 Antonio Giacoletti\* and Gianluca Sarà**

7 Dept. of Earth and Marine Sciences (DISTEM) University of Palermo, Palermo, Italy;

8 Correspondence to: Antonio Giacoletti (antonio.giacoletti@unipa.it)

9

10 Abstract. Many recent studies have revealed that the majority of environmental stressors experienced by marine 11 organisms (ocean acidification, global warming, hypoxia etc.) occur at the same time and place, and that their 12 interaction may complexly affect a number of ecological processes. Here, we experimentally investigated the 13 effects of pH and hypoxia on the functional and behavioural traits of the mussel Mytilus galloprovincialis, we 14 then simulated the potential effects on growth and reproduction dynamics trough a Dynamic Energy Budget 15 (DEB) model under a multiple stressor scenario. Our simulations showed that hypercapnia had a remarkable 16 effect by reducing the maximal habitat size and reproductive output differentially as a function of the trophic 17 conditions, where modelling was spatially contextualized. This study showed the major threat represented by the 18 hypercapnia and hypoxia phenomena for the growth, reproduction and fitness of mussels under the current 19 climate change context, and that a mechanistic approach based on DEB modelling can illustrate complex and 20 site-specific effects of environmental change, producing that kind of information useful for management 21 purposes, at larger temporal and spatial scales.

Key-words: Acidification; Climate change; DEB Model; Hypoxia; *Mytilus galloprovincialis*; Multiple-Stressor;
 Mussel.

**1 Introduction**

| 25 | Since the dawn of research investigating the possible effects of ocean acidification (OA) on aquatic organisms      |
|----|---------------------------------------------------------------------------------------------------------------------|
| 26 | (e.g. Bamber, 1990), most studies have shown that elevated $pCO_2$ levels, as predicted for the next century, may   |
| 27 | affect -to some extent the functional traits (Schoener, 1986; Koehl, 1989) of marine organisms (Feely et al.,       |
| 28 | 2004; Navarro et al., 2013). Referring to functional traits, we consider all those specific traits that define each |
| 29 | species in terms of their ecological roles (Diaz & Cabido, 2001), and thereby the species' identity. In marine      |
| 30 | ectotherms such as bivalves, crabs, sea urchins and fish, these traits include tolerance and sensitivity to         |
| 31 | environmental conditions (e.g. physiological tolerance limits - Kearney & Porter, 2009) defining the ability of     |
| 32 | each species to support their own metabolic machinery (Sokolova et al., 2012; Sarà et al., 2014).7 They further     |
| 33 | include the ability to obtain energy from food, the so-called functional response (Holling, 1959) or those          |
| 34 | behavioural (e.g. swimming behaviour, habitat use, mating system) and morphological (e.g. shape, thickness)         |
| 35 | traits (Schoener, 1986) which led to optimise the energetic income (Krebs & Davies, 1992) and lastly to reach       |
| 36 | the ultimate fitness (Roff, 1992).                                                                                  |
| 37 | Research performed over the last decade and summarized in the recent IPCC (2014) report (IPCC, 2014)-clearly        |
| 38 | shows that ocean acidification will affect marine organisms and ecosystems (Connell et al., 2017) in the coming     |
| 39 | decades, and such projections have stimulated new research that aims to understand the impact on calcifying         |
| 40 | marine organisms. Reductions in growth and calcification rates are just those kinds of the physiological impacts    |
| 41 | of ocean acidification (Thomsen et al., 2013; Byrne, 2012; Beniash et al., 2010). While much research showed        |
| 42 | that low pH may impair most functional traits (e.g. respiration), functions connected with energy uptake such as    |
| 43 | feeding and assimilation seem to be reduced at a larger extent in many species with expected implications for the   |
| 44 | amount of energy available for growth and reproduction (Kurihara et al., 2008; Appelhans, 2012; Navarro et al.,     |
| 45 | 2013; Zhang et al., 2015). Such information has been obtained through both acute and chronic exposure to OA         |
| 46 | but no studies are yet available to assess the potential effects of OA on the magnitude of other Life History (LH)  |
| 47 | traits, such as maximum habitat body size, fecundity, time to reach maturation and the number of spawning           |
| 48 | events under future conditions of environmental change (sensu Kearney and Porter, 2009; Sarà et al., 2011;          |
| 49 | 2013b). To obtain such LH traits, experiments should be long enough to assure a functional effect of lower pH       |
| 50 | for many weeks or months but probably no existing lab mesocosm could currently assure the stability of              |
| 51 | seawater acidification system for such a long time. Thus, apart from long term experiments carried out in few       |
| 52 | field sites worldwide (e.g. Ischia [Hall-Spencer et al., 2008] and Vulcano [Duquette et al., 2015] islands) in the  |
| 53 | Southern Mediterranean Sea and in other Seas (Maug Island [Pala, 2009] or-CO2 vents in the SW southwest             |
|    |                                                                                                                     |

54 Pacific [Connell et al., 2017]) where lowered pH seawater is naturally available through CO2 natural emissions 55 from vents, the recent introduction of mechanistic functional trait-based (FT) models based on the Dynamic 56 Energy Budget theory (DEB; Kooijman, 2010) can offer a reliable opportunity for disentangling the effect of 57 OAseawater acidification on LH traits. Functional trait-based DEB (FT-DEB) approach (Kearney and Porter 2009; Kearney et al., 2010; Kearney 2012; 58 59 Sarà et al., 2011, 2012, 2013a, b, 2018) relies on the quantitative prediction of organismal functional traits and 60 fecundity within the fundamental niche limits of one particular species (Hutchinson 1957). Such an approach 61 aim to exploit mechanistic rules to connect environmental human-induced variability to functional traits 62 (Schoener 1986; Diaz & Cabido 2001) and in turn functional traits to species LH (Stearns 1992) traits. The 63 novelty of the FT-DEB approach relies on its intrinsic mechanistic nature deriving from the fact that it is based 64 on flux of energy and mass through an organism which are traceable processes that are subject to conservation 65 laws (according to the new posited concept of ecomechanics; Denny & Helmuth, 2009; Denny & Benedetti-66 Cecchi, 2012; Carrington et al., 2015). This provides an exceptionally powerful tool to predict organismal 67 functional traits, capturing variation across species to solve a very wide range of problems in ecology and 68 evolutionary biology (Lika et al., 2011; Kearney, 2012; Pouvreau et al., 2006; Pequerie et al., 2010; Sarà et al., 69 2011; 2012; 2013a; 2013b; 2014). FT-DEB could provide information about the effect of seawater-acidification 70 on the fecundity (as expressed by the number of gametes per life span, the so-called Darwinian fitness; 71 Bozinovic et al., 2011) and the degree of reproductive failure of species providing theoretical predictions about 72 LH traits having implications on population dynamics and community structure throughout the species range 73 (Sarà et al., 2013a). Here, we specifically exploited the FT-DEB model spatially and explicitly contextualised 74 along the Italian coasts under subtidal conditions (Kearney et al., 2010; Sarà et al., 2011; 2012; 2013a; 2013b), 75 using four-year thermal series and satellite Chlorophyllchlorophyll-a (CHL-a) concentrations, to test the multiple 76 effect due to the combination of pH and hypoxia on the physiological and behavioural traits of our target species, 77 the bivalve Mytilus galloprovincialis (Lamarck 1819). 78 Recent insights obtained by the experimental research have shown that OA mainly affects feeding rates (FR), 79 assimilation efficiency (AE) and maintenance costs rates of marine organisms (Appelhans, 2012; Navarro et al. 80 2013; Kroeker et al. 2014; Zhang et al., 2015; Jager et al. 2016). Here, we translated the combined effects of 81 hypoxia and hypercapnia on AE-assimilation and oxygen consumption rates as measured under different 82 treatments into effects on assimilation AE and somatic maintenance costs as expressed by the DEB [ $\dot{p}_{M}$ ]

83 parameters. Somatic maintenance is a crucial suite This latter is a crucial of functional traits used in recent

84 bioenergetics based on the DEB theory that mechanistically can be used to investigate the role played by 85 multiple stressors on LH traits of organisms by using first principles (Sarà et al., 2014). We further documented 86 the effects of those stressors on M. galloprovincialis shells through the use of a scanning electron microscope 87 (SEM), and compared the maximum shell breaking load of treated vs. control specimens. A behavioural analysis 88 completed the frame concerning the individual's response to both single and combined stressors. Carried out in a 89 context of OA, this exercise comprises a first step in linking the fields of ecomechanics and climate change 90 ecology, which should yield a more mechanistic understanding of how biodiversity will respond to 91 environmental change (sensu Buckley et al., 2012).

92

93

**2 Materials and methods**

94 This study articulated consisted of three steps: 1) laboratory investigation on the effects of pH and hypoxia on 95 functional and (both behavioural and physiological) traits of *Mytilus galloprovincialis*; 2) collection of water 96 temperature data; and Chlorophyll a (CHL-a)-\_data from two Mediterranean sites (Trieste and Palermo), as a 97 further forcing variable in the DEB model and lastly 3) model running to simulate growth and fitness of *M*. 98 galloprovincialis under stressful conditions by using estimated DEB parameters estimated arising from by the 99 activities in the first step.

100

101 2.1 Sampling and eExperimental set-up. Specimens of M. galloprovincialis (45 - 55 mm) were provided by 102 the Ittica Alimentare Soc. Coop. Arl. (Palermo) and transferred within 30 minutes to the laboratory. Mussels 103 were then carefully cleaned and placed in a 300L tank filled with natural seawater at room temperature (18-104 20°C), field salinity (37-38-PSU), and fed ad libitum with cultured Isochrysis galbana (Sarà et al., 2011). 105 According to common experimental procedures for studying the bioenergetics of bivalves (Sarà et al., 2008; 106 Ezgeta Balic et al., 2011), mMussels were acclimated for two weeks to reduce stress generated by manipulation 107 and transport (Sarà et al., 2013a), and  $\Theta$  once acclimated, 200 specimens were randomly divided in groups of 25 108 organisms, transferred to 8 independent rectangular glass tanks of 120L capacity (100 cm long, 30 cm deep, 40 109 cm wide) and kept in a conditioned room at 21°C for 4 weeks according to common protocol with bivalves 110 (Braby & Somero 2006; Fields et al., 2012; Kittner and Riisgård 2005). Tanks 1 to 4 were filled with sea water 111 and continuously with aerated through air pumpsand recirculating sea water, while Tanks 5 to 8 were not aerated 112 and covered with a plastic film disposed on the water surface, in order to avoid gas-exchanges between air and 113 water. Tanks 1-2 were used as a control (CTRL), while hypercapnia was imposed in Tanks 3-4 (Tr1), hypoxia (2

| 114 | ppm) in Tanks 5-6 (Tr2), and both factor (pH 7.5 and hypoxia) in Tanks 7-8 (Tr3) (see Table 2). Mussels were                                                            |
|-----|-------------------------------------------------------------------------------------------------------------------------------------------------------------------------|
| 115 | acclimated to two different nominal pH treatments: (i) pH 8.0 in Tanks 1-2 (CTRL) and 5-6 (Tr2), corresponding                                                          |
| 116 | to present average pH at the sampling site; and (ii) pH 7.5 in Tanks 2-3 (Tr1) and 7-8 (Tr3), deviating from                                                            |
| 117 | present range of natural variability and relevant for 2100 ocean acidification scenarios. This last point is                                                            |
| 118 | considered the critical dissolution threshold of calcium carbonate in shelled animals as reported in literature                                                         |
| 119 | (Melzner et al., 2011; Gazeau et al., 2013). The carbonate system speciation ( $p$ CO2, HCO 3 - , CO 3 2- , $\Omega$ Ca and |
| 120 | $\Omega$ Ar) was calculated from pH NBS , temperature, salinity and alkalinity (T A = 2.5 mM; Rivaro et al., 2010) using                          |
| 121 | CO2SYS (see Table 2: Lewis and Wallace, 1998) with dissociation constants from Dickson & Millero (1987).                                                                |
| 122 | The pH was manually controlled 8 times a day by an electronic pH-meter (Cyberscan 510, Eutech Instruments;                                                              |
| 123 | accuracy = <math>\pm</math> 0.01 pH) and gaseous CO2 was injected directly into the aquarium through a commercial ceramic                             |
| 124 | diffusor, when required. Oxygen concentration and temperature were monitored with the same frequency                                                                    |
| 125 | through the PiroScience FirestingO2 oxygen logger equipped with a dedicated temperature sensor. Water                                                                   |
| 126 | movement and recirculation were assured by water pumps. Tanks were siphoned at the end of each working day,                                                             |
| 127 | removing all the faecal material in order to avoid the accumulation of waste products, and 20% of water was                                                             |
| 128 | weekly changed with specific pre-conditioned sea water for each treatment.                                                                                              |

130 2.2 Behavioural observations. The valve gape of mussels was recorded by means of the two simplest 131 behavioural categories reported in Jørgensen et al. (1988): closed valves and opened valves. Each observation 132 was carried out by an operator with the aim to record changes in the behavioural repertoire of bivalves in 133 response to the exposure to a single stressor (pH or hypoxia) and to both pH and hypoxia, compared to 134 individuals kept in normal environmental conditions. All experiments were conducted at environmental (37-38 135 PSU) salinity and with well-aerated sea water through a gentle flow (Ameyaw-Akumfi & Naylor, 1987), except 136 for specimens of Tank 5-6 and 7-8, that were not aerated in order to maintain the hypoxia level set through the 137 gaseous nitrogen. Behavioural observations were repeated six times a day at week 1 and 4 of exposure, on day 7, 138 14, 21, 28, and involved involving 5 random specimens for each treatment.

139

129

140**2.3 Oxygen consumption.** The rate of oxygen consumption was determined twice (week 1 and week 4) in a141respirometric glass chamber (0.3L) inside a temperature-controlled water bath, in order to compare-investigate142the effects of multiple stressors by converting rates intoon metabolic somatic maintenance costs - the DEB143parameter [ $\dot{p}_{M}$ ] (expressed as J cm-3 h-1) linked to the energetic cost of maintenance in orderand to integrate it in

| 144 | the standard DEB model. Volume-specific somatic maintenance costs , as expressed by the $[\dot{p}_{M}]$ parameter (J cm                          |
|-----|---------------------------------------------------------------------------------------------------------------------------------------------------------|
| 145 | $\frac{3}{1}$ h -1 ), represent the amount of energy needed to fuel basal metabolism ( $\dot{p}_{M}$ ) scaled with the organisms' volume.    |
| 146 | such as <math>([\dot{p}_M] = \dot{p}_M/V)</math> . All determinations were performed at 21°C using filtered seawater with the same pH and |
| 147 | oxygen content as that of the respective treatment, stirred with a magnetic stirrer bar beneath a perforated glass                                      |
| 148 | plate supporting each individual (Sarà et al., 2008; Ezgeta-Balic et al., 2011). The decline in oxyger                                                  |
| 149 | concentration was measured by a PiroScience FirestingO2 respirometer, capable of four sensor connections. We                                            |
| 150 | used a total of $n = 64$ mussels per week, 16 for each treatment (8 for each tank) acclimated as above, fed acc                                         |
| 151 | libitum until the day before the experiment. The decline was continuously recorded for at least 1 h, excluding an                                       |
| 152 | initial period (~ 10 min) when usually there is a more rapid decline in oxygen caused by a disturbance of the                                           |
| 153 | sensor's temperature equilibration. Respiration rate (RR, $\mu$ mol O 2 h -1 ) was calculated according to (Ezgeta Balic          |
| 154 | et al., 2011; Sarà et al., 2008; 2013b): $RR = (C_{t0} - C_{t1})x Vol_r x 60(t_1 - t_0)^{-1}$ , where $C_{t0}$ is oxygen                                |
| 155 | concentration at the beginning of the measurement, $C_{t1}$ is the oxygen concentration at the end of the                                               |
| 156 | measurement, and $Vol_r$ is the volume of water in the respirometric chamber. Volume-specific somatic                                            |
| 157 | maintenance costs were then calculated by converting oxygen consumption rates expressed in µmol h -1 in J h -1                    |
| 158 | through a conversion factor (Kooijman 2010) and then in J cm -3 (van der Veer et al., 2006; Ren and Schiel 2008)                             |
| 159 | (for the calculation of dry weights refer to the end of section 2.4).                                                                                   |
|     |                                                                                                                                                         |

160

161 2.4 Assimilation efficiency. Assimilation is the final step of food processing and it represents the efficiency with 162 which organic material is absorbed from the ingested food (Kooijman, 2010). The assimilation of food is 163 assumed to be independent of the feeding rate per se, but proportional to the ingestion rate (Kooijman, 2010). 164 Assimilation efficiency (AE) was measured through the Conover ratio (1966) AE = (F - E)/[(1 - E)F], where 165 F is the ratio between ash-free dry weight (AFDW) and dry weight (DW) for food, and E is the same ratio for the 166 faeces; this represents the efficiency with which organic material is absorbed from the ingested food material. 167 Here, after oxygen consumption measurement, the same 16-specimens of M. galloprovincialis per treatment 168 were-collected twice (week 1 and week 4) and placed into separate beakers containing 1L of filtered seawater 169 (specific for each treatment) and a magnetic stirrer bar. In order to allow the mussels to open their valves and 170 start their filtration activity, they were given 15 minutes before the introduction of food with an initial 171 concentration of ~ 15,000 Isochrysis galbana cells ml-1. After a period of 2 h mussels were moved to cleaned 1L 172 glass beakers with filtered seawater for a period of 12 h, after that the water contained in each beaker was filtered 173 on pre-ashed and weighted GF/C fibreglass filters. Once filtered, filters were washed with 0.5 M ammonium 174 formate (purest grade) to remove adventitious salts (Widdows & Staff, 2006Sarà et al., 2013a), dried in the oven 175 (95°C for 24 h) and then incinerated in a muffle furnace (450°C for 4 h). After each step, the samples were 176 weighted using a balance (Sartorius BL 120S  $\pm$  1µg). For the calculation of AE, together with the faeces 177 collected from the mussels, filters containing algal food were dried and incinerated as above. After respirometric 178 measurement and the collection of faeces each animal was killed by gentle freezing and dissected, and the shells 179 were separated from the body tissue in order to calculate the condition index according to Davenport & Chen 180 (1987) (CI = (body weight/shell weight)  $\times$  100), and their individual dry weights and to standardize respiration 181 rates.- to body weights.

182

183 2.5 Water temperature data. The main forcing driver of shellfish LH inside DEB models is represented by 184 mean-seawater temperature (Pouvreau et al., 2006; Kearney et al., 2010; Kooijman, 2010; Sarà et al., 2011; 185 2013). DEB simulations were run under subtidal conditions (body temperature was expressed by the mean 186 seawater temperature; Montalto et al., 2014) with 4 years-hourly data (Jan 2006 - Dec 2009) of seawater 187 temperature measured about 1 m below the surface at the closest meteo-oceanographic station held in Trieste 188 (LAT 45° 38' 57.81"; LONG 13° 45' 28.58") and Palermo (LAT 38° 07' 17.08"; LONG 13° 22' 16.79"). The 189 period of 4 years is consistent with the normal life span of most Mediterranean shellfishes (Sarà et al., 2012; 190 2013b). Both sites were chosen as they represent two opposite temperature and food conditions for mussel 191 growth in Italy, with Trieste as representative of lower temperature (average  $16.98 \pm 6.19$  °C) and higher food 192 levels (average  $1.36 \pm 0.37$  CHL-a), and Palermo of higher temperatures (average  $20.19 \pm 4.64$  °C) and lower 193 food (average  $0.19 \pm 0.09$  CHL-a). Data are available online from the Italian Institute of Environmental Research 194 (ISPRA) web page (http://www.mareografico.it/).

195

196 2.6 CHL-a dataset. Chlorophyll a (CHL-a) was derived from satellite imageries (μg L-1;
 197 http://emis.jrc.ec.europa.eu/)) was and adopted as a reliable food quantifier for suspension feeders (Kearney et al., 2010; Sarà et al., 2011; 2012) and was downloaded from the EMIS website (http://emis.jrc.ec.europa.eu/).

199

200

201 2.7 Model description. The Dynamic Energy Budget (DEB) Theory provides a general framework that allows
202 to describe how physiological mechanisms are driven by temperature and food availability, and influences
203 growth and the reproductive performances in marine organisms (Sousa et al., 2010; Monaco et al., 2014; Jusup et

| 204 | al., 2017 ). Following the $\kappa$ -rule ( <del>DEB theory;</del> Kooijman, 2010) a fixed energy fraction ( $\kappa$ ) is allocated to growth    |
|-----|----------------------------------------------------------------------------------------------------------------------------------------------------------|
| 205 | and somatic maintenance, while the remaining fraction $(1-\kappa)$ is allocated to maturity maintenance plus                                             |
| 206 | maturation or reproduction. If the general environmental conditions deviates from common natural patterns (i.e.                                          |
| 207 | changes in temperature, food availability etc.) reproduction and growth are consequently affected. According to                                          |
| 208 | the DEB theory, a reduction in growth can be caused either by reduced food assimilation ( $\dot{p}_A$ ), enhanced                                        |
| 209 | maintenance costs $[\dot{p}_M](\dot{p}_{M})$ , or enhanced growth costs $(\dot{p}_G)$ . Using this approach, and through the DEB                         |
| 210 | parameters derived from Sarà et al. (2012)reported in Table 1, except for the variation in the maintenance costs                                         |
| 211 | $[\underline{\dot{p}}_{M}](\underline{\dot{p}}_{M})$ and in the assimilation efficiency of food (AE) which were experimentally estimated throughout this |
| 212 | study, we performed simulations using thea sstandard version of the DEB model (Nisbet et al., 2010) aimed a                                              |
| 213 | investigating the potential variations in growth and fecundity of our model species. To run the DEB simulations                                          |
| 214 | local thermal series of selected sites were used together with satellite CHL-a concentrations, obtaining a first                                         |
| 215 | model with environmental conditions. A second model was run with the $[\underline{\dot{p}}_M]\underline{\dot{p}}_M$ calculated from the oxygen           |
| 216 | measurements on specimens of M. galloprovincialis from Tanks 3-4 (pH 7.5) simulating a chronic hypercapnia                                               |
| 217 | condition for the full cycle (4 years) and the relative estimated AE. Subsequently, further models were run by                                           |
| 218 | simulating one random hypoxia event (duration = $30 \text{ days}$ ) for each of the four years of the cycle, then                                        |
| 219 | simulating two yearly events, and so on up to six monthly hypoxia events. The starting month of each event was                                           |
| 220 | randomly chosen for every year with the use of a table of random digits. The $[\underline{\dot{p}}_M]\underline{\dot{p}}_M$ calculated from the oxygen   |
| 221 | consumption rate measurements on specimens from Tanks 7-8 (pH 7.5 and hypoxia) was used in substitution to                                               |
| 222 | $[\underline{\dot{p}}_{M}]\underline{\dot{p}}_{M}$ from pH 7.5 tanks 3-4, coupled with the relative estimated AE, when simulating both stressors         |
| 223 | Simulations were performed using the R routine for Standard DEB model developed by M. Kearney (2012), and                                                |
| 224 | further modified (for use in bivalve modelling) by Sarà et al. (2013). Outputs of the DEB models (Sarà et al.                                            |
| 225 | 2014)-were: the maximum theoretical total length of shellfish (TL), the maximum total weight (TW), the total                                             |
| 226 | number of eggs (TRO) produced during a life-span of 4 years, the total number of reproductive events (RE) and                                            |
| 227 | the time needed to reach gonadic maturity (TM) for each treatment.                                                                                       |
| 000 |                                                                                                                                                          |

229 2.7.1 Model limitation. DEB models are particularly useful to quantitatively assess the effects of multiple
 230 stressors on LH-traits in an integrated manner, leading to test the hypothesis on how OA may affect the
 231 maintenance costs of living organisms (Jager et al., 2016). Maintenance costs, as defined by Dynamic Energy
 232 Budget Theory (Kooijman, 2010), represent the energy requirement of an organism to survive, excluding
 233 investments in growth, reproduction and development. The volume-specific somatic maintenance costs

[revised manuscript text omitted]

 $\begin{array}{l} (Tr2) \text{ and } 12.7 \pm 0.2 \ \% \ (Tr3) \text{ of opened valves (Fig. 1; Table 3, ANOVA, p < 0.001). The percentage of closed} \\ \text{valveswas instead } 35.5 \pm 5.6 \ \% \ (CTRL), 42.7 \pm 4.8 \ (Tr1), 75.5 \pm 5.7 \ (Tr2) \text{ and } 87.3 \pm 3.1 \ \% \ (Tr3) \ (ANOVA, p < 0.001). The percentage of closed valves (Fig. 1; Table 3, ANOVA, p < 0.001). -33.3 \pm 11.2 \ (CTRL), 50 \pm 4.5 \ (Tr1), 80 \pm 8.9 \ (Tr2) \text{ and } 83.3 \pm 6.1 \ \text{of opened valves (Fig. 1; Table 3, ANOVA, p < 0.001). The percentage of closed valves can be easily calculated as 100 - open valves. No significant differences resulted between week 1 and 4 \ (ANOVA, p > 0.05), between CTRL and Tr1 and between 203 Tr2 and Tr3. \\ \end{array}$